

# The thermal state of permafrost in under climate change on the Qinghai-Tibet Plateau from 1980 to 2022: A case study of the West Kunlun

**Jianting Zhao [1, 4], Lin Zhao [1, 2, 3∗], Ze Sun [1, 5], Guojie Hu [3], Defu Zou [3], Minxuan Xiao [1], Guangyue Liu [3], Qiangqiang Pang [3], Erji Du [3], Zhibin Li [1], Xiaodong Wu [3], Yao Xiao [3], Lingxiao Wang [1], Wenxin Zhang [4]**

[1]*School of Geographical Sciences, Nanjing University of Information Science &Technology, Nanjing 210044, China*

[2]*College of Resources and Environment, University of Chinese Academy of Sciences, Beijing 101408, China*

[3]*Cryosphere Research Station on the Qinghai–Tibetan Plateau, State Key Laboratory of Cryospheric Sciences, Northwest Institute of Eco-Environment and Resources, Chinese Academy of Sciences, Lanzhou 730000, China*

[4]*Department of Physical Geography and Ecosystem Science, Lund University, Lund 22362, Sweden*

[5]*School of Geography and Planning, Nanning Normal University, Nanning 530001, China*

**Abstract:** The thermal regime is a key indicator of permafrost evolution and thaw trajectories in response to climate change, yet it remains inadequately represented in global models. In this study, an efficient and integrated numerical model, the Moving-Grid Permafrost Model (MVPM) was used to simulate the permafrost thermal regime in West Kunlun (WKL), which is approximately 55,669 km² in northwest Qinghai-Tibet Plateau with extreme arid climate conditions. We employed clustering approaches and parallel computing techniques to enhance computational efficiency. The model forcing data, remote-sensing-based land surface temperature (LST) dating back to 1980 with a spatial resolution of 1 km×1 km and a temporal resolution of 1month, was constructed using machine learning techniques that integrate field observations, satellite data and reanalysis products. Our simulations achieved high accuracies of ±0.25°C for ground temperature and ±0.25 m for active layer thickness, significantly outperforming previous simulations reported to date. The results indicated that the WKL experienced a pronounced warming trend in LST, with an average increase of 0.40°C per decade from 1980 to 2022. The responses of the permafrost regime to climate warming were closely related to the original thermal conditions shaped by historical climatic evolution. These responses exhibited a distinct altitude-dependent spatial variation and differed according to soil stratigraphic types. Despite the thermal warming trend, the areal extent of permafrost remained relatively stable across the WKL region over the past 43 years, reflecting the slow and lagged response of permafrost to climate warming. These findings are essential for enhancing our

---

∗ Corresponding author: Lin Zhao (lzhao@nuist.edu.cn)



understanding of permafrost thaw trajectories, and improving projections of potential future consequences of permafrost degradation with greater accuracy.

## 1 Introduction

Permafrost covers about 40% of the Qinghai-Tibet Plateau (QTP), making it the largest high-elevation permafrost region in mid- to high-latitude areas, with an average altitude above 4000 m a.s.l (Zou et al., 2017). Observations of the ground thermal state provide clear evidence that permafrost warming has already led to thaw subsidence and widespread near-surface permafrost degradation in many areas on the QTP (Zhao et al., 2020, 2024; Biskaborn et al., 2019; Wang et al.,

2022; Simth et al., 2022). These changes potentially trigger climatic feedbacks on both local to global scales, significantly impacting ecosystem, infrastructure, and communities on QTP (Schuur et al., 2015; Walvoord et al., 2016; Lafrenière et al., 2019; Cheng et al., 2019; O'neill, et al., 2020; Jin et al., 2021; Miner et al., 2021; Hjort et al., 2022). Consequently, accurately evaluating and understanding characteristics of current permafrost dynamics in response to climate variability is

essential and highly valuable for assessing, predicting, and mitigating the impacts of climate change (Smith et al., 2022; IPCC, 2019, 2021).

     Over the past few decades, multiple field investigations have been conducted, and a monitoring network has been established on the QTP to monitor changes in permafrost thermal conditions (Zhao et al.,2010a, 2010b, 2017, 2019, 2021). Many of these sites include borehole sensor arrays, which

provide measurements of ground temperature as deep as 50 m or more below the surface (Zhao et al., 2019b, 2021). However, these observations are limited to discrete points, with most observation sites concentrated in easily accessible areas, such as along the Qinghai-Tibet Highway and Railway (QTH), leaving vast permafrost regions across the QTP, particularly in remote areas, largely unmonitored. To address this research gap, detailed process-based models have been widely

developed to simulate hydrothermal processes in frozen soils. However, most of these models still struggle to accurately represent the thermal state of permafrost, leading to substantial errors in predicting permafrost changes (Zhao et al., 2024). This is attributed to the simplified treatment of soil properties and thermal processes in deep permafrost, driven by a lack of detailed ground information and insufficient long-term in situ ground temperature monitoring in deep permafrost

(Sun et al., 2019; Zhao et al., 2020, 2024). Moreover, these models primarily focus on near-surface hydrothermal processes, typically constrained to the active layer within the upper 2–3 m. This limitation is particularly evident in large regional modeling at high spatial resolution, where the computational demands of applying numerical models over broader spatial scales and greater depths



are considerable (Smith et al., 2022). Such shallow simulations may fail to accurately capture
changes in the thermal state of areas with thicker, colder permafrost, whereas simulations extending
to greater depths more effectively represent the thermal response of permafrost to warming (Sun et
al., 2019; Zhao et al., 2020). In addition, uncertainties in the forcing datasets also contribute to biases
in the simulation results of hydrothermal processes in frozen ground (Yi et al., 2018; Guo et al.,
2017; Hu et al., 2023). Previous studies have demonstrated that soil temperature projections from
Earth system models (ESMs) outputs of the Coupled Model Intercomparison Project Phase 5 and 6
(CMIP5, CMIP6) tend to overestimate future permafrost changes (Koven et al., 2013; Lawrence et
al., 2012; Slater and Lawrence, 2013; Burke et al., 2020). When air temperature and precipitation
inputs to land surface models are improved, the estimated permafrost degradation rates decrease by
approximately 29% (Lawrence et al., 2012). This highlights the necessity for high-resolution and
more accurate model forcing datasets.

To address model deficiencies, we developed a new numerical permafrost model known as the
Moving-Grid Permafrost Model (MVPM; Sun et al., 2019; 2022) in consideration of variations in
thermal properties between frozen and thawed soil, unfrozen water content in frozen soil, ground
ice distribution, thaw settlement of ground surface, and geothermal heat flow—factors often
overlooked in existing models (Sun et al., 2022; Zhao et al., 2024). It was used to simulate the heat
transfer processes effectively by capturing both the attenuation and time lag of heat transfer in deep
permafrost at several borehole sites and limited regions along the QTH (Sun et al., 2019, 2022, 2023;
Zhao et al., 2022), and showed sufficient accuracy to resolve the annual dynamics of ALT and
refreezing, as well as the evolution of ground temperatures in deeper layers, compared with long-
term continuous soil temperature monitoring at various depths and ALTs.

The simulation results are more closely related to the spatial resolution and accuracy of input
dataset. Several studies have utilized the gridded datasets derived from in situ meteorological
measurements at meteorological stations, climate outputs from general circulation models, or Earth
System Models (ESMs) as inputs to simulate soil thermal regime dynamics over large spatial scales
in the Circum-Arctic permafrost region (Jafarov et al., 2012; Westermann et al., 2013; Zhang et al.,
2014; Fiddes et al., 2015). However, large uncertainties remain in these climate forcing datasets due
to the harsh climate, complex terrain, and limited observations on the permafrost of QTP, making it
challenging to use these datasets to accurately drive the MVPM for simulating permafrost regimes
on the QTP (Hu et al., 2019; Qing et al., 2020; Yang et al., 2020). Moreover, gridded outputs from
ESMs are typically provided at a spatial resolution of half-degree latitude/longitude or coarser,
which cannot adequately capture the high spatial variability of the ground thermal regime in



permafrost areas of the QTP (Zhang et al., 2013; Hu et al., 2023). In contrast, satellite remote sensing enables regional detection and monitoring of land surface conditions related to permafrost thermal properties (Langer et al., 2013). Regionally refined, satellite-driven numerical models provide a

promising approach for assessing permafrost thermal state dynamics with higher spatial and temporal resolution (Westermann et al., 2015, 2017; Yi et al., 2018). This satellite-based concept for permafrost thermal state modeling and assessment has been successfully applied in various permafrost regions worldwide, including Alaska (Yi et al., 2018), Siberia (Langer et al., 2013; Westermann et al., 2017), and Canada (Zhang et al., 2013). All these model results demonstrate that

numerical modeling using remote-sensing data holds great potential for high-resolution permafrost monitoring on a regional scale. In the QTP permafrost region, Zou et al. (2017) and Cao et al. (2023) mapped permafrost spatial distribution using the Moderate Resolution Imaging Spectroradiometer (MODIS) Land Surface Temperature (LST) product as input data for an equilibrium model. Similarly, in our previous simulations, Zhao et al. (2022) evaluated and demonstrated the

performance of the MVPM modeling scheme at a spatial resolution of 1 km, driven by a time series of MODIS LST data for a localized region (less than 280 km²) on the QTP.

In this study, we aim to enhance and expand the MVPM to facilitate accurate, large-scale mapping of the ground thermal regime and its spatiotemporal changes in response to recent climate changes. We develop an integrated framework that combines numerical modeling, field observations,

remotely sensed data, and reanalysis data, allowing simulation of permafrost thermal distribution at a 1 km×1 km spatial resolution. We used a geomorphological map and field measurements to parameterize soil properties. To address the computational challenges of large-scale thermal modeling, we employ a clustering approach that groups climate forcing and soil thermal properties into discrete types, and we implement parallel computing to enable the efficient simulation of tens

of thousands of grid cells within a reasonable time frame. The MVPM was then run over a 43-year period (1980–2022) across WKL (Fig. 1) in the northwest QTP, where field measurements of ground temperature and ALT are available for validating simulation results. Finally, we aimed to quantitatively analyze the spatiotemporal dynamics of the thermal regime across different environmental settings under climate change from 1980 to 2022 based on the modeling output.

**2 Study area**

The WKL permafrost survey area (78.8–81.4°E, 34.5–36.0°N), is situated in the northwest of the QTP, with an elevation range of 4200-6200 m a.s.l (see Fig. 1a) and spans approximately $4.37×10^3$ km² (Zhao et al., 2019b). This region has a cold and arid continental climate, as the Pamir-



Tian Shan-Kunlun Mountains act as an orographic barrier, limiting moisture influx from both the
westerlies and monsoons (Cannon et al., 2016; Baldwin and Vecchi, 2016). Meteorological
observations from the automatic weather station (AWS) at Tianshuihai (TSH, 81.4°E, 36.0°N, at
5019 m a.s.l) for the period 2015-2018 report a mean annual temperature of ~-6°C and mean annual
precipitation of ~103.5mm (Zhao et al., 2021). Over 78% (~81mm) of this precipitation falls during
May-September, and the summer temperature stays above zero degrees (~5.8°C) (Zhao et al., 2021).
The glacial and periglacial landforms (e.g., block fields, stripes, and stone rings) are well developed
(Wu et al., 2018). Typically, the vegetation is dominated by sparse alpine desert, and the most of
exposed landforms are barren due to wind erosion (Li et al., 2012; Wang et al., 2016; Zhao et al.,
2019). The topsoil was dry and loose, primarily composed of Quaternary deposits resulting from
aeolian erosion (57.68%, see detail in table 1), which consist of coarse-grained sediments such as
gravel and sand (see Figure 1b). Permafrost in WKL is well-developed in WKL, encompassing both
discontinuous and continuous types, and the areal extent occupying approximately 93% of the total
area (Li et al., 2012; Zhao et al., 2019). Continuous monitoring of a borehole at the TSH
comprehensive observatory (ZK015, 59 m in depth, 79.54°E, 35.36°N, see Fig. 1b) indicates strong
changes in the thermal state of permafrost, with a warming rate of 0.11°C per decade between 2010
and 2017 at a depth of 15 m (Zhao et al., 2021, Hu et al., 2023).

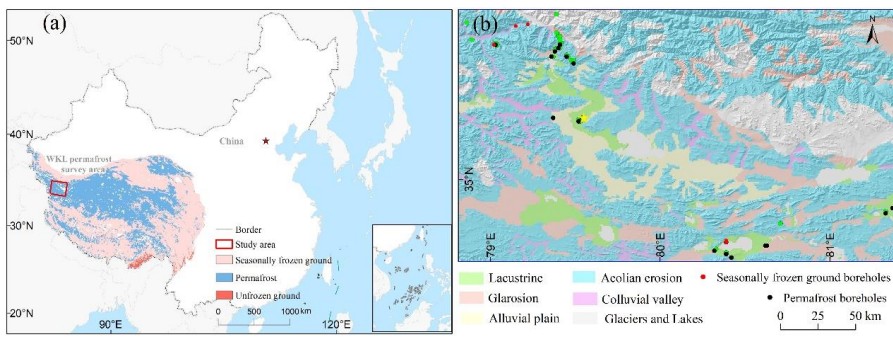

**Figure 1. (a) The geographical location of the West Kunlun (WKL) permafrost survey
area is overlaid on the frozen ground type distribution map from Zou et al. (2017) and the
background maps based on Wen et al. (2024). (b) The WKL permafrost survey area mainly
includes five stratigraphic classes distinguished in the ground thermal modelling (Sect. 3.2.2)
and boreholes with in situ observations (Sect. 3.3), which are used for model calibration and
validation. The yellow star indicates automatic weather stations (AWS); red dots denote
monitoring boreholes in seasonally frozen ground; black dots indicate monitoring boreholes
located in permafrost; and green dots represent thaw depth measurements obtained using**



**ground-penetrating radar (GPR) technology.**

**3 Methodology and data**

**3.1 The Moving-Grid Permafrost Model**

The Moving-Grid Permafrost Model (MVPM) is used to simulate the permafrost thermal regime and its dynamics under climate change. This is a transient, one-dimensional, nonlinear heat

transfer model that accounts for phase changes in soil pore water, variations in thermal properties between frozen and thawed soil, geothermal flux, and the effects of ground ice. The model also includes a settlement module, which was not part of our previous model configuration (Zhao et al., 2022). Soil temperature dynamics are simulated by numerically solving the one-dimensional nonlinear conductive heat equation using the finite difference method (Schiesser, 1991; Westermann

et al., 2013; Sun et al., 2019). The MVPM has recently been used to simulate permafrost thermal dynamics at several borehole sites across the QTP (Sun et al., 2019, 2022, 2023; Zhao et al., 2022). Detailed descriptions of the model and its parameterization are provided in these companion studies.

**3.2 Model operation**

3.2.1 Model forcing

Similar to our previous study (Zhao et al., 2022), a time series of remotely sensed LST was used to drive the MVPM. Specifically, an 8-day mean LST dataset covering the entire QTP, with a spatial resolution of 1 km × 1 km (hereafter referred to as "LST_Zou" ), was used for the period from 2003 to 2019, provided by Zou et al. (2014, 2017). This dataset was created by integrating in situ observations with satellite-based LST from the Moderate Resolution Imaging

Spectroradiometer (MODIS). In this work, to fully account for the impact of historical climate on the thermal regime evolution of the WKL permafrost region, three statistical algorithms were employed to extend the LST_Zou dataset back to 1980: the least squares linear regression (LR) model (Xing et al., 2023), the random forest regression (RFR) model (Breiman et al., 2001), and the multiple linear regression (MLR) model (Jiao et al., 2023). The LR model assumes a linear

relationship between air temperature (AT) and LST over long-term periods. For the RFR and MLR models, we considered eight auxiliary explanatory variables that influence LST: air temperature (AT), precipitation (Pre), skin temperature (ST), soil temperature in the topsoil layer (0–10 cm) (ST_1), fractional cloud cover (CFC), surface radiation budget (SRB), leaf area index (LAI), and digital elevation model (DEM). Details on these variables, including their resolution and relevant

references, are provided in Table 1. The main steps for reconstructing monthly LST from 1980 to



2022 are as follows:

(1) Pre-processing

All input variables were first resampled to a 1 km × 1 km resolution to match that of LST_Zou using the nearest neighbor method. Monthly averages were then calculated from daily data, and missing values were filled by interpolating nearby data. Notably, the latest downscaled AT and Pre data provided by Qin et al. (2022) are only available up to 2019. Therefore, we performed statistical downscaling (Su et al., 2016) of AT and Pre from the CN05.1 datasets for the years 2020–2022. CN05.1 is a gridded dataset provided by the China Meteorological Administration (CMA), which includes key meteorological variables with a spatial resolution of 0.25 degrees and a daily temporal resolution. For further details, refer to the comprehensive description by Wu et al. (2017). Additionally, since the earliest available remote sensing-based LAI data begins in 1982, we assumed no changes for the initial two years and averaged data from 1982 to 1986 to fill in the gaps.

(2) Model training and test

Data from 2003 to 2019 were selected as the training sample for the models. In the LR model, AT was used as the input variable, and LST_Zou served as the output variable. For the MLR and RFR models, eight auxiliary variables (see Table 1) were used as inputs, with LST_Zou as the output. The dataset was randomly divided into ten subsets: one subset, comprising 10% of the samples, was retained for validation, while the remaining nine subsets (90% of the samples) were used for model training. This process was repeated 2,000 times. The model's accuracy was assessed using four metrics: the coefficient of determination ($R^2$), root mean square error (RMSE), mean absolute error (MAE), and bias (Zhao et al., 2022). See Fig. 3 for details.

(3) Dataset generation

The monthly values of the eight auxiliary variables (see Table 1) from 1980 to 2022 were used as input variables for the three models (LR, MLR, and RFR) established in step (2), enabling the generation of a time series of monthly LST starting from 1980.

**Table 1: Summary of the data sources used for the linear regression model (LR), random forest regression model (MLR), and multiple linear regression model (RFR) to generate monthly land surface temperature from 1980 to 2022.**

| Variable | Data span | Resolution and | Data resource and availability | Reference |
|---|---|---|---|---|



| name | | Horizontal coverage | | |
|---|---|---|---|---|
| LST_Zou | 2003-2019 | 8-day (QTP) | -- | Zou et al. (2017) |
| AT Pre | 1961-2019 | Daily/1km×1km (China) | https://doi.org/10.1594/ PANGAEA.941329 | Qin et al. (2022) |
| ST | 1950-present | Hourly/~9km×9km (global) | ERA5-Land Reanalyst https://cds.climate.copernicus.eu/ datasets/reanalysis-era5- land?tab=overview/cdsapp#!/dataset /reanalysis-era5-land?tab=overview | Muñoz-Sabater et al. (2021) |
| ST_1 | 1979-present | 6 hour /0.312°×0.312°/ 0.204°×0.204° (global) | NCEP Climate Forecast System Reanalysis (CFSR) https://rda.ucar.edu/datasets/ ds093.0/dataaccess/ | Saha et al. (2010) |
| CFC SRB | 1979-present | Monthly/ 0.25°×0.25° (global) | EUMETSAT, CM SAF https://wui.cmsaf.eu/safira/action/view DoiDetails?acronym=CLARA_AVHRR _V003 | Karlsson et al. (2023) |
| LAI | 1982-2022 | 8-day/0.05° from AVHRR, 500 m from MODIS (global) | Global Land Surface Satellite (GLASS) and MODIS http://www.glass.umd.edu/ https://modis.gsfc.nasa.gov/data/datapr od/mod15.php | Liang et al. (2020) |
| DEM | -- | 90m (global) | SRTM/https://cgiarcsi.community/data/ srtm-90m-digital-elevation-database- v4-1 | Jarvis et al. (2008) |
| Glacier | -- | -- | National Snow and Ice Data Center https://nsidc.org/data/nsidc 0770/versions/6 | Guo et al. (2015) |
| Lakes | -- | -- | National Tibetan Plateau Data Center https://data.tpdc.ac.cn/ | Zhang et al. (2019) |

Note: LST_Zou is an enhanced land surface temperature (LST) product for the QTP permafrost zone, derived from
in situ observations and MODIS satellite data. AT refers to air temperature; Pre to precipitation; ST to skin

temperature; ST_1 to soil temperature at the top layer (0–10 cm); CFC to fractional cloud cover; SRB to surface radiation budget; LAI to leaf area index; and DEM to Shuttle Radar Topography Mission (SRTM) digital elevation model data.

3.2.2 Ground thermal properties

The stratigraphy plays a crucial role in determining variations in lithology, mineralogy, water retention, and ice content in permafrost regions (Wang et al., 2018; Sheng et al., 2019). These factors significantly influence ground thermal dynamics, controlling both soil temperature and the permafrost thermal regime (Westermann et al., 2017). In this study, soil thermal properties were parameterized using a vector geology map from the digital geomorphological database of western China, at a scale of 1:1,000,000 (Zhou and Cheng, 2007), along with ground temperature and thaw depth measurements. We identified five predominant stratigraphies within the WKL permafrost survey area, for which we defined "typical" subsurface stratigraphies based on available ground temperature observations and thaw depth measurements (Sect. 3.3). A list of major Quaternary deposits is provided in Table 2, and their spatial distribution (Fig. 1b) is based on Zhou et al. (2007), gridded to 1 km × 1 km to match our simulation resolution.

The ground thermal properties of different stratigraphic classes are estimated using observations from a representative borehole. Four of the five stratigraphic classes are represented by two or more in situ measurement sites (see details in Table 2). However, no ground temperature or ALT measurements are available for the alluvial plain. Therefore, we use properties from the lacustrine plain class as a reference. Thermophysical properties (e.g., lithological composition, ground ice content, organic matter content, dry bulk density) of distinct soil profile layers were measured or assessed through field surveys, laboratory measurements, and on-site analyses of soil samples obtained from 28 borehole cores (depths ranging from 7.5 to 59 m). Detailed information about the boreholes and soil samples can be found in Li et al. (2012), Li et al. (2014, 2015), and Zhao et al. (2019). Additionally, a 1 km × 1 km resolution ground ice distribution map (down to a depth of 10 m) by Zou et al. (2024), and data on the vertical distribution of ice content for various sedimentary classes on the QTP based on representative borehole cores from field investigations and geological surveys (Zhao et al., 2010), were used to estimate water content for each stratigraphic class.

Depth-specific soil profile thermophysical parameters (thermal conductivity and heat capacity) for each stratigraphic class were estimated by calibrating the modeled permafrost temperature and thaw depth against observed data. Calibration of ground thermal properties was performed using a





numerical solution to an inverse problem, which aimed to minimize the discrepancy between simulated and observed ground temperatures by adjusting thermal properties (Marchenko et al.,

2024; Nicolsky et al., 2016). This calibration method is described in detail in Nicolsky et al. (2007, 2016), with an example of soil thermal properties optimization procedures provided in Zhao et al. (2022) and Marchenko et al. (2024).

**Table 2: List of major geological classes and their association with borehole measurement sites. The second column shows the percentage of the study area represented by each type of**

**Quaternary sediment, while the third column lists the corresponding 'typical' boreholes.**

| Quaternary sediments type | Percent % | Boreholes |
|---|---|---|
| Aeolian | 57.68 | ZK01, ZK02, ZK04, ZK12, ZK13, ZK16, K514+950, K520+050, |
| Glarosion | 12.58 | ZK06, ZK07 |
| Alluvial plain | 5.96 | ZK08 |
| Lacustrine plain | 5.05 | ZK14, ZK15, ZK17, ZK18, ZK30, ZK31 |
| Colluvial valley | 3.67 | ZK09, ZK04 |
| Modern Glaciers | 12.54 | Excluded from the model |
| Lakes | 2.52 | Excluded from the model |

3.2.3 Model computational domain, boundary condition and initialization

The model's resolution is determined by the resolution of the input data. Specifically, the computational simulation domain has a spatial resolution of 1 km × 1 km and temporal resolution of monthly, covering the entire WKL permafrost survey area, which contains approximately 55,669

km². Similar to the setup used in our previous simulations (Zhao et al., 2022), the vertical computation for each spatial grid cell extends to a depth of 100 m and is discretized into 282 layers. Specifically, the layer thickness ranges from 0.05 m in the upper 4 m to 0.5 m for the remaining soil layers down to 100 m. For each modelling grid cell, the ground thermal regime was simulated using specific ground stratigraphy and forcing time series of LST. At a depth of 100 m within the soil

column, the Neumann lower boundary condition was applied to specify the geothermal heat flux. A constant geothermal heat flow of 0.0724 W/m² was assumed, based on measurements from a 700 m deep borehole near the WKL permafrost survey area (see details in Hu et al., 2000).To estimate a realistic initial temperature profile, a model spin-up is performed to achieve steady-state conditions based on the forcing in the initial model years. The criterion for reaching a steady-state soil



temperature profile is an equilibrium temperature difference of less than 0.0001°C at all soil levels between successive annual cycles. This steady-state profile is then used as the initial condition for subsequent modelling.

### 3.2.4 Model implementation

Excluding lake and glacier grids, approximately 47,284 grid cells were used for computation
in the study area. To reduce model running time, we employed a spatial cluster analysis approach (Cable et al., 2016) to group cells based on climate forcing and soil thermal properties. This method allowed us to simulate these clusters rather than processing each pixel individually. Specifically, we mimic the variation in upper boundary condition LST series using a harmonic function (see Sun et al., 2019 for details). Based on the fitted coefficients (initial annual mean temperature, change rate,
annual amplitude, and initial phase angle), we categorized the climate forcing of grid cells into several distinct clusters. Combined with five soil thermal property classes (Table 1), this resulted in 13,248 unique input-data combinations for the WKL permafrost survey area. These combinations account for only 28.02% of the total model grid cells, remarkable reducing computation time. Similar spatial cluster approaches have been applied in Canada (Zhang et al., 2013, 2014), Alaska
(Cable et al., 2016), and the Swiss Alps (Fiddes et al., 2015).

### 3.2.5 Simulation results diagnose

For analysing changes in the thermal state of permafrost over the past 43 years, we use key diagnostics from the modelled vertical ground temperature profile down to 50 m depth to represent the thermal state of permafrost and the active layer. Specifically, we focus on the mean annual
ground temperature at a depth of 15 m (MAGT 15m), which corresponds to the depth of zero annual amplitude (ZAA) on the QTP (Jin et al., 2008; Zhao et al., 2010b), as well as the temperature at the top of permafrost (TTOP). We also consider other depths where we compare our simulations with borehole observations. To assess changes in ALT, we calculated ALT using linear interpolation to determine the maximum depth of the 0°C isotherm over the thawing period of the year (Liu et al.,
2020). Following Zhao et al. (2022) and Wu et al. (2018), model grid cells were classified as permafrost if the maximum temperature of any soil layer at a grid point was ≤ 0°C for two consecutive years. Seasonally frozen ground was identified from the remaining cells where the minimum soil temperature of any layer during the same period was ≤ 0°C. Cells that did not meet either criterion were classified as unfrozen ground.

**3.3 Filed investigation and borehole monitoring datasets**



Extensive scientific research and monitoring programs have been conducted in WKL over the past two decades. A comprehensive monitoring system has been established WKL by the Cryosphere Research Station, Chinese Academy of Sciences (CRS-CAS) (Zhao et al., 2015, 2019b, 2021). These detailed data sets significantly enhance our understanding of processes and support

modeling efforts in the WKL (Li et al., 2012; Zhao et al., 2017, 2019, 2021). Below, we briefly describe the in-situ data sets from CRS-CAS used in this study.

### 3.3.1 The Tianshuihai (TSH) comprehensive observatory

The TSH comprehensive observatory is located in the central-northern part of the WKL permafrost survey area (see Fig. 1a). The Quaternary deposits in this region are primarily lacustrine,

consisting of fine-grained sediment from an ancient lake that dried up in the Lower Pleistocene (Li et al., 1991). Since October 2015, an automatic AWS has been measuring key meteorological variables, including hourly air temperature at 2 m, 5 m, and 10 m, relative humidity, upward and downward shortwave and longwave radiation, wind speed, and precipitation. Additionally, ground temperature data have been automatically recorded since 2010 at depths of 3 m, 6 m, 10 m, and 20

m from a 59 m deep borehole (ZK015, 79.54°E, 35.36°N, see Fig. 1b) (Zhao et al., 2021). The LST at TSH is estimated using a continuous series of measurements (since October 2015) of downward and upward longwave radiation, applying the Stefan-Boltzmann law (see Hu et al., 2024 for details). This provides a robust basis for validating satellite datasets and ground thermal modeling (see Sect. 4.1.1).

### 3.3.2 Borehole in situ data sets

Outside the TSH comprehensive observatory, 27 boreholes, drilled to depths ranging from 7.5 to 33 m, have been installed across the WKL permafrost survey area to monitor the thermal regime (see Fig. 1b). These boreholes are distributed across various geomorphic, soil, and vegetation conditions, spanning elevations from 4200 to 5200 m. Detailed information about these boreholes

is provided in Zhao et al. (2019) and Li et al. (2012). Thermistor sensors, calibrated to an accuracy of 0.1°C, are strategically placed along the cables of these boreholes at depths of 3 m, 6 m, 10 m, and 20 m (Zhao et al., 2021). Additionally, ground temperatures have been manually measured using a digital multimeter at intervals of 1 or 2 years since 2010 in 15 boreholes with diverse ground surface conditions. For this study, we used 15 boreholes with ground temperature observations for

model calibration and validation. The remaining boreholes served as additional evidence for evaluating the presence or absence of permafrost in spatial permafrost distribution modeling.

### 3.3.3 Thaw depth measurement data sets



During fieldwork in September 2010, when the maximum thaw depth was reached, 45 manual thaw depth measurements were conducted using GPR technology. Most of these measurements were taken near drill boreholes (see Fig. 1b). The thaw depth measurements from summer 2010 are thoroughly documented in Zhao et al. (2019). After excluding measurements that overlapped within a 1 km grid, we used 25 distinct thaw depth measurements for model validation.

### 3.4 Additional validation datasets

In addition to validating the site-based observations (Sec. 3.3), we also evaluated our model's performance in permafrost distribution over the WKL permafrost survey region using four typical permafrost distribution maps from different periods. These maps include: i) a 1:3,000,000 scale permafrost map of the QTP compiled by the Lanzhou Institute of Glaciology and Geocryology, CAS (Li and Cheng, 1996); ii) a comprehensive 1:4,000,000 scale map of glaciers, permafrost, and deserts in China created by the Cold and Arid Regions Environmental and Engineering Research Institute, CAS (Wang et al., 2006); iii) a 2010 permafrost distribution map of the QTP (Cao et al., 2023); iv). A 2016 permafrost distribution map of the Tibetan Plateau (Zou et al., 2017). Both of the latter two maps have the same spatial resolution of 1 km².

### 4 Result

### 4.1 Forcing dataset

4.1.1 Comparison to in situ data

We implemented and compared three algorithms from Section 3.2.1 to identify the optimal model for reconstructing monthly LST data from 1980 onward and the validation results are shown in Figure 2. The majority of the samples in the scatter plots clustered near the 1:1 line, demonstrating a strong positive correlation ($R^2 > 0.90$), which indicates good agreement between the LST_Zou and the estimated LST values. The LR model yielded MAE and RMSE values of 2.05℃ and 2.61℃, respectively. The MRL model showed slight improvement, with lower error metrics (MAE = 1.16℃, RMSE = 1.55℃). However, the RFR model outperformed the others, producing the lowest errors (MAE = 0.87℃, RMSE = 1.26℃).





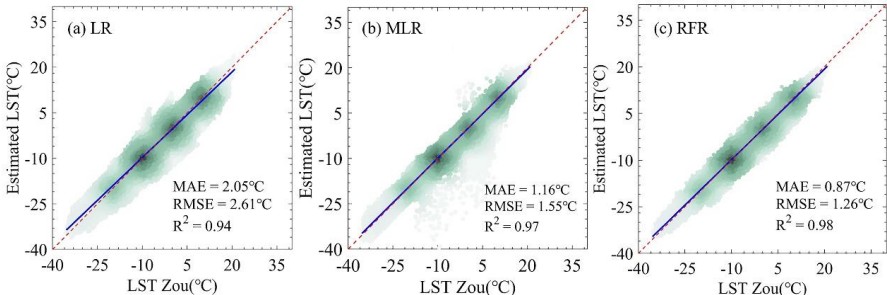

**Figure 2: Scatterplots of estimated monthly LST using (a) LR: linear regression model, (b) MLR: multiple linear regression model, and (c) RFR: random forest regression model during the validation stage (10-fold cross-validation; see details in Sect. 3.2.1). The best linear fits are shown in blue, while the 1:1 line is represented in red. Error metrics are provided in the bottom right corner of each graph.**

Figure 3 compares the mean annual cycle of LST estimated by the three statistical models (LR, MLR, and RFR) with LST_Zou and in situ monitoring data from the TSH AWS for the period 2016–2018. All four datasets display a consistent mean annual cycle with the observations, although LST_Zou shows a systematic cold bias, particularly during July, August, and September. The monthly LST estimated from the three algorithms slightly reduce this cold bias, with the RFR model performing the best. However, a residual cold bias in LST_Zou remains noticeable in the same months. Overall, the LST time series generated by the RFR model closely aligns with in situ measurements and demonstrates sufficient accuracy for subsequent ground thermal modeling. Therefore, the RFR-derived monthly time series was used as input for the ground thermal modeling in the following sections.

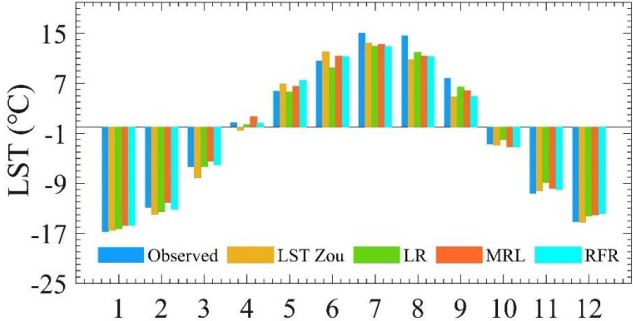

**Figure 3: Satellite-derived LST (LST_Zou), estimates from three different algorithms (LR, MLR, and RFR), and measured at the TSH AWS for the monthly average during periods when**





**in situ measurements are available (see Sect. 3.3.1).**

4.1.2 Spatiotemporal variability of forcing datasets

Figure 4 shows the regional average of annual LST anomalies relative to the 1980–2022 mean. The results reveal a consistent positive trend of +0.40°C per decade over the WKL region during this period. Interdecadal analysis highlights a remarkable warming trend in the mid-1980s, which then slowed slightly from the 2000s, during which LST deviations were relatively smaller. In the last decade, only positive anomalies were recorded, with 2016 exhibiting the largest positive

deviation (+1.45°C) compared to the 1980–2022 climate average.

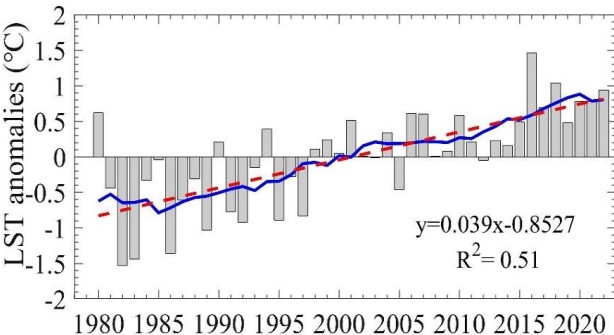

**Figure 4: Time series of regional average annual LST anomalies in the WKL permafrost survey area from 1980 to 2022. The 9-year moving average is depicted with a blue line, while the linear trend, calculated using standard linear regression (with long-term changes based on**

**the slope of the regression), is shown with a red dashed line. The anomalous LST series are obtained by subtracting the mean LST from 1980 to 2022.**

      To further examine the regional distribution of LST anomalies in WKL, Figure 5 presents the spatial patterns of anomalous LST for four decades (1980s, 1990s, 2000s, and 2010–2022) relative to the 1980–2022 average. In the 1980s, most of the WKL (63.25%) exhibited negative anomalies

between -1.5°C and -0.5°C, while 36.3% had anomalies between -0.5°C and 0°C. A small fraction (0.46%), primarily in high-elevation mountainous areas, experienced the largest negative anomalies (-2.0°C to -1.5°C). From the 1980s to the 1990s, a drastic warming occurred across WKL, with 90.95% of the region showing anomalies between -0.5°C and 0°C in the 1990s. By the 2000s, 11.87% of WKL experienced negative anomalies between -0.5°C and 0°C, while 83.78% saw positive

anomalies between 0°C and 0.5°C. Notably, 4.34% of the region recorded the highest positive anomalies (0.5°C to 1.0°C). From 2011 to 2022, a pronounced rise in LST was observed, with 63.97% of the region experiencing anomalies greater than 0.5°C, and some areas, particularly in high-





elevation mountainous regions, exceeding 1.0°C.

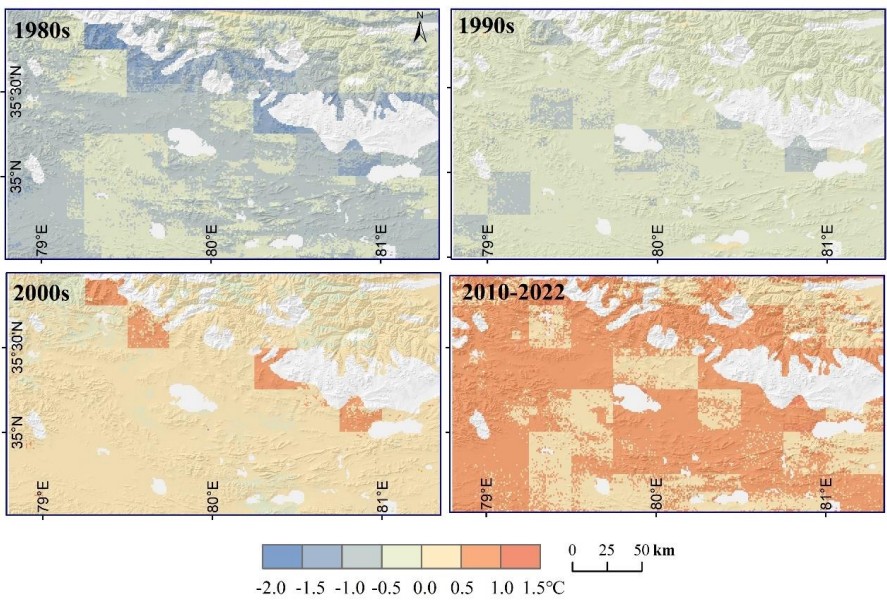

**Figure 5: Maps of decadal anomalous LST patterns over the WKL permafrost survey area for the 1980s, 1990s, 2000s, and 2010–2022, relative to the 1980–2022 average. The anomalous LST patterns are calculated by subtracting the mean LST of 1980–2022. Glaciers and lakes are excluded and shown in white.**

**4.2 Modeled the thermal state of permafrost**

4.2.1 Model validation

To ensure that the model accurately represents large-scale ground thermal conditions, the model outputs were compared with available in situ datasets (Fig. 1, Sect. 3.3). This comparison includes ground temperature measurements at a depth of 10 m (MAGT10m) from 15 sites and ALT data from 11 sites in year 2010, and thaw depth measurements from 25 sites in the same time, as well as four permafrost distribution maps from various periods.

*Ground temperatures:* The comparison between observed and modeled MAGT10m at 15 permafrost boreholes shows that 93.3% (14/15) of the data points cluster around the best-fit line, with deviations within ±0.25°C (Fig. 6a). The analysis indicates a strong overall agreement between measured and modeled MAGT10m for temperatures above -1°C, with an error of 0.10°C or less. However, for MAGT10m below -2°C, the model exhibits a slight cold bias, particularly in areas with lacustrine sediments in the lowland regions of central WKL, where ground temperatures vary





drastic due to complex local factors (Fig. 6b). Despite this, the deviations between observed and simulated temperatures within 0.3°C. Overall, the comparison suggests that the MVPM effectively replicates the measured MAGT10m, capturing the spatial variability of the validation area with an

r of 0.98 (p < 0.01), and achieving an MAE of 0.12°C and 0.15°C.

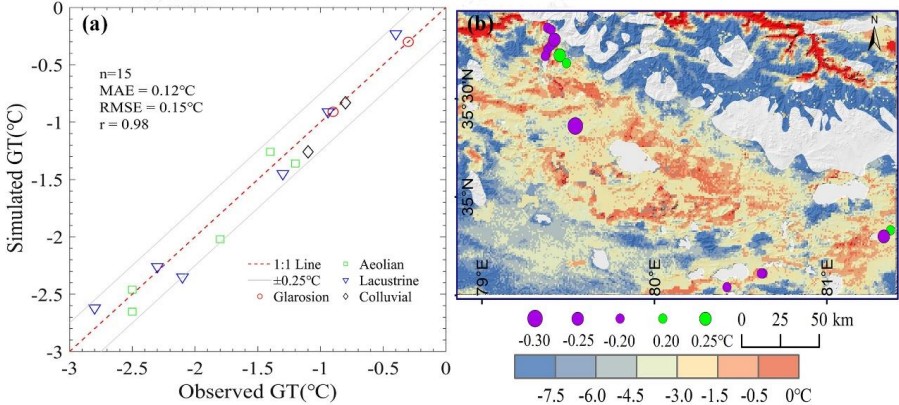

**Figure 6: (a) Scatter plot comparing borehole-observed (Zhao et al., 2019b; Li et al., 2012) and modeled mean annual ground temperature at a depth of 10 m (MAGT10m) for the year 2010. Different symbols represent various soil stratigraphic classes (Glarosion, Aeolian,**

**Lacustrine, Colluvial). Grey shading indicates biases within ±0.25°C, and the 1:1 line is shown in red. (b) Map showing spatial distribution of the modeled MAGT10m for 2010. Circle sizes and colors indicate the temperature differences between borehole-observed and modeled MAGT10m at the nearest 1 km grid point. Seasonally frozen ground is marked in red, and glaciers and lakes are depicted in grey.**

*Active layer thickness (thaw depths)*: The scatter plot and spatial map comparing measured and modeled ALT at 11 sites and thaw depths at 25 sites are shown in Figure 7. The comparison indicates that the model generally reproduces the range of ALT across WKL effectively. Simulated values for 72.7% (8/11) of the sites align closely with the best fit, exhibiting deviations of ±0.25 m from the measurements (Fig.7a). Notably, for the Aeolian sediment class, which has small ALT

values around 2 m, the agreement is excellent, with a relatively small bias of 0.05 m or less, indicating that the modeling procedure is suitable for Aeolian sediments. However, the model underestimates ALT by 0.25 m in lacustrine sediments near lake areas, where measured ALT exceeds 3 m (Fig. 8c). A similar pattern is occurred in thaw depths, where 91.3% (21/23) of the values cluster around the best fit, with deviations of ±0.25 m (Fig. 7b). For thaw depths above 3 m, the model also





shows underestimation, with the largest deviations reaching up to 0.5 m in northern marginal
        permafrost areas (Fig. 7c). Overall, although there are relatively higher biases above 0.25 m at
        certain locations, the comparison suggests that the current model can realistically reproduce the
        differences in ALT (thaw depth) between the main geomorphological units in the WKL region,
        achieving an r value of 0.96 (0.94) and an MAE of 0.13 m (0.16 m) for ALT and 0.16 m (0.18 m)
for thaw depth.

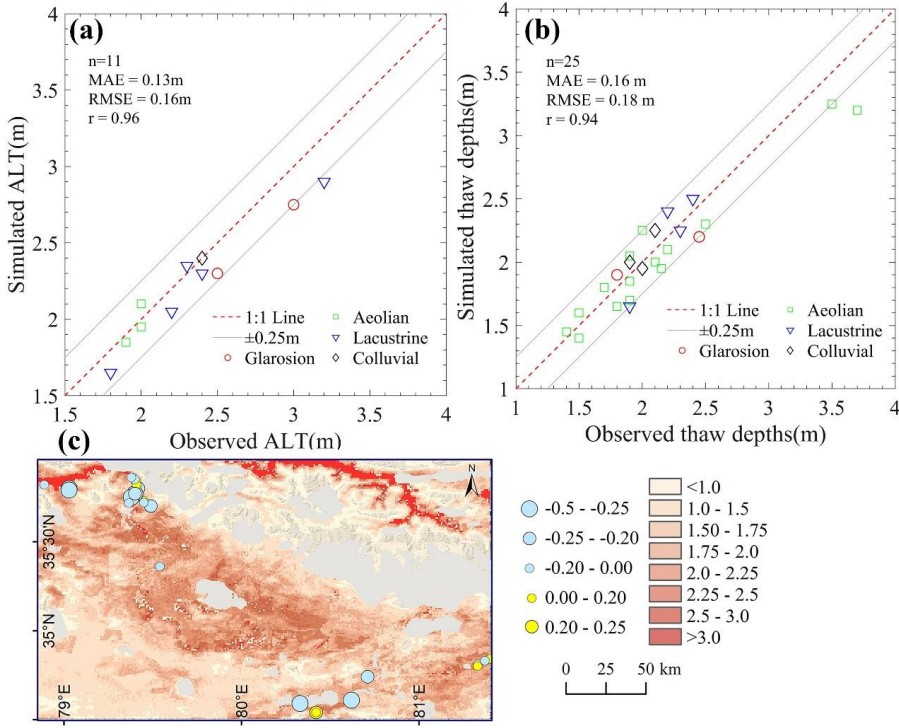

**Figure 7: (a) Scatter plot comparing borehole- observed active layer thickness (ALT) in 2010 (Zhao et al., 2019b; Li et al., 2012) with modeled values. (b) Same as (a), but for thaw depths (Zhao et al., 2019). Biases within ±0.25 m are indicated in grey, and the 1:1 line is shown in red. (c) The modeled spatial distribution of ALT and thaw depths in 2010. The sizes and colors of the circles represent the deviation between borehole (GPR) observed ALT (thaw depths) and the corresponding model grid cell nearest to each observation site (within 1 km). Seasonally frozen ground is depicted in red, while glaciers and lakes are shown in grey.**


        *Permafrost distribution*:Figure 8 compares four typical frozen soil type maps over WKL with
the corresponding our simulation outputs. In this analysis, 28 boreholes (see details in Fig. 1 and





Sect. 3.3) serve as evaluation points to verify the presence or absence of spatial permafrost distribution. The results indicate that while the maps by Li and Cheng (1996) and Wang et al. (2006) can identify permafrost distribution across the WKL region, they fail to accurately delineate areas of seasonally frozen ground (Fig. 8a-c). Additionally, these two maps show notably discrepancies in the distribution of frozen ground types in northeastern WKL, where permafrost is depicted, while our modeled outputs identify seasonally frozen ground (Fig. 8i-j). In contrast, the maps by Cao et al. (2023) and Zou et al. (2017), along with our simulations, consistently display an accurate pattern of permafrost distribution, correctly identifying nearly all locations of permafrost and seasonally frozen ground, except for one site near lakes in southern WKL (Fig. 8c-d, g-h). However, a slight discrepancy of 1.84% (1.61%) exists between the maps by Cao et al. (2023) and Zou et al. (2017) and our simulations regarding the areal extent of seasonally frozen ground. These maps delineate certain areas as seasonally frozen ground, while our simulations classify them as permafrost (Fig. 8k-l). Furthermore, our simulations indicate that 0.61% (0.58%) of the central lowland area of WKL contains scattered seasonally frozen ground, whereas the maps by Cao et al. (2023) and Zou et al. (2017) categorize these areas as permafrost (Fig. 8k-l).





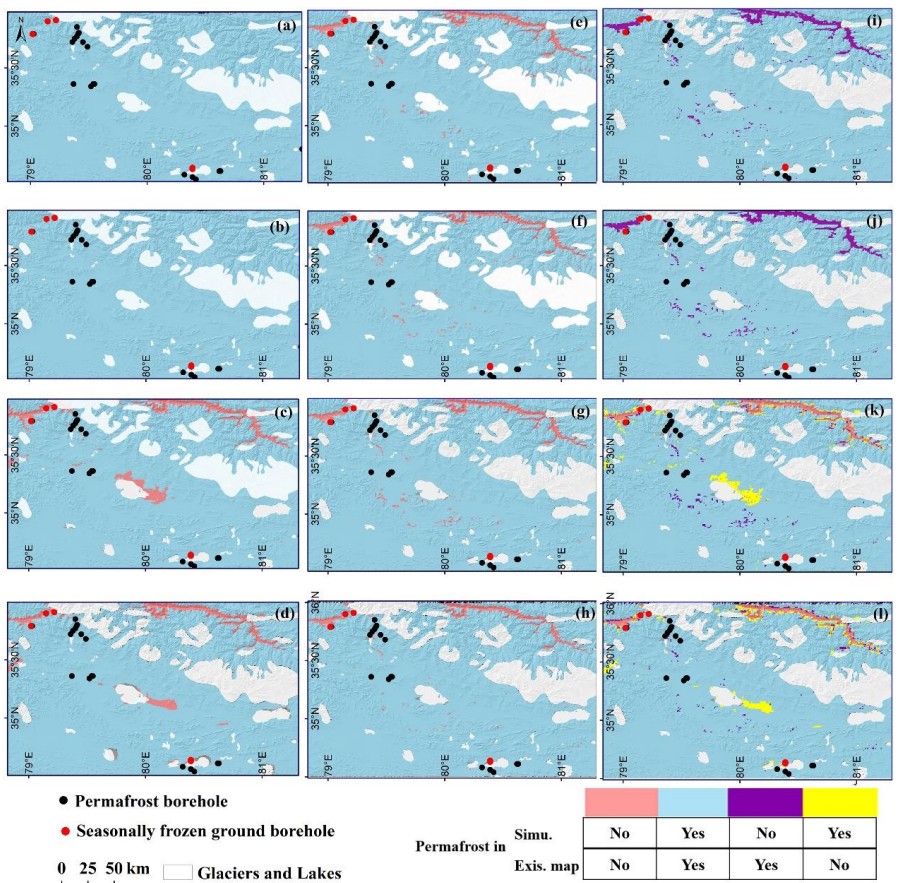

**Figure 8: Spatial distribution of frozen ground type in the WKL permafrost survey area as depicted by four different maps from various periods: (a) Li and Cheng (1996), (b) Wang et al. (2006), (c) Cao et al. (2023), and (d) Zou et al. (2017) (left panels). The corresponding modeled outputs are presented in the middle panels (e-h), while the spatial inconsistencies between the existing maps and our model outputs for each period are illustrated in the right panels (i-l).**

4.2.2 Initial thermal status of permafrost condition

To investigate how the thermal state of permafrost evolves with ongoing climate change, it is essential to first understand the characteristics of its initial conditions. Figure 9 presents the modeled initial status of MAGT15m, TTOP, and ALT for the year 1980. The results indicate drastic spatial variability in the ground thermal regime across the WKL permafrost survey area. The modeled





MAGT15m decreases markedly with increasing elevation. The highest average MAGT15m, approximately 0.5°C, is found in central low-elevation areas (below 4800 m a.s.l.), while the coldest

average MAGT15m, below -10°C, occurs in high-elevation regions (6000 m a.s.l.). Additionally, slight variations in the modeled MAGT15m are observed across different soil stratigraphic classes (see Fig. 9c). The lowest average MAGT15m, around -3.5°C, is found in the Aeolian sedimentary class, while the relatively warmer average temperature of -1°C is modeled in the Alluvial plain sedimentary class. A similar distribution pattern is evident for the modeled TTOP, although TTOP

values across the WKL permafrost survey region is slightly lower than MAGT15m values (see Fig. 10d-f). Like the ground temperature distribution pattern, the modeled ALT across the WKL also shows a strong dependence on elevation (Fig. 9j). In low-elevation areas (below 5400 m a.s.l.), modeled ALT ranges from 2.5 m to 3.0 m, with some localized regions exceeding 3.0 m. In contrast, ALT gradually decreases in high-elevation areas to less than 1.0 m, and it is modeled at 0 m in most

regions above 6000 m a.s.l., where ground temperatures remain below freezing year-round. Furthermore, remarkable variation in modeled ALT values is simulated across different ground stratigraphic classes, with the largest modeled average ALT occurring in the Alluvial sediments class and the lowest in the Glarosion sedimentary class (see Fig. 9i).

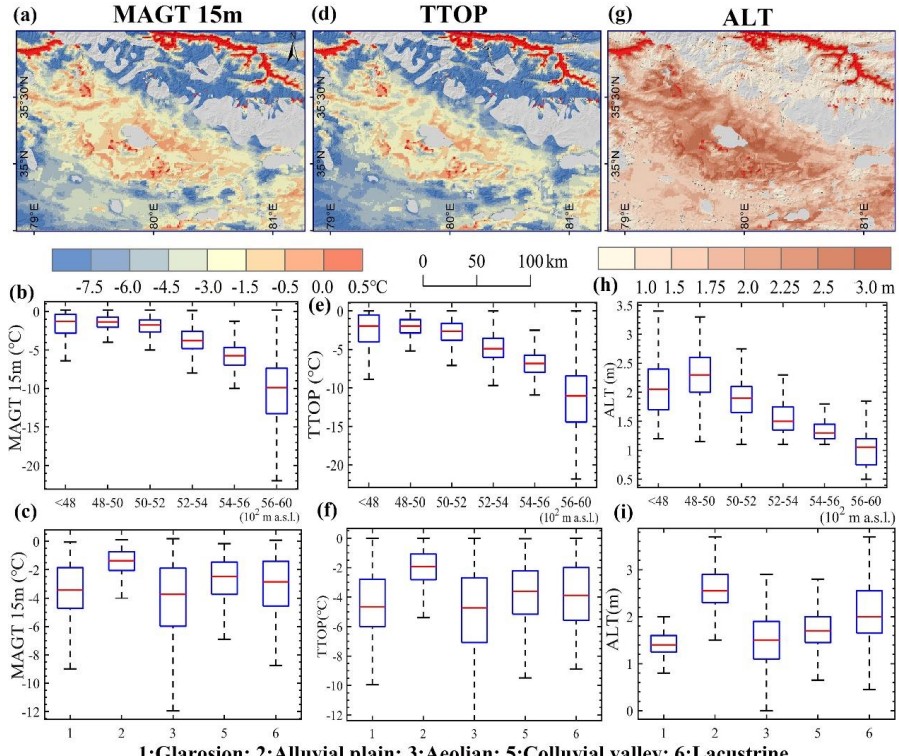

**1:Glarosion; 2:Alluvial plain; 3:Aeolian; 5:Colluvial valley; 6:Lacustrine**



**Figure 9: Spatial distribution of the simulated MAGT15m (first column, a-c), TTOP (second column, d-f), and ALT (third column, g-i) from the initial model simulation output for the year 1980. Seasonally frozen ground is depicted in red, while glaciers and lakes are shown in grey (first row). The middle row presents boxplot maps of MAGT15m, TTOP, and ALT, categorized by different elevations (ranging from 4300 m a.s.l. to 6000 m a.s.l., excluding certain areas). The third row categorizes the data by soil stratigraphic classes: Glarosion, Alluvial plain, Aeolian, Colluvial valley, and Lacustrine. In the boxplots, the top and bottom lines represent the 25th and 75th percentiles, respectively, while the whiskers extend to the highest and lowest values within 1.5 times the interquartile range. The middle line of each boxplot indicates the median.**

4.2.3 Evolution of permafrost thermal conditions

Figure 10 presents the simulated time series of spatial snapshot relative changes in interdecadal MAGT15m, TTOP, and ALT across the WKL permafrost survey region for 1980–2022. From the 1980s to the 1990s, the modeled MAGT15m across most of the WKL region (62.40%) exhibited relatively small variability, ranging from -0.3°C to 0.3°C (Fig. 10a). A warming trend prevailed from the 1990s to the 2000s, with the modeled MAGT15m increasing in 67.20% of the region, and the highest modeled values exceeding 1.8°C in certain local areas (Fig. 10b). From the 2000s to 2010–2022, the warming trend showed more pronounced fluctuations, with some areas even experiencing cooling compared to the previous decade. However, during this period, about 47.10% of the region experienced a cooling trend, with the largest decrease in MAGT15m reaching up to -1.8°C (Fig. 10c). Overall, during the simulation period from 1980 to 2022, the MAGT15m for most of the WKL permafrost survey region (58.58%) shows a warming trend, with increases ranging from 0.01°C to 1.8°C. Meanwhile, approximately 25.50% of the central region displays a noticeable cooling trend, with the most pronounced decreases modeled below -1.8°C (Fig. 10d). Similarly, across different decades, the most pronounced increase in TTOP occurred between the 1990s and 2000s, with 86.66% of the region experiencing a warming trend, and 16.35% of the area showing a warming magnitude exceeding 0.8°C. Following this, from the 2000s to 2010–2022, about 70.46% of the area experienced a warming trend, with increases ranging from 0.1°C to 1.8°C (Fig. 10e-g). According to the modeled TTOP outputs for the period from 1980 to 2022, approximately 81.68% of the area displayed an increasing trend, with about 17.20% of regions showing a warming magnitude above 1.3°C. However, in a small area (about 7.42%) located in the central part of WKL, the simulated TTOP showed a drastic decrease, ranging from -0.3°C to -1.3°C (Fig. 10h).

For ALT, the most pronounced increase was simulated across WKL from the 1980s to the 1990s,




with74.20% of the region experiencing an increase ranging from 0.1m to 1.5m, and in some areas, even exceeding 1.5 m during this period (Fig. 10i). Following this, from the 2000s to 2010–2022,

approximately 59.0% of the region experienced an increase in ALT, in which 6.14% of area shown a pronounced increase ranging from 0.3m to 1.5m. Meanwhile, approximately 0.97% of the area experienced a drastic decrease in ALT, with reductions exceeding -0.8m during the same period (Fig. 10k). However, compared to the previous decade, a slowdown in ALT increase was observed from the 1990s to the 2000s, with 58.0% of the region experiencing a modeled ALT increase (Fig. 10j).

Notably, during this period, approximately 7.61% of the region experienced a noticeable decline in ALT, with reductions ranging from -0.3m to -1.0m. Overall, from 1980 to 2022, the average ALT in WKL increased by 0.17m, with 83.10% of the region exhibiting increasing trend, and the highest increases exceeding 1.5m. Meanwhile, approximately 16.90% of the central region experienced a decline in ALT, ranging from -0.1m to -0.8m, with the most pronounced reductions locally

exceeding -0.8m (Fig. 10d).

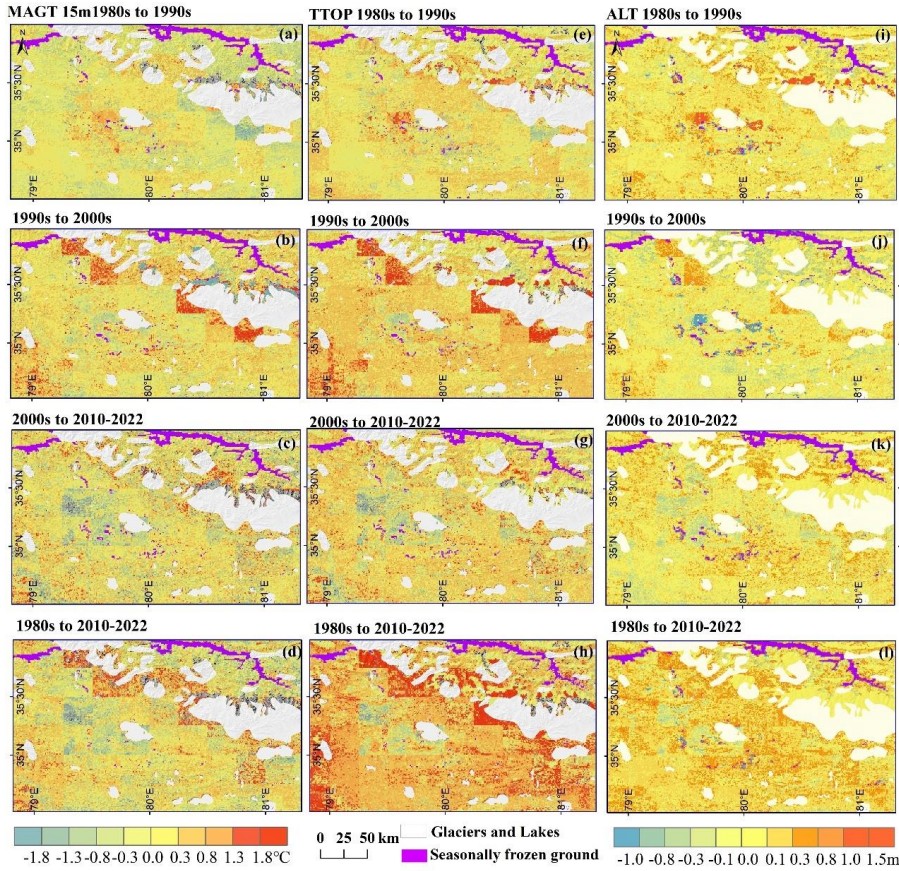





**Figure 10: Spatial relative changes in the modeled MAGT15m (left column: a-d), TTOP (middle column: e-h), and ALT (right column: i-l) for the decades of the 1980s, 1990s, 2000s, and the period from 2010 to 2022. Seasonally frozen ground is depicted in purple, while**

**glaciers and lakes are shown in grey.**

Figure 11 illustrates the interdecadal variation characteristics of MAGT15m, TTOP, and ALT across different elevations and soil stratigraphic classes. The results indicate that the modeled MAGT15m exhibited relatively small interdecadal variations and a slight overall increase from the 1980s to 2010-2022. During the simulation period, the most distinct changes in modeled MAGT15m

occurred in the highest elevation range (5600-6000 m a.s.l.), which showed an upward trend over time. However, the magnitude of these changes was less intense compared to those simulated in the simulated TTOP (Fig. 11a-b). Furthermore, the interdecadal changes in MAGT15m did not show apparent discrepancies across different soil stratigraphic classes (Fig. 11d). In contrast, a clear warming trend was observed in the modeled TTOP across various soil stratigraphic classes, with the

exception of the alluvial sedimentary class. As for ALT, the simulated values showed a remarkable increasing trend across various elevation ranges from the 1980s to the 2000s). (Fig. 11c). Furthermore, the modeled ALT values exhibited high interdecadal variability across different soil stratigraphic classes. In the alluvial and lacustrine sediment classes, the modeled ALT showed a substantial increase from the 1980s to 2010–2022, with an increase exceeding 0.17 m. In contrast,

the smallest change, an increase of 0.11 m, was simulated in the glarosion sediment class (see Fig. 1f).

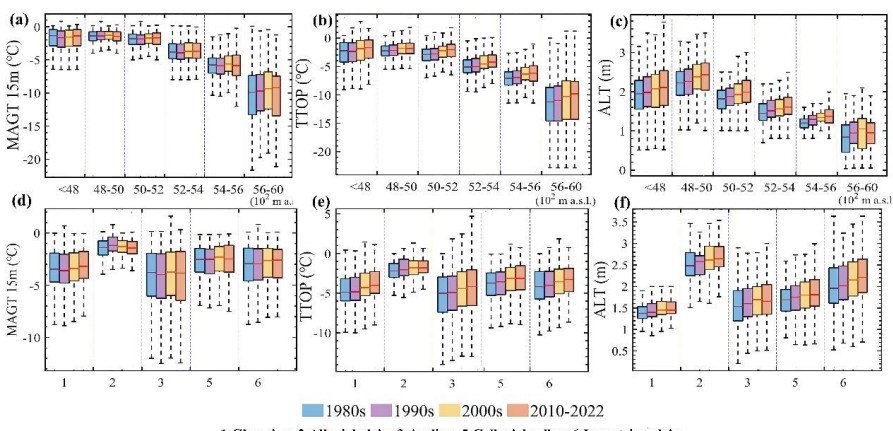

**Figure 11: Boxplot maps of modeled MAGT15m (first column: a,d), TTOP (middle column: b,e), and ALT (right column: c,f) for four period (1980s, 1990s, 2000s, and 2010-2022, depicted**

**in different colored boxes). The data is categorized by different elevations (ranging from 4300**





m a.s.l. to 6000 m a.s.l.) in the first row, and by soil stratigraphic classes—Glarosion, Alluvial
plain, Aeolian, Colluvial valley, and Lacustrine—in the bottom row. The top and bottom lines
of the boxplots represent the 75th and 25th percentiles, respectively. The whiskers extend to
the highest and lowest values within 1.5 times the interquartile range. The middle line of each
boxplot indicates the median.

4.2.4 Evolution of permafrost extent

Table 3 shows the permafrost aggradation/degradation in response to climate variability during
the 1980-2022 across the WKL permafrost survey area. According to the initial simulation outputs
from the 1980s, approximately 82.27% of the total area in WKL was underlain by permafrost. Of
this, 55.58% was found in areas classified as Aeolian ground stratigraphy, and 67.9% was located
at elevations ranging from 4800 to 5600 m a.s.l. Throughout the study period, the number of grid
cells with simulated permafrost remained constant between the 1980s and 1990s. However, from
the 1990s to the 2000s, a modest reduction of 0.15% in permafrost areas was modeled, followed by
a 0.44% increase between the 2000s and 2010–2022. Spatially, most of these changes occurred in
low-elevation regions below 4800 m a.s.l. and in areas classified as Alluvial plain sediments (Table
3). Overall, the simulation results suggest a relatively stable permafrost extent in WKL over the past
43 years.

**Table 3. Variations in the areal extent of frozen ground types across the WKL permafrost
survey region from 1980 to 2022. The data is categorized by elevation and soil stratigraphic**
**classes.**

| Altitude range $(10^2 m\,a.s.l)$ | Permafrost areal extent (%) | | | | | | | | SF. (%) |
|---|---|---|---|---|---|---|---|---|---|
| | <48 | 48-50 | 50-52 | 52-54 | 54-56 | 56-60 | >60 | Sum | |
| 1980s | **2.74** | 12.58 | 18.79 | 24.32 | 12.22 | 9.07 | 2.56 | 82.27 | 2.67 |
| 1990s | **2.87** | 12.48 | 18.76 | 24.32 | 12.22 | 9.07 | 2.56 | 82.27 | 2.67 |
| 2000s | **2.84** | 12.46 | 18.67 | 24.31 | 12.22 | 9.07 | 2.56 | 82.13 | 2.81 |
| 2010-2022 | **3.01** | 12.62 | 18.78 | 24.32 | 12.22 | 9.07 | 2.56 | 82.57 | 2.36 |
| Stratigraphic class | 1 | 2 | 3 | 5 | 6 | | | | |
| 1980s | 12.32 | **5.92** | 55.58 | 3.51 | 4.95 | | | 82.27 | 2.67 |
| 1990s | 12.36 | **5.75** | 55.71 | 3.51 | 4.94 | | | 82.27 | 2.67 |
| 2000s | 12.36 | **5.70** | 55.62 | 3.51 | 4.95 | | | 82.13 | 2.81 |





| 2010-2022 | 12.34 | **5.81** | 55.94 | 3.51 | 4.97 | | 82.57 | 2.36 |

Note: SF. indicates seasonally frozen ground. The numbers for soil stratigraphy correspond to the following sediment classes: 1: Glarosion; 2: Alluvial plain; 3: Aeolian, 5: Colluvial valley, and 6: Lacustrine. The glacier and lake area, accounting for 15.06%, was excluded from this statistic. Mainly changes are shown in bold.

**5 Discussion**

5.1 Applicability of the forcing data

Most previous evaluations indicated that soil temperature products derived from atmospheric circulation models or ESMs, which typically have coarse resolutions (~300 km), show larger uncertainties over the QTP, particularly permafrost region (Hu et al., 2019; Qing et al., 2020; Yang et al., 2020; Xi et al., 2023). At such resolutions, model forcing may not accurately represent permafrost thermal conditions due to the strong spatial variability of soil temperature and active layer thickness, driven by surface cover and soil moisture heterogeneity on the QTP (Hu et al., 2023). These inaccuracies in model forcing directly affect the accuracy of thaw depth simulations, often leading to an overestimation of permafrost degradation rates, which is inconsistent with current observed patterns (Lawrence et al., 2012; Zhao et al., 2024). In this study, we applied machine learning techniques to reconstruct remote-sensing-based land LST back to 1980 with a spatial resolution of 1 km and a monthly temporal resolution based on field observations, satellite data, and reanalysis products. Compared to in situ measurements, we found a slight cold bias in our reconstructed LST series, averaging approximately -0.80°C, particularly during summer months (see details in Sect. 3.1). This bias may be attributed to residual biases in the LST_Zou dataset, which exhibits a noticeable cold bias during these months. Nevertheless, the reconstructed thermal state of permafrost achieved an accuracy of ±0.25°C at 10 m depth, and active layer thickness (ALT) was reproduced within ±0.25 m—an improvement over previous studies that used coarser atmospheric forcing in similar domains (e.g., Chen et al., 2015; Wu et al., 2018). This confirms the appropriateness of the models to accurately simulate the permafrost thermal reges over long time scales.

Additionally, our analysis showed that the WKL permafrost survey area has experienced pronounced LST warming since the mid-1980s, with an accelerated warming trend in the last decade. This pattern aligns well with the documented warming trends on the QTP in recent studies (Jin et al., 2011; You et al., 2021; Yao et al., 2021; Li et al., 2024), indirectly validating the accuracy of our reconstructed LST data. Overall, we conclude that, compared to traditional atmospheric forcing, this new model forcing is a suitable choice for modeling ground thermal regime dynamics over the WKL.



It enables simulations of the permafrost thermal state with a much higher degree of precisions and spatial resolution than traditional model forcing, which is consistent with previous findings (Zhao et al., 2022). Similar findings suggest that enhancing the spatial resolution of the forcing dataset can

greatly improve the accurate representation of the effects of varying subsurface and surface properties on ground temperatures and thermal modeling accuracy (e.g., Zhang et al., 2013; Fiddes et al., 2015).

5.2 Spatiotemporal variations in thermal responses to climatic variability

In this study, we investigated the spatiotemporal dynamics of the permafrost thermal regime in

response to climate variability from 1980 to 2022 in a representative permafrost region on the northwest QTP (WKL). We found that the most drastic permafrost thermal warming (MAGT15m and TTOP) was modeled between the 1990s and 2000s, which does not correspond to the periods of the strongest warming in LST, observed during the 1980s and warming accelerated last from 2010. However, ALT variations were more consistent with LST fluctuations, and the most

pronounced increase in ALT was modeled between the 1980s and 1990s. Similar warming effects on ALT have been observed in the QTH (Li et al., 2012), where observed variations in ALT are primarily influenced by seasonal air temperatures and shallow ground temperatures, and thus exhibit inter-annual variation. Furthermore, our simulation indicated that the TTOP responded more rapidly and experienced more pronounced warming from 1980 to 2022. These phenomena can be explained

that changes in TTOP are characterized by short time lags in relation to ground surface temperature fluctuations, along with large amplitudes and high rates of temperature increase (Wu et al., 2010). Thus, the response of near-surface ground temperature to changing climate forcing is fast to immediate. In comparison, due to the exponential attenuation of heat transfer with depth, the amplitudes (seasonal variations) and rates of temperature increase diminished, resulting in longer

time lags at the depth of zero annual amplitude (Jin et al., 2008, 2011; Noetzli et al., 2009). Consequently, relatively stable thermal conditions were modeled for the MAGT15m during the same period.

Spatially, the characteristics of permafrost thermal warming vary widely across the study area and are strongly dependent on its initial thermal state. The magnitude of change was relatively low,

typically less than ±0.3°C, in transitional (−1.5°C < MAGT15m < −0.5°C) and unstable (−0.5°C < MAGT15m < 0.5°C) permafrost. In contrast, the most pronounced warming in MAGT15m was modeled in stable permafrost (e.g., MAGT15m<−1.5°C), with warming magnitudes reaching up to 1.8°C, and some localized areas experiencing even greater increases. This changing pattern generally exhibits an obviously altitude-dependent in spatial distribution characteristic. A similar




permafrost thermal warming pattern has been observed in the QTH and Arctic regions, including
Alaska and Russia (Smith, 2022; Zhao et al., 2020). This occurs because when cold permafrost is
warms, its temperature initially rises. However, as the permafrost temperature approaches the
thawing point (e.g., 0°C), ground ice begins to melt over a range of sub-zero temperatures. This
phase change absorbs a large amount of heat, resulting in lower apparent thermal diffusivity and

less energy directed toward increasing ground temperature (Langer et al., 2013, 2024). Consequently,
under similar external conditions, stable permafrost experiences relatively higher warming rates and
amplitudes.

However, what is more noteworthy is that, although the increase in permafrost temperature
within unstable and transitional zones was smaller, slower, and less sensitive to climate warming,

ground ice melt will lead to permafrost degradation. This degradation will convert permafrost into
seasonally frozen ground, subsequently inducing substantial geomorphic changes, such as ground
surface subsidence or hillslope failures, which result in challenges for engineering and natural
hazard prediction and mitigation. As shown in our simulation results, there was a change in
permafrost extent across the WKL, particularly in low-elevation regions below 4800 m a.s.l., where

the MAGT was greater than −0.5°C, especially in permafrost boundary areas near the thawing point
(0°C). Thus, regions with the least warming modeled may still be the most susceptible to permafrost
degradation in response to climate warming. Unlike the spatial characteristics of changes in the
thermal state of permafrost, which are almost entirely governed by elevation, the simulated
variations in ALT are influenced not only by elevation but also significantly by ground stratigraphy.

The highest average ALT of 2.5 m was modeled in the alluvial sediments class, showing a substantial
increase during the simulation period, whereas the lowest ALT of 1.5 m was modeled in the
glarosion sedimentary class, with a relatively minor increase over the same period. Similar patterns
of ALT changes have also been simulated in circumpolar Arctic permafrost regions (Langer et al.,
2013; Westermann et al., 2015).

5.3 Comparison with previous studies

It is well recognized that global-scale warming has impacted the thermal regime of permafrost
worldwide; however, it is still not fully understood how, at what rate, and to what extent permafrost
responds to climate fluctuations. This is partly because permafrost thermal dynamics are generally
poorly represented in global models, particularly in complex regions such as the QTP, leading to

huge uncertainties in predicting permafrost thermal state responses and climate feedbacks. In this
study, we used the MVPM to simulate thermal state in the WKL region in response to climate change





from 1980 to 2022. We quantified the spatial changes in permafrost and seasonally frozen ground across the WKL region during the simulation period (1980–2022). The model outputs are compared with published maps from four different periods (see details in Fig. 9). Overall, most of the simulated

permafrost distribution aligns with existing maps. However, notable differences occur in areas of seasonally frozen ground, similar to the permafrost distribution modeling conducted by Zhao et al. (2022) in the Xidatan region of the QTP. These discrepancies may result from variations in model forcing, models, study periods, spatial resolution, and other local factors (Zhao et al., 2022; Zou et al., 2017). The maps by Li and Cheng (1996) and Wang et al. (2006) synthesize available field data

for permafrost occurrent and properties, literature, aerial photographs, satellite images, and other sources. In these maps, permafrost boundaries were primarily determined using air temperature or mean annual ground temperature (MAGT) isotherms as thresholds and were manually delineated on topographic maps at scales of 1:3,000,000 and 1:4,000,000, respectively (Cao et al., 2016). The manual cartographic techniques used to delineate permafrost boundaries are prone to artifactual

errors, inevitably introducing uncertainties in the maps by Li and Cheng (1996) and Wang et al. (2006). Additionally, the coarse resolution of these maps does not adequately capture fine-scale variations in ground conditions, making it difficult to validate the results against field observations.

In contrast, the maps by Cao et al. (2023) and Zou et al. (2017), as well as our simulation outputs, are based on enhanced remote-sensing LST products with a spatial resolution of 1 km.

These maps show higher accuracy in identifying both permafrost and seasonally frozen ground compared to borehole observations. Moreover, the results reveal more spatial details. This finding underscores the potential advantages of remote sensing methods for studying regional-scale permafrost thermal regimes on the QTP, as noted by Zhao et al. (2022). The differences arise primarily because the maps by Cao et al. (2023) and Zou et al. (2017) assume that permafrost is in

equilibrium with long-term climate conditions—specifically, averaging the periods from 2005 to 2010 for Cao et al. (2023) and 2003 to 2012 for Zou et al. (2017). In reality, ground temperature observations and modeling studies have shown that the permafrost conditions are not in equilibrium with the atmospheric climate. Additionally, these maps do not account for the thermal state of deep permafrost, as the modeled soil column typically extends to depths less than 3 m. In the meantime,

as supra-permafrost subaerial taliks exist in some areas, the criterion of subzero TTOP used in Cao et al. (2023) and Zou et al. (2017) for determining permafrost occurrence may underestimate the extent of permafrost. Consequently, the areal extent of permafrost distribution estimated in these equilibrium-based maps is likely slightly underestimated compared to our simulation outputs (Zhao et al., 2022).



Compared to the equilibrium approach of Cao et al. (2023) and Zou et al. (2017), we employed a numerical model, that can simulate the transient changes in ground temperature and the spatial extent of permafrost distribution under climate change. Our modeling output suggests that the original thermal condition of permafrost over the WKL is relatively cold, with 80% of MAGT15m ranging from −7.5°C to −1.5°C. Of this, 34.13% corresponds to very stable permafrost (MAGT <

−5°C), 22.12% to stable permafrost (−5°C < MAGT < −3°C), and 23.08% to sub-stable permafrost (−3°C < MAGT < −1.5°C). In response to climate changes from 1980 to 2022, permafrost areal extent distribution in the WKL remained relatively stable, with less than 0.5% of permafrost experiencing aggradation or degradation due to climate fluctuations during this period. These results align well with borehole observations (Jin et al., 2011), which indicate that permafrost is more

thermally stable in continental climate regions influenced by stronger westerlies, particularly in the interior, as well as in the western and northern areas of the QTP, including the WKL. However, some studies report contrary findings, suggesting a faster response of permafrost to climate change (Guo and Wang et al., 2012, 2016; Ni et al., 2021; Shen et al., 2023). This discrepancy is largely due to model deficiencies that fail to capture the time lag between climate warming and permafrost thawing

in areas with thick permafrost. These limitations often stem from scarce observational data and an incomplete understanding of the complex physical processes in permafrost on the QTP (Sun et al., 2019; Hu et al., 2023). Most simulations of hydrothermal processes focus on shallow layers, and changes in heat and moisture within frozen soil are often ignored, hindering accurate predictions of heat and water exchanges in permafrost under climate change (Hu et al., 2023).

Moreover, permafrost on the QTP developed over a prolonged period of cold paleoclimate, resulting in a ground thermal state characterized by low temperatures and ground ice (Jin et al., 2011; Zhao et al., 2020). Current ground temperatures at various depths reflect cumulative historical climate changes, especially in deeper soil layers (i.e., tens to hundreds of meters) (e.g., Lachenbruch and Marshall, 1986; Allen et al., 1988; Osterkamp and Gosink, 1991; Harrison, 1991; Buteau et al.,

2004; Kneier et al., 2018; Langer et al., 2024). Therefore, the initial thermal state of permafrost, shaped by past climates, is essential for accurately simulating permafrost's thermal responses to climate warming. However, many numerical models often overlook the historical energy accumulated in permafrost and the effects of ground ice conditions below a depth of 1 m (Zhao et al., 2020; Hu et al., 2023). Furthermore, many models simplify the geothermal heat flux by setting

a zero-flux or constant-temperature condition at the bottom boundary (Wu et al., 2010; Xiao et al., 2013, Zhao et al., 2022). This potentially omits critical factors in long-term permafrost evolution and introduces large uncertainties in modeling both the present state and projected changes in permafrost temperature. Another limitation is that most current models do not account for the thaw




settlement process. As ground ice thaws, it causes settling and consolidation of ground material,
potentially leading to an underestimation of permafrost thawing, as the permafrost table may be
closer to the ground surface than predicted. This omission can also introduce errors in simulating
ALT and the burial depth of the permafrost table (Sun et al., 2023).

In comparison, our model fully accounts for the thermal-property differences between frozen
and thawed soil, phase changes of unfrozen water in frozen soil, ground ice distribution, thaw
settlement, and geothermal heat flow. The modeled ground temperature, ALT, thaw depth, and
permafrost distribution align well with observations, indicating that the MVPM accurately simulates
the heat transfer process in permafrost and effectively captures the attenuation and time lag of heat
transfer in deep permafrost. Notably, most numerical models overlook the thermal properties and
processes in deeper permafrost, while our findings emphasize that these factors play a critical role
in permafrost evolution and thaw trajectories in response to climate change.

## 5.4 Current model shortcoming and future improvements

*Model physics:* Current MVPM configuration does not account for a range of processes that
may influence the ground thermal regime in permafrost areas, such as non-conduction heat transfer
due to soil water convection and the exchange of lateral heat or water fluxes. Previous studies have
reported that lateral processes can significantly influence the ground thermal regime and contribute
to the lateral degradation of permafrost. This has been confirmed by field investigations conducted
at the margins of discontinuous permafrost zones and in areas around taliks, water bodies, and lakes
(Boike et al., 2015; Bense et al., 2012; Sjöberg et al., 2016; Kurylyk et al., 2016). The impact of
these local hydrological processes on permafrost thermal regimes remains unknown with the current
model configuration. Thus, our simulations do not accurately represent areas with significant lateral
heat fluxes, such as sharp mountain peaks, edges, or regions very close to water bodies. Despite
these limitations, our model reasonably reproduces ground temperature and ALT, aligning well with
observations in the WKL region. We believe that the dominant processes of permafrost heat transfer
in the WKL permafrost survey region are effectively captured by the one-dimensional heat
conduction approach. In addition, the MVPM uses satellite-derived skin temperatures (i.e., LST) to
define the upper boundary conditions, which do not account for the effects of vegetation canopies
on ground thermal conditions and may introduce uncertainties in areas with dense vegetation.
However, based on our field investigations, the vegetation in the WKL region is predominantly
sparse alpine desert, with most areas being barren (Li et al., 2012; Wang et al., 2016; Zhao et al.,
2019). This suggests that vegetation has a relatively limited effect on the estimated soil temperature.



Nevertheless, the identified shortcomings suggest the need for further validation studies, particularly focused on hydrogeological processes impact on permafrost thermal regime, which will be a key focus for future work. Additionally, further model development is necessary to better represent surface heterogeneities.

*Model initialization:* In this study, our model is initialized to an equilibrium condition using the forcing from the first year. This approach assumes that the initial permafrost regime was formed by land-atmosphere heat exchange under constant climate conditions for hundreds of years prior to 1980, which does not account for the transient nature of the ground temperature profile at that time. Although the impact of model initialization diminishes over time, grid cells where the initial forcing

is near the threshold for permafrost occurrence remain strongly affected. However, our model results provide a good reproduction of permafrost temperature, ALT, thaw depth, and four existing benchmark permafrost distribution maps across the WKL permafrost survey region. Based on this, we believe the uncertainties from model initialization should have a relatively limited effect on our long-term simulation results.

*Ground thermal properties:* An accurate description of soil properties is essential for modeling water and heat processes in frozen soils at both global and regional scales (Dai et al., 2019; Lawrence and Slater, 2008; Harp et al., 2016; Hu et al., 2023). However, most soil datasets used in models are derived from seasonally frozen areas, while data coverage is extremely limited in the permafrost regions of the QTP, with significant gaps for deeper soil layers (Hengl et al., 2017; Li et

al., 2015; Shangguan et al., 2013). Westermann et al. (2017) used geomorphological classification maps to parameterize large-scale patterns of ground thermal properties (e.g., sediment types, ground ice content, and surface properties) in the Siberian permafrost region. Similarly, in this study, we utilized an existing stratigraphic classification map, gridded to a 1 km² resolution, to account for the spatial-scale patterns of sediment types, ground ice, and surface properties in WKL, which were

used to parameterize subsurface properties. The MVPM was calibrated using ground temperature measurements from boreholes, specific to each soil class and geographical location. This approach effectively captures large-scale differences, particularly in ALT (see Sect. 3.3.2). However, notable small-scale variability in ground properties is superimposed on these large-scale patterns, leading to significant variability in ALT and ground temperature that cannot be resolved at the 1 km scale.

Furthermore, the stratigraphies assigned to each sediment class (Table 1) also may exhibit strong variability within each class, which can result in biased model outputs. Despite these limitations, we are confident that the key properties of the different sediment classes—critical for driving the dynamic response of ground temperatures to climate change—are accurately represented in our



model. To further enhance the accuracy of permafrost thermal dynamics modeling, we emphasize the importance of improving soil property datasets, particularly for permafrost regions.

## 6 Conclusions

The thermal state of permafrost is critically important for studying climate, ecology, hydrology, and engineering on the QTP. In this study, we quantitatively analyzed the spatiotemporal dynamics of the thermal regime across diverse environmental settings in a remote region (e.g., WKL) of the northwestern Tibetan Plateau from 1980-2020. This is based on an enhanced numerical model, MVPM. We employed clustering approaches and parallel computing techniques to enhance computational efficiency. The model forcing data, remote-sensing-based land surface temperature dating back to 1980, with a spatial resolution of 1 km×1 km and a temporal resolution of 1 month, was constructed by using machine learning techniques to integrate field observations, satellite data and reanalysis products. Soil properties were parameterized using a geomorphological classification map, supplemented by in situ measurements of ground temperature and ALT. The key conclusions drawn from this study are summarized below:

- Compared to traditional atmospheric forcing, new reconstructed model forcing enables simulations of the permafrost thermal state with a much higher degree of precisions and spatial resolution than traditional methods. The thermal state of permafrost was reproduced within a range of ±0.25°C at 10 m, and ALT was reproduced within ±0.25 m.

- The 80% permafrost thermal regime in WKL is relatively stable, with initial MAGT15m ranging from −7.5°C to −1.5°C. Spatially, the highest MAGT15m around −0.5°C, and the deepest ALT, ranging from 2.5m to 3.0m, were modeled in low-elevation areas below 4,800m a.s.l. In contrast, the lowest ground temperatures, dropping below −10°C, and the shallowest ALT, less than 1 m, were modeled in high-elevation regions above 5,600m a.s.l. Additionally, the alluvial plain sedimentary class exhibited the deepest ALT, with an average of 2.5m, while the glarosion sedimentary class had the shallowest ALT, averaging 1.5 m.

- From 1980 to 2022, the WKL permafrost survey area experienced a significant warming trend in LST, with an average increase of 0.40°C per decade. In response to this warming, 58.58% of the area exhibited a warming trend in MAGT15m. The most pronounced warming, averaging 0.3°C, was simulated in high-elevation areas above 5,600 m a.s.l., characterized by stable permafrost. Changes in ALT were closely linked to regional climate fluctuations and soil stratigraphic classifications. The greatest deepening of the ALT was modeled in alluvial and





lacustrine sediment classes, with an increase exceeding 0.17 m. In contrast, the smallest change, an increase of 0.11 m, was observed in the glarosion sediment class. Meanwhile, the spatial distribution of permafrost in the WKL remained relatively stable during this period, with less than 0.5% experiencing recover or degradation.

**Code and data availability**

In situ monitoring data from the field observation sites provided by the Cryosphere Research Station on Qinghai–Xizang Plateau of the Chinese Academy of Sciences (CAS) are available online at https://data.tpdc.ac.cn/en/ disallow/789e838e-16ac-4539-bb7e-906217305a1d/, December 12, 2024 (Zhao et al.,2021); Zhao et al. (2019b) (Permafrost and environment changes on the Qinghai-Tibetan Plateau. Beijing, China: Science Press); Li et al. (2012) (Permafrost distribution in typical

area of west Kunlun Mountains derived from a comprehensive survey (in Chines with English abstract), J. GLACIOL.). Enhanced MODIS LST data were provided by Zou et al. (2017) (https://doi.org/10.5194/tc-11-2527-2017). Daily Air temperature and precipitation from 1961-2019 were provided by Qin et al. (2022) (https://doi.pangaea.de/10.1594/PANGAEA.941329, December 12, 2024). CN05.1 datasets available on request from: wangjun@mail.iap.ac.cn

(https://ccrc.iap.ac.cn/resource/detail?id=228, December 12, 2024). Skin temperature dataset was download from ECMFW https://cds.climate.copernicus.eu/datasets/reanalysis-era5-land?tab=overview, December 12, 2024). Soil temperature dataset was download from NCEP Climate Forecast System Reanalysis (CFSR, https://rda.ucar.edu/datasets/ds093.0/dataaccess/ December 12, 2024). Fractional cloud cover and surface radiation budget dataset was download

from EUMETSAT, CM SAF (https://wui.cmsaf.eu/safira/action/viewDoiDetails?acronym=CLARA_AVHRR_V003, December 12, 2024). Leaf area index dataset is from Global Land Surface Satellite (GLASS) and MODIS (Global LAnd Surface Satellite (GLASS); https://modis.gsfc.nasa.gov/data/dataprod/mod15.php; December 12, 2024 ). The Shuttle Radar Topography Mission (SRTM) with a 1 arcsec (~30 m)

DEM data were from Hole-filled seamless SRTM data V4, International Center for Tropical Agriculture (CIAT), available at http: //srtm.csi.cgiar.org (Jarvis et al., 2008). The background maps of China are provided by Wen et al. (2024, https://doi.org/10.1007/s10584-024-03712-7). The Tibet Plateau boundary (Zhang, 2019a) and the geological sediment classification map (Zhou et al., 2007), along with the lake dataset (Zhang et al., 2019b) is freely available from the National Tibetan Plateau

Data Center (http://data.tpdc.ac.cn/zh-hans/, December 12, 2024). The glacier inventory comes from the Second Glacier Inventory Dataset of China (Guo et al., 2015, doi: 10.3189/2015JoG14J209). Four existing permafrost distribution maps: Li et al. (1996, Gansu Culture Press, Lanzhou) and



Wang et al. (2006) (Chinese Map Press, Beijing, China), and Zou et al. (2017) (https://doi.org/10.5194/tc-11-2527-2017) can free download from National Tibetan Plateau Data Center (http://data.tpdc.ac.cn/zh-hans/, December 12, 2024); Cao et al. 2023 (https://doi.org/10.5194/essd-15-3905-2023, December 12, 2024).

The new permafrost model source code is available on request from the following co-authors of this study: Jianting Zhao (first author), jt.zhao@nuist.edu.cn; Lin Zhao (corresponding author), lzhao@nuist.edu.cn; and Zhe Sun, sunzhe@lzb.ac.cn.

**Author contributions**

LZ conceived and conceptualized the idea; JZ and ZS developed the methodology; LZ, ZS, GH, and WZ supervised the study; JZ performed data processing and analyses. LZ, ZS and GH acquired the funding and provided the resources; DZ, GL, QP, ED, ZL, XW, and YX participated in the fieldwork and maintained the observation sites; JZ wrote the manuscript, and LZ, ZS, GH, MX, LW, and WZ reviewed and edited the writing.

**Declaring of competing interest**

The contact author has declared that none of the authors has any competing interests.

**Acknowledgments**

Warm thanks to all the scientists, engineers, and students who participated in the field investigations and measurements, and helped maintain the observation network for data acquisition.

**Financial support**

Financial support for this research was provided by the National Natural Science Foundation of China (grant no.41931180, 42322608, and 42401149); the Second Tibetan Plateau Scientific Expedition and Research (STEP) Program, China (grant no. 2019QZKK0201); China Postdoctoral Science Foundation funded project (grant no.2022M721670); Guangxi Natural Science Foundation (grant no. 2024GXNSFBA010295) and China Scholarship Council (grant no. 202309040044).

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
