# Peer review of "The thermal state of permafrost under climate change on the Qinghai-Tibet Plateau (1980-2022): A case study of the West Kunlun"

_EGUsphere, 2024_

## Referee Comment (RC1)

The Tibetan Plateau hosts the world's most extensive high-altitude permafrost areas, covering approximately 40% of the region. Over the past few decades, widespread permafrost degradation on the Plateau has been primarily driven by climate warming. However, the harsh environmental conditions and logistical challenges have severely limited the establishment of observational sites.

To better understand the evolution of the permafrost thermal regime over recent decades, Zhao et al. present a modeling approach—the Moving-Grid Permafrost Model. This model is forced by remote sensing data that have been corrected using a machine learning technique, thereby enabling the reconstruction of the historical permafrost thermal regime in West Kunlun. Furthermore, to enhance the spatial resolution of the simulations, the study employs clustering techniques and parallel computing methods to accelerate model runs. Consequently, the model demonstrated improved performance relative to available observational data and provided valuable insights into the spatiotemporal evolution of permafrost thermal regimes in West Kunlun.

Overall, this work merits attention from The Cryosphere, provided that the authors adequately address the review comments and incorporate additional information.

General Comments

1. In terms of the transient numerical model, First, I would like to understand the distinct advantages of the Moving-Grid Permafrost Model (MVPM) compared to existing models, like GIPL, Noah-MP, CLM, and CryoGrid. The authors state that the MVPM accounts for the thermal properties between frozen and thawed soil, unfrozen water content in frozen soil, ground ice distribution, thaw settlement of the ground surface, and geothermal heat flux to address model deficiencies. However, these physical processes and parameterization schemes are also implemented in other land surface models. Could the authors clarify what specific improvements or innovations MVPM provides over these existing models? Second, how does the MVPM model deal with the water balance in the soil domain? Which scheme does it use for the dynamics of soil water contents? No flow (constant water plus ice contents)? Bucket scheme? Richards equation? Third, although the snow cover on the Tibetan Plateau is relatively thin, it can significantly affect the hydrothermal state of the permafrost beneath it. Does the MVPM model consider the insulation and cooling effect of snow cover? Fourth, does this study activate the ground subsidence module of MVPM? This means that does this study consider the existence of excess ice?

2. Aiming at model forcing, as the only model forcing variable, this study adopted three statistical and machine-learning approaches to extend the land surface temperature from Zou et al (2014, 2017). I was wondering why the authors selected these eight specific input variables—surface air temperature, precipitation, skin temperature, soil temperature, fractional cloud cover, surface radiation budget, leaf area index, and digital elevation model—for the statistical and machine learning approaches. Could the authors clarify whether including more (or fewer) variables might help to avoid issues of model underfitting or overfitting? Besides, Furthermore, what is the basis for selecting the particular datasets used for these variables? For example, the study utilizes the uppermost soil temperature from CFSR, while skin temperature is taken from ERA5Land. Given that ERA5Land also provides uppermost soil temperature data at a higher spatiotemporal resolution compared to CFSR.

Specific Comments

1. Line 40: Simth et al 2022? maybe it is a wrong reference?

2. Line 49: The reference (Zhao et al., 2019) seemed missing.

3. Line 51: Should the Qinghai-Tibet Highway and Railway be abbreviated as QTH? Not sure.

4. Line 84: "ALT" should be given its full name, this is the first time it has been abbreviated. And "refreezing" of what?

5. Line 161: Does this study activate the settlement module?

6. Line 183: What is the surface radiation budget? net radiation? net shortwave radiation? or net longwave radiation? It is not clear.

7. Line 189: The resolution of all input data is not daily.

8. Line 271: How to deal with initial water/ice content?

9. Line 274: "approximately"? It should be an exact number for the grid cell to be simulated.

10. Line 300: "Filed investigation and borehole monitoring datasets"? I guess it is "Field".

11. Line 430: "Grey shading"? only saw the grey line.

12. Section 4.2.4: The author states, " Permafrost area in the West Kunlun kept stable from 1980 to 1999, decreased in the 2000s, while increased between 2010 and 2022." However, the MVPM model is just forced by land surface temperature, which showed an increasing trend between 1980 and 2022 (Figure 4). Could the author explain why the permafrost area increased between 2010 and 2022?

13. Line 601–609: So how about the reanalysis data (like Chinese meteorological forcing datasets, ERA5 Land)? Compared with the forcing data from ESMs, the spatiotemporal resolution of them is better.

14. Could the author explain how this study can be extended to the future projections? Due to its inability to obtain the land surface temperature with higher resolution from remote sensing data in the future, how to diagnose the future condition of permafrost in West Kunlun.

---

## Author Comment (AC1)

**Response to Referee#1 egusphere-2024-3956**

The Tibetan Plateau hosts the world's most extensive high-altitude permafrost areas, covering approximately 40% of the region. Over the past few decades, widespread permafrost degradation on the Plateau has been primarily driven by climate warming. However, the harsh environmental conditions and logistical challenges have severely limited the establishment of observational sites.

To better understand the evolution of the permafrost thermal regime over recent decades, Zhao et al. present a modeling approach—the Moving-Grid Permafrost Model. This model is forced by remote sensing data that have been corrected using a machine learning technique, thereby enabling the reconstruction of the historical permafrost thermal regime in West Kunlun. Furthermore, to enhance the spatial resolution of the simulations, the study employs clustering techniques and parallel computing methods to accelerate model runs. Consequently, the model demonstrated improved performance relative to available observational data and provided valuable insights into the spatiotemporal evolution of permafrost thermal regimes in West Kunlun.

Overall, this work merits attention from The Cryosphere, provide that the author adequately addresses the review comments and incorporate additional information.

**Response:**

Thanks a lot for your comments on our paper, which have helped us improve the quality of the manuscript. Below is our detailed response to these comments, describing how we have addressed them in our revised manuscript according to the Referee's comments. The original reviewer comments are in black font, while our responses appear in blue font. The corresponding edits in the manuscript are highlighted in red font.

**General Comments:**

1. In terms of the transient numerical model, First, I would like to understand the distinct advantages of the Moving-Grid Permafrost Model (MVPM) compared to existing models, like GIPL, Noah-MP, CLM, and CryoGrid. The authors state that the MVPM accounts for the thermal properties between frozen and thawed soil, unfrozen water content in frozen soil, ground ice distribution, thaw settlement of the ground surface, and geothermal heat flux to address model deficiencies. However, these physical processes and parameterization schemes are also implemented in other land surface models. Could the authors clarify what specific improvements or innovations MVPM provides over these existing models? Second, how dose the MVPM model deal with the water balance in the soil domain? Which scheme does it use for dynamics of soil water contents? No flow (constant water plus ice contents)? Bucket scheme?

Richards equation? Third, although the snow cover on the Tibetan Plateau is relatively thin, it can significantly affect the hydrothermal state of the permafrost beneath it. dose the MVPM model consider the insulation and cooling effect of snow cover? Fourth, dose this study activate the ground subsidence module of MVPM? This means that does this study consider the existence of excess ice?

**Response:**

Currently, permafrost degradation on the Qinghai-Tibetan Plateau (QTP) has been recognized, but scholars struggle to reach consensus on quantitative assessments of permafrost changes in response to climate change. Drastic discrepancies exist in the timing, rate, and magnitude of permafrost degradation among modeling results, driven by differences in datasets, model structures and parameterization scheme (Hu et al., 2023). In general, Land surface models (LSMs) have advanced obviously over the past decades, evolving from simple schemes that produce lower boundary conditions for the atmosphere to complex models incorporating several key physical processes crucial to understanding permafrost dynamics, yet, a number of challenges persist (Matthes et al., 2025). Here, the challenges arise from:

Firstly, there are issues related to the structure of model soil columns. A key example is the subsurface grid structure and maximum depth, which remains generally fixed in LSMs coupled with ESMs due to computational constraints (Matthes et al., 2025). LSMs have primarily focused on optimizing parameterization schemes for near-surface hydrothermal processes, typically constrained to the active layer within the upper 2–3 m (Sun et al., 2019). While some models have attempted to extend the soil column simulation depth (e.g., the new version of CLM), they have inadequately accounted for the effects of ground ice and the thermal state of deep permafrost. Refining the stratification of deep soil columns and fully considering the vertical heterogeneity of ground ice distribution in deep permafrost is essential. Without these considerations, simulations may fail to accurately capture changes in the thermal state of areas with thicker, colder permafrost, whereas simulations extending to greater depths more effectively represent the thermal response of permafrost to warming (Alexeev et al., 2007; Sun et al., 2019).

Secondly, concerning of the model's lower boundary conditions, i.e. a geothermal steady heat flow at depths greater than 40 m is a fundamental prerequisite for accurately capturing long-term permafrost thermal dynamics (Mendoza et al., 2020; Sun et al., 2021). However, models such as Noah-MP and CLM often ignore geothermal heat flux by setting zero flux or constant temperature as the bottom boundary condition (Wu et al., 2010; Xiao et al., 2013, Guo et al., 2012). Consequently, this simulation can't accurately simulate permafrost thermal dynamics on the QTP.

Third, issues related to the treatment of soil freezing and thawing arise from model limitations. LSMs, such as CLM, typically assume that phase transition occurs at a specific temperature point (generally 0°C). However, observations from heat-water dynamics monitoring sites in the QTP confirm that the ice-water phase transition in the active layer occurs over a temperature interval, rather than as an abrupt change at a specific temperature point. This suggests that the apparent heat capacity method, which assumes phase change occurs over a temperature interval, more accurately represents the actual freezing and thawing processes (Sun et al., 2021).

Fourth, in addition to process representation aforementioned, horizontal resolution is a critical feature for capturing the high landscape heterogeneity of mountainous permafrost areas. Currently, LSMs in ESMs remain relatively coarse in horizontal resolution, e.g., with the approximately 140 km in the Coupled Model Intercomparison Project Phase 6 (CMIP6) (Chen et al., 2021).

By comparison, permafrost models typically involve modeling a soil column depth greater than 20 m to describe the annual ground temperature cycle and evaluate long-term permafrost evolution, with a primary focus on heat transfer processes in permafrost (Buteau et al., 2004; Riseborough et al., 2009; Burn et al., 2009). Numerous studies have demonstrated that permafrost models can effectively capture the sluggish nature and attenuation of heat transfer in deep permafrost, with simulation results closely matching long-term observational data (Wu et al., 2010; Etzelmüller et al., 2011; Hipp et al., 2012; Zhao et al, 2022). Moreover, unlike LSMs in ESMs, permafrost models can be driven by near-surface meteorological variables (e.g., air temperature, land surface temperature) from diverse sources (Westermann et al., 2017). These forcing data are static and not influenced by the state of the land surface. Furthermore, such models face fewer runtime constraints, enabling high-resolution spatial analysis at scales ranging from meters to hundreds of meters with high computational efficiency, focusing on specific regions of interest and allowing simulations across centennial to millennial timescales (Matthes et al., 2025).

One-dimensional numerical models of ground heat conduction incorporating phase change effects serve as crucial and reliable tools for understanding complex permafrost dynamics (Riseborough et al., 2009). The Move-Grid Permafrost Model (MVPM) is a numerical framework used to infer time series of ground temperature with the land surface as the model's upper boundary (Sun et al., 2019). Its model physics is similar to other widely employed models, such as GIPL2.0 (Dmitry et al., 2017) and CryoGrid2.0 (Westermann et al., 2013): the change of internal energy and temperature in the ground is entirely determined by Fourier's law of heat conduction, and the latent heat generated or consumed by soil freezing and thawing. Movement of water or water vapor in the

ground is not included, so the soil water content can only change over time due to freezing processes.

The differences lie in how soil thermophysical properties (e.g., soil heat capacity and soil thermal conductivity) are estimated. The GIPL and CryoGrid models used parameterization schemes based on soil texture derived from de Vries (1963) or modified after this scheme. It is evident that an accurate description of soil property data is essential for modeling. However, most of the soil data are from within the seasonally frozen areas, while data coverage is extremely poor in permafrost regions on the QTP (Li et al., 2015; Shangguan et al., 2013). Moreover, a comprehensive comparison by He et al. (2021) demonstrated that all of the analyzed 39 approaches to calculate soil thermal conductivity in frozen soils performed inadequately. No parameterization scheme is suitable for the permafrost on the QTP, which suggests that new parameterizations are needed (Chen et al., 2019).

In MVPM, the soil heat capacity and soil thermal conductivity under frozen and thawed stages were estimated by a piecewise function:

$$
C_{\text{eff}}(T) = \begin{cases} c_f & , \ T \leq T_1 \\ \dfrac{L(\theta - \theta_u)}{T_2 - T_1} + \dfrac{c_f + c_u}{2} & , \ T_1 < T < T_2 \\ c_u & , \ T \geq T_2 \end{cases}
$$

$$
C_{\text{eff}}(T) = \begin{cases} c_f & , \ T \leq T_1 \\ \dfrac{L(\theta - \theta_u)}{T_2 - T_1} + \dfrac{c_f + c_u}{2} & , \ T_1 < T < T_2 \\ c_u & , \ T \geq T_2 \end{cases}
$$

Where the subscripts $f$ and $u$ represent the frozen and thawed states of soil, respectively. $\theta$ and $\theta_u$ represent the total volumetric water content and volumetric unfrozen water content in frozen soil, respectively. L = 334.54 MJ/m³ is the volumetric latent heat from the ice-water phase transition.

We utilized our established permafrost monitoring network and large-scale field investigations in the WKL survey region. Soil samples were collected from 15 borehole cores (depths 15−59 m) across various geomorphic classes. Thermophysical properties (e.g., stratigraphies, soil texture, dry bulk density, and ground ice content) of soil layers were assessed through laboratory measurements.

The apparent heat capacity ($C_{\text{eff}}$ (z,T)) (J/(m³·°C)) was to handle the phase change latent heat of unfrozen water within a specified temperature range of −0.3 to 0 °C based on observations. The slope of the unfrozen water-temperature curve $k = \dfrac{(\theta - \theta_u)}{T_2 - T_1}$ can be determined by the hydrothermal

measurement from borehole core. As for thermophysical parameters (e.g., thermal conductivity and heat capacity), based on soil texture, we pre-selected plausible values from Principles of Geocryology (Yershov, 2016), and then refined these during model calibration by manual stepwise optimization procedures recommended by Hipp et al. (2012). The well-calibrated ground thermal properties upscaled using WKL geomorphological classification map. Our simulation results were carefully validated against soil temperature monitoring, ALT, and permafrost maps from different periods between 1996 and 2016 across the WKL permafrost survey region. Similar simulations were conducted before at several borehole sites and limited regions along the QTP (Zhao J. et al., 2022; Sun et al., 2022, 2023). These studies consistently demonstrate that the MVPM has high performance in modeling long-term permafrost thermal dynamics on the QTP.

Second, how does the MVPM model deal with the water balance in the soil domain? Which scheme does it use for dynamics of soil water contents? No flow (constant water plus ice contents)? Bucket scheme? Richards equation?

We set no-flow conditions in the model soil columns, and the soil water content can only change over time due to soil freezing processes. Related description, please refer to the answer to the first sub-question above.

Third, although the snow cover on the Tibetan Plateau is relatively thin, it can significantly affect the hydrothermal state of the permafrost beneath it. dose the MVPM model consider the insulation and cooling effect of snow cover?

In our current simulation time scale, the MVPM model does not explicitly consider the effect of snow cover. While we acknowledge that snow cover can influence the hydrothermal state of permafrost, we believe this omission is justified for several reasons:

Firstly, the near-surface ground temperature is greatly affected by seasonal variations of air temperature and surface conditions (e.g., snow cover, vegetation conditions), characterized by frequent fluctuations and complex patterns of variation (Lunardini et al., 1995). However, the ground acts as a natural low-pass filter for short-term meteorological signals, with the amplitude of the seasonal cycle progressively decreasing with depth. At the depth of Zero Annual Amplitude (ZAA), temperature variations become undetectable (Jin et al., 2011; Dobiński et al., 2022). The variation trend of mean annual ground temperature at ZAA generally aligns with the long-term air temperature trend (Smith and Riseborough, 1983; Buteau et al., 2004; Jin et al., 2011).

Secondly, snow cover distribution is spatially variable over the QTP, with persistently snowcovered areas primarily occurring in the southeastern QTP and in alpine regions with elevations higher than 6000m (Qin et al., 2006; Pu et al., 2007; Yan et al., 2022). In the vast permafrost zone of the QTP, due to strong solar radiation and wind, snow cover is rare, thin (approximately 3cm) with short duration that mostly lasts less than 1 day for a single snow event (Che et al., 2008; Zou et al., 2017).

Thirdly, the subsurface thermal model MVPM, using land surface temperature as model forcing at the upper model boundary, applied a modified clear-sky LST product from MODIS developed by Zou et al. (2017). This product partially accounts for the influence of surface processes, including snow cover effects on LST by incorporating automatic weather station (AWS) observations from a typical permafrost region in the central QTP. These observations, which reflect the actual climate conditions at satellite overpass times, were included in the model training datasets. Moreover, our model simulations reasonably reproduce the vertical ground temperature profile, and the simulated active layer thickness is in good agreement with observations. While thin snow cover might have a cooling effect on ground surface temperature due to the high albedo of fresh snow and rapid snowmelt processes (Zhang et al., 2005), we believe this cooling effect is of short duration and has minimal impact at our simulation time scales. On the contrary, snow cover effects would be more significant for centennial to millennial time scale simulations rather than the decadal scale used in our study.

Fourth, dose this study activate the ground subsidence module of MVPM? This means that does this study consider the existence of excess ice?

No, in this study, we did not activate the ground subsidence module of the MVPM model. Our focus in this work was primarily on the permafrost thermal regime evolution under climate change. Additionally, turn off the ground subsidence module helped improve modeling computational cost. Therefore, the existence of excess ice and its potential effects on ground subsidence were not considered in the present study.

**Reference:**

Alexeev, V. A., Nicolsky, D. J., Romanovsky, V. E., Lawrence, D. M.: An Evaluation of Deep Soil Configurations in the CLM3 for Improved Representation of Permafrost: How Deep Should the CLM3 Soil Layer Be? Geophysical Research Letters 34, no. 9, L09502, https:// doi.org/ 10. 1029/ 2007G L029536, 2007.

Burn, C.R., Zhang, Y.: Permafrost and climate change at Herschel Island (Qikiqtaruq), Yukon Territory, Canada. J. Geophys. Res. Earth Surf. 114(F2), https://doi.org/10.1029/2008JF001087, 2009.

Buteau, S., Fortier, R., Delisle, G., Allard, M.: Numerical simulation of the impacts of climate warming on a permafrost mound. Permafr Periglac Process., 15(1):41-57. https://doi.org/10.1002/ppp.474, 2004.

Chen, H., Nan, Z., Zhao, L., Ding, Y., Chen, J., Pang, Q.: Noah Modelling of the Permafrost Distribution and Characteristics in the West Kunlun Area, Qinghai-Tibet Plateau, China. Permafr. Periglac. Process. 26, 160–174, https://doi.org/10.1002/ppp.1841, 2015.

Chen, D., Rojas, M., Samset, B.H.: Framing, Context, and Methods, in Climate Change, The Physical Science Basis (Cambridge, UK: Cambridge University Press): 147–286, 2021.

Che, T., Xin, L., Jin, R., Armstrong, R., and Zhang, T.: Snow depth derived from passive microwave remote-sensing data in China, Ann. Glaciol., 49, 145–154, https://doi.org/10.3189/172756408787814690 , 2008.

D. de Vries, Thermal Properties of Soils, in Physics of the Plant Environment, eds. W. R. Wijk and A. J. W. Borghorst (Amsterdam, The Netherlands: North-Holland): 210–235, 1963.

Dobiński, W., and Marek K.: Permafrost Base Degradation: Characteristics and Unknown Thread with Specific Example from Hornsund, Svalbard, Front. Earth Sci. 10:802157, doi: 10.3389/feart.2022.802157, 2022.

Etzelmüller, B., Schuler, T.V., Isaksen K, Christiansen, H., Farbrot, H, Benestad, R.: Modeling the temperature evolution of Svalbard permafrost during the 20th and 21st century. Cryosphere.5(1):67-79. https://doi.org/10.5194/tc-5-67-2011, 2011.

Guo, D., Wang, H., and Li, D.: A projection of permafrost degradation on the Tibetan Plateau during the 21st century, 117, D05106, J. Geophys. Res. Atmos., 117, D05106, https://doi.org/10.1029/2011JD016545, 2012.

He, H., N, G., Flerchinger, Y., Kojima., Dyck, M.: A Review and Evaluation of 39 Thermal Conductivity Models for Frozen Soils, Geoderma, 382: 114694, https://doi.org/10.1016/j.geoderma.2020.114694, 2021.

Hengl, T., de Jesus, J.M., Heuvelink, G.B., Gonzalez, M.R., Kilibarda, M., Blagoti´c, A., Shangguan, W., Wright, M.N., Geng, X., Bauer-Marschallinger, B.: SoilGrids250m: Global gridded soil information based on machine learning. PLoS. One 12 (2), e0169748, https://doi.org/10.1371/journal.pone.0169748, 2017.

Hipp, T., Etzelmüller, B., Farbrot, H., Schuler, TV.: Modelling borehole temperatures in southern Norway–insights into permafrost dynamics during the 20th and 21st century. Cryosphere, 6(3):553-571. https://doi.org/10.5194/tc-6-553-2012, 2012.

Hu, G., Zhao, L., Li, R., Park, H., Wu, X., Su, Y., Guggenberger, G., Wu, T., Zou, D., Zhu, X., Zhang, W., Wu, Y., and Hao, J: Water and heat coupling processes and its simulation in frozen soils: Current status and future research directions, Catena, 222, 106844, 985 https://doi.org/10.1016/j.catena.2022.106844, 2023.

Jin, H., Luo, D., Wang, S., Lü, L., and Wu, J.: Spatiotemporal variability of permafrost degradation on the Qinghai-Tibet Plateau, Sci. Cold Arid Reg., 3, 281–305, DOI: 10.3724/SP.J.1226.2011.00281, 2011.

Li, Q., Sun, S., 2015. The Simulation of Soil Water Flow and Phase Change in Vertically Inhomogeneous Soil in Land Surface Models. Chin. J. Atmos. Sci. 39 (4), 827–838.

Lunardini, V.: Permafrost Formation Time. CRREL Report 95-8, US Army Corps of Engineers, Cold Regions Research and Engineering Laboratory, 1995.

Hermoso de Mendoza, I., Beltrami, H., MacDougall, A. H., and Mareschal, J.-C.: Lower boundary conditions in land surface models – effects on the permafrost and the carbon pools: a case study with CLM4.5, Geosci. Model Dev., 13, 1663–1683, https://doi.org/10.5194/gmd-13-1663-2020, 2020.

Nicolsky, D. J., Romanovsky, V. E., Panda, S. K., Marchenko, S. S., Muskett, R. R.: Applicability of the ecosystem type approach to model permafrost dynamics across the Alaska North Slope, Journal of Geophysical Research: Earth Surface, 122(1), 50-75, https://doi.org/10.1002/2016JF003852, 2017.

Pu, Z., Xu, L., and Salomonson, V. V.: MODIS/Terra observed seasonal variations of snow cover over the Tibetan Plateau, Geophys. Res. Lett., 34, L06706, https://doi.org/10.1029/2007GL029262, 2007.

Qin, D., Liu, S., and Li, P.: Snow cover distribution, variability, and response to climate change in western China, J. Clim., 19, 1820–1833, https://doi.org/10.1175/JCLI3694.1, 2006.

Riseborough, D., Shiklomanov, N., Etzelmüller, B., Gruber, S., Marchenko, S.: Recent advances in permafrost modelling. Permafr. Periglac. Process.:19(2):137-156. https://doi.org/10.1002/ppp.615, 2009.

Smith, M., Riseborough, D.: Permafrost sensitivity to climatic change. In Proceedings, 4th International Conference on Permafrost, Vol. 1. Fairbanks, Alaska, National Academy Press: Washington, DC; 1178–1183, 1983.

Shangguan, W., Dai, Y., Liu, B., Zhu, A., Duan, Q., Wu, L., Ji, D., Ye, A., Yuan, H., Zhang, Q., Chen, D., Chen, M., Chu, J., Dou, Y., Guo, J., Li, H., Li, J., Liang, L., Liang, X., Liu, H., Liu, S., Miao, C., Zhang, Y.: A China data set of soil properties for land surface modeling. J. Adv. Model. Earth Syst. 5 (2), 212–224, https://doi.org/10.1002/jame.20026, 2013.

Sun, Z., Zhao, L., Hu, G., Qiao, Y., Du, E., Zou, D., Xie, C.: Modeling permafrost changes on the Qinghai-Tibetan plateau from 1966 to 2100: A case study from two boreholes along the Qinghai-Tibet engineering corridor. Permafr. Periglac. Process. 31 (1), 156–171, https://doi.org/10.1002/ppp.2022, 2020.

Sun, Z., Zhao, L., Hu, G., Zhou, H., Liu, S., Qiao, Y., Du, E., Zou, D., and Xie, C.: Effects of Ground Subsidence on Permafrost Simulation Related to Climate Warming, Atmosphere,15(1), 12,

https://doi.org/ 10.3390/atmos15010012, 2023.

Sun, Z., Zhao, L., Hu, G., Zhou, H., Liu, S., Qiao, Y., Du, E., Zou, D., and Xie, C.: Numerical simulation of thaw settlement and permafrost changes at three sites along the Qinghai-Tibet 1145 Engineering Corridor in a warming climate, Geophys. Res. Lett., 49, e2021GL097334, https://doi.org/10.1029/2021GL097334, 2022.

Sun, Z.: Simulation of permafrost dynamics and thaw settlement along the Qinghai-Tibetan Engineering Corridor, University of Chinses Academy of Science, 2021(Chinese with English abstract).

Westermann, S., Peter, M., Langer, M., Schwamborn, G., Schirrmeister, L., Etzelmüller, B., and Boike, J.: Transient modeling of the ground thermal conditions using satellite data in the Lena River delta, Siberia, The Cryosphere, 11, 1441–1463, https://doi.org/10.5194/tc-11-1441-2017, 2017.

Westermann, S., Schuler, T. V., Gisnås, K., and Etzelmüller, B.: Transient thermal modeling of permafrost conditions in Southern Norway, The Cryosphere, 7, 719–739, https://doi.org/10.5194/tc-7-719-2013, 2013.

Wu, J., Sheng., Y., Wu, Q., Zhi, W.: Processes and modes of permafrost degradation on the Qinghai-Tibet plateau. Sci China Earth Sci. 2010;53(1):150-158. https://doi.org/10.1007/s11430-009-0198-5

Yershov, E.: Principles of Geocryology, Lanzhou University Press, Lanzhou, China, ISBN 9787311048570, 2016 (in Chinse).

Yan, D., Ma, N., Zhang, Y.: Development of a fine-resolution snow depth product based on the snow cover probability for the Tibetan Plateau: Validation and spatial–temporal analyses. Journal of Hydrology,604, 127027,2022.

Zhao, J., Zhao, L., Sun, Z., Niu, F., Hu, G., Zou, D., Liu, G., Du, E., Wang, C., Wang, L., Qiao, Y., Shi, J., Zhang, Y., Gao, J., Wang, Y., Li, Y., Yu, W., Zhou, H., Xing, Z., Xiao, M., Yin, L., and Wang, S.: Simulating the current and future northern limit of permafrost on the Qinghai–Tibet Plateau, The Cryosphere, 16, 4823–4846, https://doi.org/10.5194/tc-16-4823-2022, 2022.

Zou, D., Zhao, L., Sheng, Y., Chen, J., Hu, G., Wu, T., Wu, J., Xie, C., Wu, X., Pang, Q., Wang, W., Du, E., Li, W., Liu, G., Li, J., Qin, Y., Qiao, Y.,Wang, Z., Shi, J., and Cheng, G.: A new map of permafrost distribution on the Tibetan Plateau, The Cryosphere, 11, 2527–2542, https://doi.org/10.5194/tc-11-2527-2017, 2017.

2. Aiming at model forcing, as the only model forcing variable, this study adopted three statistical and machine-learning approaches to extent the land surface temperature from Zou et al (2014, 2017). I was wondering why the authors selected these eight specific input variables—surface air temperature, precipitation, skin temperature, soil temperature, fractional cold over, surface radiation budget, leaf area index, and digital elevation model—for the statistical and machine learning approaches. Could the authors clarify whether including more (or fewer) variables

might help to avoid issues of model underfitting or overfitting? Besides, furthermore, what it is the basis for selecting the particular datasets used for these variables? For example, the study utilizes the uppermost soil temperature from CFSR, while skin temperature is taken from ERA5-land. Given that ERA5-land also provides uppermost soil temperature data at a higher spatiotemporal resolution compared to CFSR.

**Response:**

We selected these eight specific input variables based on physical relevance, expert knowledge, and a thorough review of related published literature (Wang et al., 2022; Xu et al., 2018; Janatian et al., 2017; Yang et al., 2023), which guided us to select variables having close relationships with LST as input. Moreover, data quality and availability on QTP were even more important considerations. We selected variables that had consistent, long-term records that satisfied the requirements of our study.

Regarding model complexity and model underfitting or overfitting, we conducted preliminary experiments and for each machine learning approach, we tested both training and validation errors across different variable combinations. For example, models using only air temperature, as shown in our previous work (Xing et al., 2023), demonstrated signs of underfitting with systematic errors. However, adding related variables significantly improved model performance according to our cross-validation tests. However, adding related variables significantly improved model performance according to our cross-validation tests. Thus, we believe the eight-variable configuration that closely represents LST dynamics provided the optimal balance between model performance and parsimony, performing better than simpler models with fewer variables. Conversely, whether adding additional parameters, such as wind speed, humidity, soil moisture, and snow cover, would lead to model overfitting is unknown at present. These additional data products exhibit considerable variability in quality across the permafrost zone of the QTP. Investigating their potential integration will be the focus of future work.

As for soil temperature selection, our choices were primarily driven by data quality assessments. While ERA5-land is a good choice that provides soil temperature at different depths at a higher spatiotemporal resolution than CFSR, we selected CFSR soil temperature data based on its better performance in preliminary validation against our long-term continuous observations in the permafrost zone on the QTP. The validation results suggested CFSR soil temperature products were closer to the observations at different depths in the permafrost zone on the QTP despite its coarser resolution (Hu et al., 2018). This explains our use of different sources for shallow soil temperature and skin temperature. This mixed-source approach allowed us to leverage the strengths of each

dataset while compensating for their respective limitations in our specific study area.

In any way, we will try to do more work on the issues raised by the reviewers.

Reference:

Wang, X., Zhong, L., Ma, Y.: Estimation of 30 m land surface temperatures over the entire Tibetan Plateau based on Landsat-7 ETM+ data and machine learning methods. International Journal of Digital Earth,15(1), 1038-1055, 2022.

Xu, Y., Knudby, A., Shen, Y., Liu, Y.: Mapping monthly air temperature in the Tibetan Plateau from MODIS data based on machine learning methods. IEEE journal of selected topics in applied earth observations and remote sensing, 11(2), 345-354, 2018.

Janatian, N., Sadeghi, M., Sanaeinejad, S. H., Bakhshian, E., Farid, A., Hasheminia, S. M., Ghazanfari, S.: A statistical framework for estimating air temperature using MODIS land surface temperature data. International Journal of Climatology, 37(3), 1181-1194, 2017.

Yang, Y., You, Q., Jin, Z., Zuo, Z., Zhang, Y., Kang, S.: The reconstruction for the monthly surface air temperature over the Tibetan Plateau during 1901–2020 by deep learning. Atmospheric Research, 285, 106635, 2023.

Xing, Z., Zhao, L., Fan, L., Hu, G., Zou, D., Wang, C., Liu, S., Du, E., Xiao, Y., Li, R., Liu, G., Qiao, Y., and Shi, J.: Changes in the ground surface temperature in permafrost regions along the Qinghai–Tibet engineering corridor from 1900 to 2014: a modified assessment of CMIP6, Adv. Clim. Chang. Res.,14(1), 85-96, https://doi.org/10.1016/j.accre.2023.01.007, 2023.

Hu, G., Zhao, L., Li, R., Wu, X., Wu, T., Xie, C., Zhu, X., and Su, Y.: Variations in soil temperature from 1980 to 2015 in permafrost regions on the Qinghai-Tibetan Plateau based on observed and reanalysis products, Geoderma, 337, 893-905, https://doi.org/10.1016/j.geoderma.2018.10.044, 2019.

**Specific comments:**

1.  Line 40: Smith et al 2022? Maybe it is a wrong reference?

**Response:**

Yeah, it was indeed a typo. The correct reference should be **Smith et al., 2022**, referring to: "Smith, S., O'Neill, H., Isaksen, K., Noetzli, J., and Romanovsky, V.: The changing thermal state of permafrost, Nat. Rev. Earth Environ., 3, 10–23, https://doi.org/10.1038/s43017-021-00240-1, 2022." We have corrected this in the revised manuscript.

2.  Line 49: the reference (Zhao et al., 2019) seemed missing?

**Response:**

Here should be cited as Zhao et al., 2019a, referring to the following reference: "Zhao, L., Hu, G., Zou, D., Wu, X., Ma, L., Sun, Z., Yuan, L., Zhou, H., and Liu, S.: Permafrost Changes and Its Effects on Hydrological Processes on the Qinghai-Tibet Plateau, Bull. Chin. Acad. Sci., 34, 1233–1246, DOI: 10.16418/j.issn.1000-3045.2019.11.006, 2019a."

Thank you for pointing this out. In the revised manuscript, we have carefully checked and made the necessary technical corrections throughout the text.

3.  Line 51: should the Qinghai-Tibet Highway and Railway be abbreviated as QTH? Not sure.

**Response:**

Exactly, the Qinghai-Tibet Highway and Railway should not be abbreviated as QTH. After reviewing the related literature, we found that QTH is a commonly recognized abbreviation for the Qinghai-Tibet Highway, while the Qinghai-Tibet Railway is widely abbreviated as QTR. In the revised manuscript, we use separate abbreviations (QTH and QTR) to avoid confusion.

**Reference:**

Ma, W., Cheng, G., Wu, Q.: Construction on permafrost foundations: lessons learned from the Qinghai–Tibet railroad. Cold regions science and technology,59(1), 3-11, doi: 10.1016/j.coldregions.2009.07.007, 2009.

Gu, W., Yu, Q., Qian, J., Jin, H., Zhang, J.: Qinghai-Tibet expressway experimental research. Sci. Cold Arid Reg, 2(5), 396-404, DOI: 10.3724/ SP.J.1226.2010.00396, 2010.

Jin, H., Luo, D., Wang, S., Lü, L., Wu, J.: Spatiotemporal variability of permafrost degradation on the Qinghai-Tibet Plateau. Sciences in Cold and Arid Regions, 3(4), 281-305, DOI: 10.3724/SP.J.1226.2011.00281, 2011.

Lin, Z., Niu, F., Luo, J., Lu, J., Liu, H.: Changes in permafrost environments caused by construction and maintenance of Qinghai-Tibet Highway, Journal of Central South University, 18(5), 1454-1464. DOI: 10.1007/s11771−011−0861−9, 2011.

4.  Line 84: "ALT" should be given tis full name, this is the first time it has been abbreviated. And "refreezing" of what?

**Response:**

Here, I want to express that, based on our previous simulations, the MVPM can provide sufficient accuracy to capture the annual dynamics of active layer thawing and refreezing, but it seems that 'thawing' was missing. In the revised manuscript, we have corrected this in revised manuscript.

5. Line 161: dose this study activates the settlement module?

**Response:**

No, in this study, we did not activate the ground subsidence module of the MVPM model. Please refer to the answer to the first question.

6. Line 183: what is the surface radiation budget? Net radiation? Net shortwave radiation? Or net longwave radiation? It is not clear.

**Response:**

Net radiation, and we have corrected this in revised manuscript.

7. Line 189: the resolution of all input data is not daily.

**Response:**

Exactly, we have corrected in revised manuscript. The text there reads as:

"Monthly averages were then calculated from the available data (which varied in temporal resolution across datasets), and missing values were filled by interpolating from nearby data."

8. Line 271: how to deal with initial water/ice content?

**Response:**

Initial water/ice content was determined using moisture content measurements from representative borehole cores of various Quaternary sedimentary types (fluvioglacial, lacustrine, alluvial, and aeolian sediments) collected during field investigations. These initial values were subsequently fine-tuned during model calibration to optimize performance. Volumetric ice content is highest in fluvioglacial sediments, followed by lacustrine sediments and weathered residual slide rock. In the vertical profile, ground ice is concentrated at the permafrost table on the plateau, where the ALT typically ranges from 2-3 m. Ice content increases with depth from 3 to 10 m and then remains relatively stable below 10 m.

9. Line 274: "approximately"? it should be an exact number for the grid cell to be simulated.

**Response:**

I agree, this was not an appropriate expression and has been corrected in the revised manuscript. The text there reads as:

"Excluding lake and glacier grids, 47,284 grid cell were used for computation in this study area."

10. Line 300: "filed investigation and borehole monitoring datasets"? I guess it is "Field".

**Response:**

Yes, exactly, we made a typo. Thank you so much for pointing it out. We have also carefully checked and corrected similar typographical errors throughout the text.

11. Line 430: "Grey Shading"? only saw the grey line.

**Response:**

Yes, it should be 'grey line' instead of 'Grey Shading.' This has been corrected.

12. Section 4.2.4: the author states, "permafrost area in the West Kunlun kept stable from 1980 to 1999, decrease in the 2000s, while increase between 2010 and 2022." However, the MVPM model is just forced by land surface temperature, while showed an increasing trend between 1980 and 2022 (Figure 4). Could the author explain why permafrost area increase between 2010 and 2022?

**Response:**

Thank you for this insightful question. This apparent contradiction can be explained by two key factors:

Firstly, while the regional average land surface temperature indeed shows an overall increasing trend from 1980 to 2022, considerable interannual and spatial variability exists within the study area. When analyzed across different periods, some localized areas experienced cooling periods within the overall warming trend. We think these periodic cooling events contributed to new permafrost formation and expansion in those specific areas.

Secondly, the thermal response times of different soil layers to changes in surface temperature vary significantly and include substantial time lags, especially in deeper soil layers. This delayed response is heavily influenced by the presence of ground ice and specific soil properties within the permafrost. The thermal inertia of ice-rich permafrost can delay warming responses by years or even decades. Therefore, the simulated permafrost coverage showing a slight increase between 2010-

2022 may partially reflect the complex delayed response to earlier climate conditions rather than simply the contemporaneous surface temperature trends. These results highlight the complex, non-linear, and slow delaying processes in the response of the permafrost thermal regime in WKL to a warming climate.

13. Line 601-609: so how about the reanalysis data (like Chinese meteorological forcing datasets, ERA5 land)? Compared with the forcing data from ESMs, the spatiotemporal resolution of them is better.

**Response:**

Exactly, reanalysis or assimilated data products indeed offer spatiotemporal resolution than ESM outputs, but their accuracy in permafrost regions on QTP remains problematic due to limited observational constraints (Jiao et al., 2023). Hu et al. (2018) evaluated soil temperature products derived from reanalysis products and found that GLDAS-NOAH and ERA-Interim showed poor performance in the permafrost region of the QTP when compared with soil temperature observations. Similarly, Yang et al. (2020) reported that widely used reanalysis products, such as CFSv2, ERA-Interim, GLDAS-Noah, and ERA5, can capture temporal dynamics of soil temperature but drastically underestimate soil temperature during the thawing period. Regarding Chinese reanalysis datasets specifically, Hu et al. (2024) found that the China Meteorological Administration Land Data Assimilation System (CLDAS) land surface temperature dataset performs well in most regions of China. However, due to the lack of measurement data in the permafrost region of the QTP and insufficient consideration of the unique underlying surface characteristics of permafrost, CLDAS showed significant errors (bias=2.09°C, MAE=3.64°C, RMSE=4.67°C), primarily overestimating temperatures compared to observations. While reanalysis products indeed offer better spatiotemporal resolution than ESM outputs, their accuracy in permafrost regions on QTP remains problematic.

In the revised manuscript, we have supplemented information about current reanalysis or assimilated soil temperature products and their application in the permafrost region of the QTP. The text there reads as: Most previous evaluations indicated that soil temperature products derived from atmospheric circulation models or ESMs, which typically have coarse resolutions (~300 km), show larger uncertainties over the QTP, particularly in the permafrost region (Hu et al., 2019; Xi et al., 2023). While reanalysis products or assimilated soil temperature datasets indeed offer better spatiotemporal resolution than ESM outputs, their accuracy in permafrost regions remains problematic due to limited observational constraints (Jiao et al., 2023; Hu et al., 2024; Yang et al., 2020).

**Reference:**

Yang, S., Li, R., Wu, T., Hu, G., Xiao, Y., Du, Y., Qiao, Y.: Evaluation of reanalysis soil temperature and soil moisture products in permafrost regions on the Qinghai-Tibetan Plateau. Geoderma, 377, 114583, 2020.

Hu, J., Zhao, L., Wang, C., Hu, G., Zou, D., Xing, Z., Jiao, M., Qiao, Y., Liu, G., Du, E.: Applicability evaluation and correction of CLDAS surface temperature products in permafrost region of Qinghai-Tibet Plateau, Climate Change Research, 20 (1): 10-25, 2024.

Hu, G., Zhao, L., Li, R., Wu, X., Wu, T., Xie, C., Zhu, X., and Su, Y.: Variations in soil temperature from 1980 to 2015 in permafrost regions on the Qinghai-Tibetan Plateau based on observed and reanalysis products, Geoderma, 337, 893-905, https://doi.org/10.1016/j.geoderma.2018.10.044, 2019.

Jiao, M., Zhao, L., Wang, C., Hu, G., Li, Y., Zhao, J., Zou, D., Xing, Z., Qiao, Y., Liu, G.: Spatiotemporal Variations of Soil Temperature at 10 and 50 cm Depths in Permafrost Regions along the Qinghai-Tibet Engineering Corridor. Remote Sens. 2023, 15, 455. https://doi.org/10.3390/rs15020455, 2023.

Xing, Z., Zhao, L., Fan, L., Hu, G., Zou, D., Wang, C., Liu, S., Du, E., Xiao, Y., Li, R., Liu, G., Qiao, Y., and Shi, J.: Changes in the ground surface temperature in permafrost regions along the Qinghai–Tibet engineering corridor from 1900 to 2014: a modified assessment of CMIP6, Adv. Clim. Chang. Res.,14(1), 85-96, https://doi.org/10.1016/j.accre.2023.01.007, 2023.

14. Could the author explain how this study can be extended to be the future projections? Due to its inability to obtain the land surface temperature with higher resolution from remote sensing data in the future, how to diagnose the future condition of permafrost in West Kunlun.

**Response:**

Near-surface ground temperature responds to seasonal variations in air temperature (AT), but these fluctuations gradually diminish with depth until becoming negligible at the zero annual amplitude (ZAA) depth, typically 10–20m in the permafrost zone of the QTP (Lunardini et al., 1995; Jin et al., 2011; Dobiński et al., 2022). At the ZAA, seasonal signals are absent, and long-term climate trends dominate, causing ground temperatures to reflect persistent changes in air temperature rather than short-term variability (Smith and Riseborough, 1983; Buteau et al., 2004). This relationship forms a robust basis for projecting permafrost responses to climate change, even in the absence of high-resolution remote sensing data. In particular, the thermal response of permafrost under various future climate scenarios can be investigated by modeling ground temperature profiles driven by a linearly increasing air temperature. For example, the Sixth Assessment Report of the Intergovernmental Panel on Climate Change Working Group I (IPCC

WG1 AR6) (IPCC, 2021) has evaluated and projected climate change over the QTP during the 21st century (https://interactive-atlas.ipcc.ch, last access: April 1, 2025). The model estimated warming between 1995–2014 and 2081–2100 of the mean annual AT in the QTP under three RCP scenarios as 0.013°C $a^{-1}$ (RCP2.6, low concentration of emissions), 0.028°C $a^{-1}$ (RCP4.5, stable concentration of emissions), and 0.060°C $a^{-1}$ (RCP8.5, high concentration of emissions) calculated from the multi-model ensemble median (21–29 model outputs) of CMIP5. The mean warming rate is 0.017°C $a^{-1}$ (SSP1-2.6, strong climate change mitigation), 0.032°C $a^{-1}$ (SSP2-4.5, moderate mitigation), and 0.064°C $a^{-1}$ (SSP5-8.5, no mitigation), estimated from the CMIP6 ensemble median of 31–34 model outputs. A similar projection can be found in our following published literature: Hu et al. (2015), Li et al. (1999), Sun et al. (2020), and Zhao et al. (2022).

**Reference:**

Smith, M., Riseborough, D.: Permafrost sensitivity to climatic change. In Proceedings, 4th International Conference on Permafrost, Vol. 1. Fairbanks, Alaska, National Academy Press: Washington, DC; 1178–1183, 1983.

Buteau, S., Fortier, R., Delisle, G., and Allard, M.: Numerical simulation of the impacts of climate warming on a permafrost mound, Permafrost and Periglac. Process., 15, 41-57, https://doi.org/10.1002/ppp.474, 2004.

Hu, G., Zhao, L., Wu, X., Li, R., Wu, T., Xie, C., Pang, Q., Xiao, Y., Li, W., Qiao, Y., Shi, J.: Modeling permafrost properties in the Qinghai-Xizang (Tibet) Plateau. Science China: Earth Sciences, 58: 2309–2326, doi: 10.1007/s11430-015-5197-0, 2015.

Sun, Z., Zhao, L., Hu, G., Qiao, Y., Du, E., Zou, D., Xie, C., 2020. Modeling permafrost changes on the Qinghai-Tibetan plateau from 1966 to 2100: A case study from two boreholes along the Qinghai-Tibet engineering corridor. Permafr. Periglac. Process. 31 (1), 156–171.

Zhao, J., Zhao, L., Sun, Z., Niu, F., Hu, G., Zou, D., Liu, G., Du, E., Wang, C., Wang, L., Qiao, Y., Shi, J., Zhang, Y., Gao, J., Wang, Y., Li, Y., Yu, W., Zhou, H., Xing, Z., Xiao, M., Yin, L., and Wang, S.: Simulating the current and future northern limit of permafrost on the Qinghai–Tibet Plateau, The Cryosphere, 16, 4823–4846, https://doi.org/10.5194/tc-16-4823-2022, 2022.

Lunardini, V.: Permafrost Formation Time. CRREL Report 95-8, US Army Corps of Engineers, Cold Regions Research and Engineering Laboratory, 1995.

Jin, H., Luo, D., Wang, S., Lü, L., and Wu, J.: Spatiotemporal variability of permafrost degradation on the Qinghai-Tibet Plateau, Sci. Cold Arid Reg., 3, 281–305, DOI: 10.3724/SP.J.1226.2011.00281, 2011.

Dobiński, W., and Marek K.: Permafrost Base Degradation: Characteristics and Unknown Thread With Specific Example From Hornsund, Svalbard, Front. Earth Sci. 10:802157, doi:

10.3389/feart.2022.802157, 2022.

IPCC. Climate change 2021: the physical science basis, https://www.ipcc.ch/report/ar6/wg1/downloads/report/IPCC_AR6_WGI_Full_Report.pdf., 2021.

---

## Author Comment (AC2)

**Response to Referee#2 egusphere-2024-3956**

This study investigates permafrost dynamics in the West Kunlun region of the Qinghai-Tibet Plateau (QTP) using a high-resolution permafrost modelling approach. The authors employ the previously developed Moving-Grid Permafrost Model (MVPM) to reconstruct permafrost thermal conditions over the past four decades, integrating remote sensing data from a previously published dataset, machine learning techniques, and filed observations. The study provides valuable insights into the effects of climate change on permafrost, demonstrating a significant warming trend in land surface temperature (LST) while indicating a relatively stable permafrost extent. The methodology is reasonably innovative and well-structured, offering an improvement over previous large-scale permafrost models by considering detailed processes at depth. While the study is well-executed and contributes to our understanding of permafrost changes in this region, there are areas that require further clarification and refinement as detailed below:

**Response:**

We greatly appreciate your detailed comments, revisions, and suggestions, which have helped us improve the quality of the manuscript. We provide our responses to each comment individually. The original reviewer comments are in black font, while our responses appear in blue font. The corresponding edits in the manuscript are highlighted in red font.

**Major Comments:**

1. Model validation and uncertainties

The authors report high model accuracy ($\pm0.25\,°C$ for ground temperature and $\pm0.25\,m$ for active layer thickness). However, the discussion of model uncertainties could be expanded. Key areas for improvement include:

* a sensitivity analysis of key model parameter such as soil thermal properties and initial boundary conditions to assess their impact on the results.

* a deeper discussion on the limitations of the forcing datasets, particularly the machine-learning-based reconstruction of LST prior to 2003 and impact of the cold bias of 0.8degC (Compared to in situ measurements, we found a slight cold bias in our reconstructed LST series, averaging approximately $-0.80°C$)

**Response:**

To quantitatively assess model uncertainties, we conducted a one-at-a-time sensitivity analysis

(Figure 1) using three representative boreholes that span different types of frozen ground—stable permafrost, unstable permafrost, and seasonally frozen ground (see Table 1 for details). Key model parameters, including soil thermal conductivity, heat capacity, water/ice content, initial temperature profile, and upper boundary temperature, were systematically perturbed by ±10% to evaluate their relative influence on the simulated mean annual ground temperature (MAGT, at 15 m depth) and active layer thickness (ALT).

Our results show that upper boundary temperature (e.g., model forcing) has the strongest influence on MAGT across all ground types. However, the magnitude of its impact remains relatively small: around ±0.5 °C in seasonally frozen ground and ≤ ±0.1 °C in both stable and unstable permafrost. ALT exhibits similarly modest sensitivity, with variations of approximately ±0.1 m in stable permafrost and ±0.05 m in unstable permafrost. Soil thermal conductivity and water/ice content have a more pronounced effect on ALT, particularly in unstable permafrost, where a 10% change in these parameters can lead to a 0.05–0.1 m variation. In contrast, heat capacity has minimal influence on both MAGT and ALT. Initial ground temperature shows a moderate effect in seasonally frozen ground (~±0.12 °C), but exerts negligible influence in permafrost areas. These findings suggest that the model achieves thermal stability over the simulation period and that uncertainties in individual parameters exert only limited influence on overall model performance.

We acknowledge that the machine-learning-based reconstruction of LST, particularly before 2003, introduces uncertainty, most notably an average cold bias of approximately -0.8 °C relative to in situ measurements. While this bias may affect the absolute values of the upper boundary forcing, our sensitivity analysis suggests that the model is relatively robust to such small shifts in LST, especially in permafrost regions. Nonetheless, we recognize this cold bias as a key source of systematic uncertainty and will expand the discussion in the revised manuscript.

**Table 1. Information on three representative borehole sites used for one-at-a-time sensitivity analysis**

| Borehole | Description |
|---|---|
| ZK30 | The borehole reaches a depth of 15 m, with the ground primarily composed of fine sand and silty sand. The MAGT is -1.66 °C, and the ALT is 2.4 m, classifying the site as stable permafrost. |
| ZK12 | The borehole has a drilling depth of 13.5 m, with a vegetation-free surface. The core consists primarily of Fluvial sand and sand. Frozen soil was first encountered at a depth of 4.9 m, where small ice crystals are evenly distributed within a granular soil structure. Below 5.5 m, the frozen layer disappears, accompanied by a noticeable increase in ground temperature. The 4.9–5.5 m interval represents a transition zone, and the site is classified as unstable permafrost. |

| ZK13 | No frozen soil was encountered during the drilling process, and the site is classified as seasonally frozen ground |
|---|---|

Note: This information is compiled from Li et al. (2012) and Zhao et al. (2019).

[Figure]

**Figure 1. One-at-a-time sensitivity analysis showing the effects of ±10% variation in individual model parameters, e.g., soil thermal conductivity, heat capacity, water/ice content, initial temperature, and upper boundary temperature—on (top row) mean annual ground temperature (MAGT) at 15 m depth and (bottom row) active layer thickness (ALT), across three ground conditions: stable permafrost (left), unstable permafrost (middle), and seasonally frozen ground (right).**

**Reference:**

Li, K., Chen, J., Zhao, L., Zhang, X., Pang, Q., Fang, H., Liu, G.: Permafrost distribution in typical area of west Kunlun Mountains derived from a comprehensive survey (in Chines with English abstract), J. GLACIOL.,2012.

Zhao L, Sheng Y. Permafrost and environment changes on the Qinghai-Tibetan Plateau. Beijing, China: Science Press.; 2019.

\* The effect of snow cover on LST reconstruction and subsurface thermal dynamics is not sufficiently addressed. How are snow-insulated period accounted for? Is there any bias introduced by could-covered or snow-covered days in the satellite-derived LST? (linked to point 2)

**Response:**

   Firstly, in the vast permafrost zone of the QTP, strong solar radiation and wind result in rare, thin (~3cm) snow cover that typically lasts less than one day per snow event (Wu and Zhang, 2008;

Che et al., 2008; Zou et al., 2017; Yan et al., 2022). Vegetation in the alpine ecosystem of the permafrost region consists of grassland characterized by dwarf and sparsely distributed plants, with vegetation cover below 10% in the western QTP (Wang et al., 2016). Given the specific conditions of snow cover and vegetation in the permafrost zone of the QTP, the average thermal offset between GST and LST is minimal (Hachem et al., 2012).

Secondly, the subsurface thermal model MVPM, which uses land surface temperature as the upper boundary forcing, employed a modified LST product from MODIS developed by Zou et al. (2017). This product partially accounts for the influence of surface conditions including the effects of snow cover, vegetation and cloud cover on LST through a cloud gap filling algorithm, as well as incorporating automatic weather station (AWS) observations from representative permafrost regions in the central QTP. These AWS observations, which reflect climate conditions at satellite overpass times, were used to calibrated MODIS LST and calculate mean daily LST values at those times and included in the model training dataset.

Moreover, our model simulations reasonably reproduce the mean annual ground temperature, and the simulated active layer thickness is in good agreement with observations as well as permafrost distribution in different periods. While thin snow cover might have a cooling effect on ground surface temperature due to the high albedo of fresh snow and rapid snowmelt processes (Zhang et al., 2005), we believe this cooling effect is of short duration and has minimal impact at our simulation time scales. On the contrary, snow cover effects would be more significant for centennial to millennial time scale simulations rather than the decadal scale used in our study (**please see response to the second question about 'use of MODIS LST as forcing data'**).

**Reference:**

Che, T., Xin, L., Jin, R., Armstrong, R., and Zhang, T.: Snow depth derived from passive microwave remote-sensing data in China, Ann. Glaciol., 49, 145–154, https://doi.org/10.3189/172756408787814690 , 2008.

Yan, D., Ma, N., Zhang, Y.: Development of a fine-resolution snow depth product based on the snow cover probability for the Tibetan Plateau: Validation and spatial–temporal analyses. Journal of Hydrology,604, 127027,2022.

Wu, Q. and Zhang, T.: Recent permafrost warming on the Qinghai-Tibetan Plateau, J. Geophys. Res., 113, 1–22, 2008.

Zou, D., Zhao, L., Sheng, Y., Chen, J., Hu, G., Wu, T., Wu, J., Xie, C., Wu, X., Pang, Q., Wang, W., Du, E., Li, W., Liu, G., Li, J., Qin, Y., Qiao, Y.,Wang, Z., Shi, J., and Cheng, G.: A new map of permafrost distribution on the Tibetan Plateau, The Cryosphere, 11, 2527–2542,

https://doi.org/10.5194/tc-11-2527-2017, 2017.

2. Use of MODIS LST as forcing data

While MODIS LST provides high spatial resolution and extensive temporal coverage, it poses several challenges when used to force model models:

* MODIS measures skin temperature rather than subsurface ground temperature, which can differ significantly—especially under snow cover or vegetation (vegetation is acknowledged in this study)

* Snow cover introduces thermal insulation, decoupling surface LST from the subsurface thermal regime (Albeit often nonexistent or thin snow cover). However, wind driven snow drift can be significant and give spatial heterogeneity at high model resolutions. The significance of these effects needs to be addressed in this study region.

* Cloud cover causes data gaps, which can lead to temporal inconsistencies or biases if gap-filling methods are not robust—particularly problematic in winter months?

* MODIS LST captures only clear-sky conditions, potentially biasing the dataset toward colder nighttime or warmer daytime extremes, depending on retrieval timing.

The paper should better articulate how these limitations are mitigated in the LST reconstruction, and what implications they have for subsurface heat fluxes and permafrost thermal state. A comparison with measured air-temperature or reanalysis-based forcing would also help.

**Response:**

Surface (i.e., skin) temperature serves as model forcing at the upper model boundary in our study, where we applied machine learning techniques to reconstruct remote-sensing-based LST. We acknowledge the limitations highlighted by the reviewer: MODIS LST is skin temperature e.g., vegetation canopy, snow rather than ground surface temperature. Snow cover introduces thermal insulation effects; cloud cover causes data gaps; and MODIS LST captures only clear-sky conditions. We implemented several preprocessing procedures to mitigate these uncertainties:

Firstly, we employed a modified Moderate Resolution Imaging Spectroradiometer land surface temperature (MODIS LST) product provided by Zou et al. (2017), available since 2003. This product was derived from the clear-sky MOD11A2 (Terra MODIS) and MYD11A2 (Aqua MODIS) datasets (Collection 6), which provide two observations per day (daytime and nighttime) for the same pixel. Prior to analysis, irregularly spaced time series caused by cloud cover or other factors were identified, and data gaps were filled using the Harmonic Analysis of Time Series (HANTS) algorithm (Xu et al., 2013; Zou et al., 2017). This method has proven effective for filling gaps in MODIS LST data over the QTP (Xu et al., 2013) and helps mitigate the cold bias associated with

clear-sky temporal averages.

Secondly, to estimate daily mean LST values from Aqua and Terra instantaneous daytime and nighttime observations, multi-stepwise statistical model were established using ground-based ground surface temperature (GST) data from AWS in typical permafrost regions of the central QTP. These relationships, which reflect actual climate conditions at satellite overpass times, were developed by Zou et al. (2014, 2017). The resulting empirical correction model was then applied across the entire QTP permafrost zone to upscale and estimate reliable LST values. Previous validation by Zou et al. (2014) at three typical permafrost monitoring sites with different land cover types—alpine steppe (Xidatan), alpine meadow (Tangula), and alpine desert (Wudaoliang)— demonstrated strong model performance, with coefficients of determination ($R^2$) ranging from 0.91 to 0.93, mean errors between –0.21°C and 1°C, mean absolute errors (MAE) from 2.28°C to 2.42°C, and root mean square errors (RMSE) between 2.96°C and 3.05°C. In the West Kunlun permafrost zone, we further evaluation using monthly in situ data from the TSH AWS (81.4°E, 36.0°N, 5019 m a.s.l.) for the period 2016–2018 also showed strong agreement, with $R^2$ exceeding 0.90, MAE of 1.62°C, RMSE of 2.09°C. These results confirm that the empirical model reliably captures the spatial patterns of LST across the QTP.

Third, in this study, the modified LST product from Zou et al. (2017) was further refined to reconstruct historical LST data prior to 2003—extending back to 1980—using machine learning techniques that integrate eight specific variables derived from ground-based observations, satellite data, reanalysis datasets, and other sources. The reconstructed LST was evaluated against monthly in situ observations from the TSH AWS for the period 2016–2018, showing strong correlations above 0.95, with RMSE values from 1.62°C to 1.91°C and MAE ranging from 1.29°C to 1.50°C. The validation results show that the reconstructed LST performs slightly better than the satellite-derived LST_Zou, with a particularly notable improvement over reanalysis product ERA5-Land (Figure 2)

Fourth, while direct validation of pre-2003 LST is not possible due to the lack of satellite or ground observations in the West Kunlun region, we employed an indirect validation approach. Specifically, the reconstructed LST was used to force the MVPM to simulate permafrost thermal dynamics from 1980 onward. The simulation results were then evaluated against existing permafrost monitoring network data and previously published permafrost distribution maps from different periods, i.e., the 1980s (Li et al., 1996), the 2000s (Wang et al., 2006), 2010 (Cao et al., 2023), and after 2010 (Zou et al., 2017). The strong agreement between the MVPM outputs and these

independent data sources supports the reliability of the pre-2003 LST reconstruction. Furthermore, our analysis showed that the West Kunlun permafrost survey area has experienced pronounced LST warming since the mid-1980s, with an accelerated warming trend in the last decade. This pattern aligns well with the documented warming trends on the QTP in recent studies (Jin et al., 2011; You et al., 2021; Yao et al., 2019; Li et al., 2024), indirectly validating the accuracy of our reconstructed LST data. These multi-faceted validation approach provides reasonable confidence in our LST dataset, despite the absence of direct observations for the earlier period. While we acknowledge this limitation, we believe the methodology offers the most robust solution given the data constraints in this remote and observational challenging region.

The above comparisons demonstrate that the reconstructed LST time series closely align with in situ measurements and provide sufficient accuracy for ground thermal modeling in our study. However, a systematic cold bias is evident in the mean annual cycle of LST, particularly during the summer months of July, August, and September (Figure 2). Compared to LST_Zou, the reconstructed monthly LST reduce this bias to varying degrees. Nevertheless, a residual cold bias remains noticeable in the reconstructed LST during these months. This bias likely contributes to the underestimation of shallow soil temperatures, which in turn affects the simulation of active layer thickness or thaw depth. Near-surface ground temperature is highly sensitive to seasonal variations in model forcing, which are often marked by frequent fluctuations and complex patterns (Lunardini et al., 1995). This may help explain the underestimation of active layer thickness or melt depth observed in our simulations. Similarly, Westermann et al. (2015) reported that inaccuracies in summer LST forcing can directly impact the accuracy of thaw depth simulations; with an LST uncertainty of $\pm 2$ °C, the resulting thaw depth is reproduced with an uncertainty of approximately $\pm 3$ cm. However, in permafrost regions, the seasonal signal becomes increasingly attenuated with depth due to the complex coupling among environmental conditions, thermal properties, phase change, ground ice, and cryoturbation. At the depth of Zero Annual Amplitude (ZAA) temperature fluctuations become undetectable (Jin et al., 2011; Dobiński et al., 2022). The trend in mean annual ground temperature at the ZAA generally mirrors long-term air temperature trends (Smith and Riseborough, 1983; Buteau et al., 2004; Jin et al., 2011). Our sensitivity analysis also demonstrated that the model is relatively robust to such small shifts in LST, especially in permafrost regions. We therefore believe that the existing cold bias, as a limitation of our LST reconstruction approach, is seasonal and transient, and has minimal influence on the long-term permafrost thermal status within the time scale of our simulations.

[Figure]

**Figure 2. Monthly average LST at the TSH AWS from 2016 to 2018, including reanalysis-derived LST (ERA5-Land), modified satellite-derived LST (LST_Zou), estimates from three machine learning models (LR, MLR, and RFR), and in situ observations.**

**Reference:**

Buteau, S., Fortier, R., Delisle, G., Allard, M.: Numerical simulation of the impacts of climate warming on a permafrost mound. Permafr Periglac Process., 15(1):41-57. https://doi.org/10.1002/ppp.474, 2004.

Cao, Z., Nan, Z., Hu, J., Chen, Y., and Zhang, Y.: A new 2010 permafrost distribution map over the Qinghai–Tibet Plateau based on subregion survey maps: a benchmark for regional permafrost modeling, Earth Syst. Sci. Data, 15, 3905–3930, https://doi.org/10.5194/essd-15-3905-2023, 2023.

Dobiński, W., and Marek K.: Permafrost Base Degradation: Characteristics and Unknown Thread with Specific Example from Hornsund, Svalbard, Front. Earth Sci. 10:802157, doi: 10.3389/feart.2022.802157, 2022.

Jin, H., Luo, D., Wang, S., Lü, L., and Wu, J.: Spatiotemporal variability of permafrost degradation on the Qinghai-Tibet Plateau, Sci. Cold Arid Reg., 3, 281–305, DOI: 10.3724/SP.J.1226.2011.00281, 2011.

Li, N., Cuo, L., Zhang, Y., and Ding, J.: The synthesis of potential factors contributing to the asynchronous warming between air and shallow ground since the 2000s on the Tibetan Plateau, Geoderma,441, 116753, https://doi.org/10.1016/j.geoderma.2023.116753, 2024.

Li, S., Cheng, G., and Guo, D.: The future thermal regime of numerical simulating permafrost on the Qinghai-Xizang (Tibet) Plateau, China, under a warming climate, Sci. China D, 39, 434–441, 1996.

Lunardini, V.: Permafrost Formation Time. CRREL Report 95-8, US Army Corps of Engineers,

Cold Regions Research and Engineering Laboratory, 1995.

Smith, M., Riseborough, D.: Permafrost sensitivity to climatic change. In Proceedings, 4th International Conference on Permafrost, Vol. 1. Fairbanks, Alaska, National Academy Press: Washington, DC; 1178–1183, 1983.

Wang, T., Wang, N., and Li, S.: Map of the glaciers, frozen ground and desert in China, 1 V 4 000 000, Chinese Map Press, Beijing, China, 2006.

Westermann, S., Østby, T. I., Gisnås, K., Schuler, T. V., and Etzelmüller, B.: A ground temperature map of the North Atlantic permafrost region based on remote sensing and reanalysis data, The Cryosphere, 9, 1303–1319, https://doi.org/10.5194/tc-9-1303-2015, 2015.

Xu, Y., Shen, Y., and Wu, Z.: Spatial and Temporal Variations of Land Surface Temperature Over the Tibetan Plateau Based on Harmonic Analysis, Mt. Res. Dev., 33, 85–94, 2013.

Yao, T., Xue, Y., Chen, D., Chen, F., Thompson, L., Cui, P., Koike, T., Lau, W. K., Lettenmaier, D., Mosbrugger, V., Zhang, R., Xu, B., Dozier, J., Gillespie, T., Gu, Y., Kang, S., Piao, S., Sugimoto, S., Ueno, K., Wang, L., Wang, W., Zhang, F., Sheng, Y., Guo, W., , Yang, X., Ma, Y., Shen, S. S. P., Su, Z., Chen, F., Liang, S., Liu, Y., Singh, V. P., Yang, K., Yang, D., Zhao, X., Qian, Y., Zhang, Y., and Li, Q.: Recent Third Pole's Rapid Warming Accompanies Cryospheric Melt and Water Cycle Intensification and Interactions between Monsoon and Environment: Multidisciplinary Approach with Observations, Modeling, and Analysis, B. Am. Meteorol. Soc., 100, 423-444, https://doi.org/10.1175/BAMS-D-17-0057.1, 2019.

You, Q., Cai, Z., Pepin, N., Chen, D., Ahrens, B., Jiang, Z., Wu, F., Kang, S., Zhang, R., Wu, T., Wang, P., Li, M., Zou, Z., Gao, Y., Zhai, P., and Zhang, Y.: Warming amplification over the Arctic Pole and Third Pole: Trends, mechanisms and consequences. Earth Sci. Rev.,217, 103625, https://doi.org/10.1016/j.earscirev.2021.103625, 2021.

Zou, D., Zhao, L., Sheng, Y., Chen, J., Hu, G., Wu, T., Wu, J., Xie, C., Wu, X., Pang, Q., Wang, W., Du, E., Li, W., Liu, G., Li, J., Qin, Y., Qiao, Y.,Wang, Z., Shi, J., and Cheng, G.: A new map of permafrost distribution on the Tibetan Plateau, The Cryosphere, 11, 2527–2542, https://doi.org/10.5194/tc-11-2527-2017, 2017.

Zou, D., Zhao, L., Wu, T., Wu, X., Pang, Q., and Wang, Z.: Modeling ground surface temperature by means of remote sensing data in high-altitude areas: test in the central Tibetan Plateau with application of moderate-resolution imaging spectroradiometer Terra/Aqua land surface temperature and ground based infrared radiometer, J. Appl. Remote Sens., 8, 083516, https://doi.org/10.1117/1.JRS.8.083516, 2014.

**3.  Representation of soil stratigraphy**

The study highlights variations in permafrost responses based on soil stratigraphy, but further clarity is needed regarding:

* The specific role of soil moisture and ice content in modulating permafrost temperature trends.

* Potential biases in stratigraphic classification and how these might affect regional variability in permafrost degradation.

* The extent to which subsurface heterogeneity is accounted for in the model.

**Response:**

In our modeling framework, we incorporated detailed thermophysical characterization of the subsurface based on measurements from 15 boreholes distributed across the WKL permafrost survey area, with depths ranging from 15 to 59 m. Core sampling, field observations, and borehole descriptions (Li et al., 2012; Zhao et al., 2019) indicate that ground ice content varies between 5% and 50% in WKL, depending on Quaternary sediment type. Higher ice contents are observed in fine-grained glarosional and lacustrine sediments due to enhanced segregation ice formation, while lower values are typical of coarse-grained alluvial and colluvial deposits. Vertically, ice-rich layers are consistently present near the upper boundary of permafrost, where generally ranges from 2 to 3 m. Ice content tends to slight increase with depth between 3 and 10 m and remains relatively stable below 10 m (Zhao et al., 2010). The site-level stratigraphic and thermophysical data were spatially upscaled using vector-based geomorphological classification maps of western China. The five stratigraphic classes commonly found in the West Kunlun region are glarosional, alluvial plain, aeolian, colluvial valley, and lacustrine deposits

Our simulation results highlight the critical role of ground ice content in shaping permafrost thermal dynamics. Modeled ALT varies remarkable change across stratigraphic classes, with the greatest ALT found in alluvial sediments and the shallowest ALT in glarosional sediments. While some uncertainty in stratigraphic classification and spatial representation is inevitable, our approach is grounded in field observations and measured thermal properties. Furthermore, sensitivity analysis shows that the model outputs are relatively robust to variations in key input parameters related to soil stratigraphy. That said, considerable small-scale heterogeneity in ground properties exists beyond what can be captured at the 1 km modeling resolution, especially complex mountainous. Variability within each sediment class may also lead to biases in local model estimates. Despite these limitations, we are confident that our model captures the essential thermal characteristics of each sediment class, which are central to simulating permafrost thermal regime response to climate change. Continued improvements in soil property datasets particularly in permafrost regions will be vital for refining future model performance.

**Reference:**

Li, K., Chen, J., Zhao, L., Zhang, X., Pang, Q., Fang, H., Liu, G.: Permafrost distribution in typical

area of west Kunlun Mountains derived from a comprehensive survey (in Chines with English abstract), J. GLACIOL.,2012.

Zhao, L., Ding, Y., Liu, G., Wang, S., and Jin, H.: Estimates of the reserves of ground ice in permafrost regions on the Tibetan plateau, J. Glaciol. Geocryol., 32, 1–9, 2010.

Zhao L, Sheng Y. Permafrost and environment changes on the Qinghai-Tibetan Plateau. Beijing, China: Science Press.; 2019.

4.  Permafrost stability and warming trends

The study concludes that despite significant warming trends in LST, permafrost extent remains stable. While plausible given the thermal inertia of deep permafrost, the paper could be benefit from:

* A clearer discussion on why this stability is observed and how it compares with degradation rates reported in other high-altitude or Arctic regions.

* Consideration of potential threshold effect (e.g., rapid degradation once a critical warming threshold is exceeded)

* More discussion on subsurface processes such as latent heat effects and talik formation, which may delay degradation despite rising surface temperatures.

* The observed increase in permafrost area despite a warming trend of 0.4°C per decade is surprising and warrants closer examination and justification.

**Response:**

Permafrost thermal degradation is a complex process characterized by a time lag in response to climate warming and is further modulated by local environmental factors such as soil type, ground ice content, geothermal heat flux, and the initial thermal state of the permafrost (Zhao et al., 2020; 2024; Hu et al., 2023). In response to climate change, permafrost gradually adjusts its thermal regime across multiple timescales—ranging from years to centuries or even millennia. Based on their classification of ground temperature profiles, Wu et al. (2010) suggested that the current diversity in permafrost thermal conditions across the QTP may reflect different stages of degradation (e.g., warming stage, zero geothermal gradient stage, talik development stage, and complete disappearance) dating back to the cold climatic conditions of the Last Glacial Maximum (LGM).

Our study found that approximately 70.98% of the permafrost in the Western Kunlun region is in the temperature-rising stage, characterized by an initial MAGT below –2.0 °C and an ALT of less than 1.5m. This type predominantly occurs in high-elevation areas (above 4800 m a.s.l.). Additionally, 17.58% of the permafrost is transitioning from the temperature-rising stage to the zero

geothermal gradient stage. Only 11.44% is either in the zero geothermal gradient stage or transitioning toward the talik development stage, and may be facing degradation. This type of permafrost is typically located in lower-elevation areas (below 4800 m a.s.l.) and is characterized by a relatively high MAGT (above –1 °C).

Permafrost is a state variable that forms when heat loss at the ground surface exceeds heat gain over a prolonged period under a severely cold climate (Wu et al., 2010). Under a warming climate, sustained increases in ground surface temperature disrupt the thermal equilibrium established under historical climate conditions. Consequently, the active layer begins to accumulate more heat each year than it can release, leading to progressive ground warming from the surface downward and a reduced temperature gradient within the permafrost. However, during the early phase of warming, permafrost temperatures rise more readily than ground thaw occurs, since most of the incoming energy is used to warm the frozen soil to its thawing point. This explains why the areal extent of permafrost in the Western Kunlun region remained relatively unchanged during our simulation period, despite the pronounced warming trend.

Moreover, although the regional average LST shows an overall increasing trend from 1980 to 2022, considerable interannual and spatial variability exists within the study area. We suggest that periodic cooling events contributed to the formation and expansion of new permafrost in specific areas through a complex, delayed response. This is supported by our simulated permafrost coverage, which shows a slight increase between 2010 and 2022 despite the pronounced warming trend during this period.

Under future climate warming scenarios, MAGT will continue to rise. As heat penetrates deeper, the thermal gradient at the permafrost base becomes smaller than the geothermal gradient, causing heat to flow upward from the underlying unfrozen ground. This initiates basal thawing, resulting in a gradual upward retreat of the permafrost base and overall thinning of the permafrost layer. Compared to Arctic and sub-Arctic regions, the QTP has a relatively high geothermal gradient, which contributes to a longer permafrost response time to atmospheric warming (Jin et al., 2011). As a result, the rate of ground temperature increase is noticeably lower in the QTP than in circumpolar regions (Zou et al., 2017).

When permafrost temperatures approach 0°C, ground ice near the permafrost table begins to melt, consuming large amounts of latent heat, a phenomenon known as the "zero curtain effect." This process significantly slows or even temporarily halts further warming for a much longer period than observed in unfrozen soil, and substantially reduces seasonal temperature variations within the

shallow permafrost. Simultaneously, geothermal heat from below is almost entirely used for thawing permafrost from the bottom up.

The zero geothermal gradient stage represents a critical transitional state in permafrost degradation. At this stage, nearly all available heat from the surface is consumed by ice melt, leading to a rapid downward shift in the permafrost table. When the maximum seasonal freezing depth no longer reaches the permafrost table, a talik (unfrozen ground within permafrost) forms and expands. Numerical simulations conducted by Sun et al. (2019) indicated that talik formation coincides with accelerated permafrost thaw and marks the onset of irreversible degradation that continues until complete permafrost loss.

The revised manuscript will include an expanded discussion on permafrost stability.

Reference:

Hu, G., Zhao, L., Li, R., Park, H., Wu, X., Su, Y., Guggenberger, G., Wu, T., Zou, D., Zhu, X., Zhang, W., Wu, Y., and Hao, J: Water and heat coupling processes and its simulation in frozen soils: Current status and future research directions, Catena, 222, 106844, 985 https://doi.org/10.1016/j.catena.2022.106844, 2023.

Zhao, L., Hu, G., Liu, G., Zou, D., Wang, Y., Xiao, Y., Du, E., Wang, C., Xing, Z., Sun, Z., Zhao, Y., Liu, S., Zhang, Y., Wang, L., Zhou, H., and Zhao, J.: Investigation, Monitoring, and Simulation of Permafrost on the Qinghai-Tibet Plateau: A Review, Permafrost Periglac. Process., 35:412–422, https://doi.org/10.1002/ppp.2227, 2024.

Zhao, L., Zou, D. Hu, G., Du, E., Pang, Q., Xiao, Y., Li, R., Sheng, Y., Wu, X., Sun, Z., Wang, L., Wang, C., Ma, L., Zhou, H., and Liu, S.: Changing climate and the permafrost environment on the Qinghai–Tibet (Xizang) Plateau, Permafrost Periglac., 31, 396– 405, https://doi.org/10.1002/ppp.2056, 2020.

Wu, J., Sheng, Y.,Wu, Q., andWen, Z.: Processes and modes of permafrost degradation on the Qinghai-Tibet Plateau, Sci. China D, 53, 150–158, https://doi.org/10.1007/s11430-009-0198-5, 2010.

Jin, H., Luo, D., Wang, S., Lü, L., and Wu, J.: Spatiotemporal variability of permafrost degradation on the Qinghai-Tibet Plateau, Sci. Cold Arid Reg., 3, 281–305, DOI: 10.3724/SP.J.1226.2011.00281, 2011.

Zou, D., Zhao, L., Sheng, Y., Chen, J., Hu, G., Wu, T., Wu, J., Xie, C., Wu, X., Pang, Q., Wang, W., Du, E., Li, W., Liu, G., Li, J., Qin, Y., Qiao, Y.,Wang, Z., Shi, J., and Cheng, G.: A new map of permafrost distribution on the Tibetan Plateau, The Cryosphere, 11, 2527–2542, https://doi.org/10.5194/tc-11-2527-2017, 2017.

5. Implications for future projections

While the paper effectively documents historical changes, it lacks a forward-looking component. While I realize this is beyond the scope of the paper perhaps some additional point could be added to the discussion to enhance its relevance, such as:

* Discussion how the observed trends might evolve under different climate scenarios.

* Assess the potential for abrupt future permafrost degradation/non-linearly of the system and its implications for infrastructure and carbon release.

* Offer suggestions for integrating MVPM outputs into Earth System Models to improve global climate projections.

**Response:**

Thank you for this thoughtful and constructive suggestion. Understanding the current status of permafrost in the context of its historical evolution is essential for projecting future changes, and we agree that incorporating a forward-looking component enhances the relevance of the paper.

In fact, in our previous work (Sun et al., 2019; Zhao et al., 2022), we used MVPM to simulate and assess permafrost thermal dynamics under various future climate change scenarios. Our modeling results indicated that permafrost degradation, particularly in terms of areal extent, does not follow a linear trajectory, and the response of permafrost temperature to climate warming is not as rapid as projected in many published reports (Guo et al., 2012; Ni et al., 2021). Even under the most extreme warming scenario (RCP8.5), the permafrost table was projected to deepen only gradually. By 2050, permafrost would still remain at a depth of 40 m at both Wudaoliang and Tanggula, two borehole sites located in the continuous permafrost zone characterized by cold ground temperatures and thick permafrost layers. In contrast, at Xidatan, a site located at the lower boundary of the permafrost zone, with warmer ground and a thinner permafrost layer of about 32m, the permafrost base is projected to move upward significantly. Nevertheless, permafrost is still expected to persist at this site through 2100 based on projected changes in the deep permafrost ground temperature, ground ice, and thermal gradients.

Similarly, along the northern margin of the permafrost zone on the QTP, MVPM simulation results (Zhao et al., 2022) showed that MAGT would continue to rise under gradual warming scenarios. The warming rate was projected to be slightly higher under Shared Socioeconomic Pathway (SSP) scenarios compared to Representative Concentration Pathway (RCP) scenarios.

However, no significant differences were modeled in the projected areal extent of permafrost between the SSP and RCP scenarios. These results indicate that although permafrost temperatures on the QTP are rising rapidly under climate warming, the rate of permafrost loss, particularly in terms of areal extent is relatively slow, which have important implications for simulating the magnitude and timing of permafrost carbon feedback and associated hydrological processes.

It is worth noting that the slow response of the permafrost thermal regime to future climate warming may exhibit substantial variability in ice-rich permafrost zones (e.g., those containing excess ground ice), largely due to the melting of massive ground ice and the vertical movement of water. These processes exert a strong influence on permafrost thaw trajectories, often leading to landscape changes such as thermokarst pond formation and surface subsidence (Westermann et al., 2016). The associated hydrological dynamics can either accelerate or delay permafrost degradation. Specifically, when meltwater from thawed ice-rich layers drains effectively, both ground subsidence and talik formation are delayed (Westermann et al., 2016). In contrast, if meltwater accumulates at the surface, it can form ponds that enhance heat transfer into the ground, thereby accelerating talik development and intensifying permafrost thaw (Jan et al., 2020). This process holds substantial potential to unlock vast stores of currently frozen organic carbon, particularly greenhouse gases like $CO_2$ and $CH_4$ stored in cold, ice-rich lowlands. As such, thermokarst-related permafrost degradation in a warming climate could significantly amplify the global permafrost carbon–climate feedback (Turetsky et al., 2015).

We agree that integrating MVPM outputs into Earth System Models (ESMs) is essential for improving global climate projections. Based on our findings and previous modeling experience, we offer the following suggestions for improving land surface models (LSMs) within ESMs:

First, enhancing lower boundary conditions is critical for accurately simulating long-term permafrost thermal dynamics. Many existing LSMs use shallow soil profiles (e.g., less than 10 m) and simplified zero-flux lower boundaries, which are inadequate for capturing deep ground thermal processes. We recommend implementing deeper soil configurations (e.g., 50–100 m) combined with geothermal heat flux boundary conditions to better represent the ground thermal regime over decadal to centennial timescales.

Second, accurately simulating permafrost degradation requires high vertical resolution and robust model initialization. This involves resolving both seasonal and long-term thermal changes across deep soil profiles with sufficient layering, and accounting for the extended thermal memory in permafrost systems (Razavi et al., 2015). We recommend increasing the number of soil layers to

improve vertical resolution and carefully initializing (longer spin up) deep soil thermal and moisture states, ideally informed by long spin-up runs or in situ observations.

Third, improving the representation of ground ice is vital. Ground ice plays a key role in controlling the thermal and hydrological regimes of permafrost regions by affecting soil thermal conductivity, heat capacity, and moisture content (Hu et al., 2023). It also significantly affects permafrost thaw trajectories, causes surface subsidence or thermokarst pond (Lee et al., 2014; Westermann et al., 2016; Sun et al., 2022). However, current LSMs often apply overly simplistic ground ice parameterizations, typically limited to near-surface layers and lacking representations of excess and segregated ice and their formation mechanisms (Lu et al., 2017). We suggest incorporating sub-grid scale representations of ground ice distribution, modeling the dynamic formation and melt of excess and segregated ice, and explicitly including thaw-induced ground subsidence and thermokarst pond processes.

Finally, MVPM outputs, which provide high-resolution, observation-constrained simulations of ground thermal dynamics, can serve as benchmarks for evaluating and calibrating LSMs across diverse regions of the QTP. Furthermore, integrating satellite remote sensing products parameter optimization can significantly improve model realism and reduce uncertainties.

The revised manuscript will include an expanded discussion on implication for future projections.

**Reference:**

Guo, D., Wang, H., and Li, D.: A projection of permafrost degradation on the Tibetan Plateau during the 21st century, 117, D05106, J. Geophys. Res.-Atmos., 117, D05106, https://doi.org/10.1029/2011JD016545, 2012

Lee, H., Swenson, S., Slater, A., and Lawrence, D.: Effects of excess ground ice on projections of permafrost in a warming climate, Environ. Res. Lett., 9, 124006, https://doi.org/10.1088/1748-9326/9/12/124006, 2014.

Lu, J., Zhang, M., Zhang, X., Pei, W., 2017. Review of the coupled hydro-thermomechanical interaction of frozen soil. J. Glaciol. Geocryol. 39 (1), 102–111.

Ni, J., Wu, T., Zhu, X., Hu, G., Zou, D., Wu, X., Li, R., Xie, C., Qiao, Y., Pang, Q., Hao, J., and Yang, C.: Simulation of the present and future projection of permafrost on the Qinghai-Tibet Plateau with statistical and machine learning models, J. Geophys. Res.-Atmos., 126, e2020JD033402, https://doi.org/10.1029/2020JD033402, 2021.

Nitzbon, J., Westermann, S., Langer, M., Martin, L. C., Strauss, J., Laboor, S., Boike, J. Fast

response of cold ice-rich permafrost in northeast Siberia to a warming climate. Nature communications, 11(1), 2201, 2020

Razavi, S., Elshorbagy, A., Wheater, H., Sauchyn, D.: Toward understanding nonstationarity in climate and hydrology through tree ring proxy records. Water Resour. Res. 51 (3), 1813–1830, 2015.

Sun, Z., Zhao, L., Hu, G., Qiao, Y., Du, E., Zou, D., Xie, C.: Modeling permafrost changes on the Qinghai-Tibetan plateau from 1966 to 2100: a case study from two boreholes along the Qinghai-Tibet engineering corridor, Permafrost and Periglac. Process., 32:156-171, https://doi.org/10.1002/ppp.2022, 2019.

Sun, Z., Zhao, L., Hu, G., Zhou, H., Liu, S., Qiao, Y., Du, E., Zou, D., and Xie, C.: Numerical simulation of thaw settlement and permafrost changes at three sites along the Qinghai-Tibet Engineering Corridor in a warming climate, Geophys. Res. Lett., 49, e2021GL097334, https://doi.org/10.1029/2021GL097334, 2022.

Turetsky, M. R., Benscoter, B., Page, S., Rein, G., Van Der Werf, G. R., Watts, A.: Global vulnerability of peatlands to fire and carbon loss. Nature Geoscience,8(1), 11-14, 2015

Westermann, S., Langer, M., Boike, J., Heikenfeld, M., Peter, M., Etzelmüller, B., and Krinner, G.: Simulating the thermal regime and thaw processes of ice-rich permafrost ground with the land-surface model CryoGrid 3, Geosci. Model Dev., 9, 523–546, https://doi.org/10.5194/gmd-9-523-2016, 2016.

Zhao, J., Zhao, L., Sun, Z., Niu, F., Hu, G., Zou, D., Liu, G., Du, E., Wang, C., Wang, L., Qiao, Y., Shi, J., Zhang, Y., Gao, J., Wang, Y., Li, Y., Yu, W., Zhou, H., Xing, Z., Xiao, M., Yin, L., and Wang, S.: Simulating the current and future northern limit of permafrost on the Qinghai–Tibet Plateau, The Cryosphere, 16, 4823–4846, https://doi.org/10.5194/tc-16-4823-2022, 2022.

6. Spatial resolution

The use of 1km spatial resolution may not adequately capture topographic effects (e.g., slope, aspect), which can critically influence local permafrost dynamics. The paper should be further justifying the adequacy of this resolution, particularly in complex mountainous terrain and variables snow cover (controlled by wind redistribution, slope, aspect)

**Response:**

We appreciate the reviewer's insightful comment regarding the potential limitations of using a 1 km spatial resolution in capturing fine-scale topographic effects on permafrost thermal regime.

Compared with large-scale studies employing coarser resolutions (e.g., 10 km in Zhang et al., 2022; ~62km in Guo et al., 2012) or point-scale simulations (e.g., Sun et al., 2019, 2022), our use of a 1 km resolution represents a meaningful improvement in balancing spatial coverage with

topographic sensitivity. Specifically, our simulations at this resolution successfully capture two important characteristics of the regional permafrost thermal regime in the West Kunlun region: (a) the differentiation of ALT among various soil stratigraphic units (i.e., glarosion, aeolian, lacustrine, colluvial), and (b) the elevation-dependent spatial variability of ground temperatures.

Nonetheless, we fully acknowledge that in complex mountainous terrain, a grid cell size of 1 km is insufficient to resolve micro-topographic features such as slope, aspect, and wind-driven snow redistribution—factors that significantly influence local permafrost hydrothermal conditions. As such, our modeling scheme should be regarded as providing a first-order approximation of permafrost thermal distribution in mountainous areas, rather than capturing detailed topographic controls necessary for slope-scale permafrost assessments.

Despite this limitation, we believe our simulations effectively capture the broader spatial patterns of the permafrost thermal regime across the West Kunlun region. This conclusion is supported by a multi-faceted validation approach, which combines in situ measurements from the regional permafrost monitoring network with previously published permafrost distribution maps. These validations lend reasonable confidence to our simulation results. More importantly, although our current model resolution is limited by the available satellite-derived LST products, the approach produces promising results for the permafrost thermal state and ALT distribution in the West Kunlun region. It provides valuable insights into the regional permafrost thermal regime in remote and data-sparse areas of the western QTP, where observations are difficult. Future improvements will depend on incorporating higher-resolution remote sensing data that better capture topographic and snow cover variability.

The revised manuscript will include an expanded discussion on limitations of model resolution in complex mountainous terrain.

**Reference:**

Guo, D., Wang, H., and Li, D.: A projection of permafrost degradation on the Tibetan Plateau during the 21st century, 117, D05106, J. Geophys. Res.-Atmos., 117, D05106, https://doi.org/10.1029/2011JD016545, 2012

Sun, Z., Zhao, L., Hu, G., Qiao, Y., Du, E., Zou, D., Xie, C.: Modeling permafrost changes on the Qinghai-Tibetan plateau from 1966 to 2100: a case study from two boreholes along the Qinghai-Tibet engineering corridor, Permafrost and Periglac. Process., 32:156-171, https://doi.org/10.1002/ppp.2022, 2019.

Sun, Z., Zhao, L., Hu, G., Zhou, H., Liu, S., Qiao, Y., Du, E., Zou, D., and Xie, C.: Numerical

simulation of thaw settlement and permafrost changes at three sites along the Qinghai-Tibet Engineering Corridor in a warming climate, Geophys. Res. Lett., 49, e2021GL097334, https://doi.org/10.1029/2021GL097334, 2022.

Zhang, G., Nan, Z., Hu, N., Yin, Z., and Zhao, L., Cheng, G., and Mu, C.: Qinghai-Tibet Plateau permafrost at risk in the late 21st Century, Earth's Future, 10, e2022EF002652, https://doi.org/10.1029/2022EF002652, 2022.

7. Figure 5 and Figure 10

These figures display a gridded pattern at approximately 25km resolution, which appears inconsistent with the stated 1km model resolution. the source of this pattern should be clarified and discussed. Is it an artefact of the clustering used? This deserves explanation, especially given the emphasis on fine-scale modelling.

**Response:**

We believe the gridded pattern occurred in the figures, particularly Figure 5 is not a result of the clustering approaches used in our modeling. Figure 5 presents maps of decadal anomalous LST patterns over the West Kunlun permafrost region, derived from our reconstructed LST dataset spanning 1980 to 2022. Notably, no clustering algorithms were applied in this part of the analysis. The spatial distribution maps in Figures 6, 7, and 9 were indeed produced using clustering algorithms, but the gridded pattern does not appear, further suggesting that it is not an inherent artifact of the clustering process.

The apparent gridded artifacts are primarily due to uncertainties introduced during the resampling process of input datasets used for LST reconstruction. Specifically, several key input variables fed into the machine learning models,such as skin temperature (ST, $0.312° \times 0.312°$), fractional cold cover (CFC, $0.25° \times 0.25°$), surface net radiation budget (SRB, $0.25° \times 0.25°$), and leaf area index (LAI, $0.05° \times 0.05°$) have coarser spatial resolutions than the 1 km target resolution of our model. To match the modeling grid, we resampled all input variables to a 1 km $\times$ 1 km resolution using the nearest neighbor method. This resampling process—from coarse-scale model data to higher resolution is subject to uncertainties. In particular, the coarse resolution of inputs like surface temperature (ST), which is approximately 25–30 km at this latitude, introduced visible gridded patterns in the outputs, even though the resampling enabled analysis at our target 1 km resolution. We acknowledge this as a limitation and have added a discussion to the revised manuscript to clarify this issue during model forcing reconstruction.

**Minor Comments**

\* Line 283: Typo: "account for only 28.02% of the total model grid cells, **remarkable** reducing computation time"

**Response:**

Yes, we have corrected this typo in the revised manuscript. The revised sentence now reads:
"account for only 28.02% of the total model grid cells, remarkably reducing computation time."

\* Line 300: Typo: Section title should read "3.3 Filed investigation and borehole monitoring datasets"

**Response:**

Yeah, thanks, we have made the revision. We have also carefully checked and corrected similar typos throughout the text.

\*Figure 7: Missing units on the lagend: overall presentation could be improved.

**Response:**

In the revised manuscript, we have added the units to the legend in Figure 7

[Figure]

*Line 538: Add space in "with 74.20%" to read "with 74.20%"

**Response:**

We have added space in the revised manuscript.

* Figure 11: Legend and labelling could be enhanced for readability.

**Response:**

We have revised Figure 11 to enhance readability by improving the resolution of the legend and increasing the font size of the labels for better clarity. We hope these improvements make the figure easier to interpret.

[Figure]

1:Glarosion; 2:Alluvial plain; 3:Aeolian; 5:Colluvial valley; 6:Lacustrine plain

* Line 600: Discussion: "Most previous evaluations indicated that soil temperature products derived from atmospheric circulation model or ESMs, which typically have coarse resolutions (~300km) …" Consider replacing "ESMs" with "GCMs" for clarity. Or revise to more moderate number typical of historical forcing datasets such as ERA5 (25km) or ERA5-Land (9km) as you do not use GCMs in this study.

**Response:**

Yeah, exactly. We agree that "GCMs" would be a more accurate term than "ESMs" in this context and have revised the sentence accordingly. The revised sentence now reads:

"Most previous evaluations indicated that soil temperature products derived from atmospheric reanalysis datasets such as ERA5 (~25 km) or ERA5-Land (~9 km), which typically have coarser spatial resolutions"

* Line 613: Clarify what is mean by "compared to in situ measurements, we found a slight cold bias in our reconstructed LST series, averaging approximately -0.80°C"-is this computed over the entire period? Please specify the period.

**Response:**

The reconstructed LST was evaluated against monthly in situ observations from the TSH AWS for the period 2016–2018. The average bias across all months during this period was approximately –0.80°C. A systematic cold bias is evident in the mean annual cycle of LST, particularly during the summer months of July, August, and September, as shown in Figure 1.

* Line 858: Typo: "experiencing recover or degradation" should be revised to "experiencing recovery or degradation."

**Response:**

The revision has been made in the revised manuscript.

* Line 185: How and what in situ measurement integrated? Please add additional clarification here. (this dataset was created by integrating in situ observations with satellite-based LST from the Moderate Resolution Imaging Spectroradiometer (MODIS).)

**Response:**

To estimate daily mean LST values from Aqua and Terra's instantaneous daytime and nighttime observations, Zou et al. (2014, 2017) developed a multi-stepwise statistical model based on GST data from AWS located in typical permafrost regions of the central QTP. These relationships capture actual climate conditions corresponding to satellite overpass times. The resulting empirical correction model was then applied across the entire QTP permafrost zone to upscale and generate reliable LST estimates. We therefore believe that the modified MODIS LST dataset developed by Zou et al. (2017) partially accounts for the influence of surface conditions—including snow cover, vegetation, and cloud cover—through the use of a cloud-gap filling algorithm and the incorporation of AWS observations from representative permafrost regions in the central QTP. Please see details in response of secondary question "Use of MODIS LST as forcing data"

---

## Author Response (AR1)

Dear Editors:

We would like to express our sincere appreciation to both reviewers for their thoughtful evaluations and constructive feedback on our manuscript. Below is a summary of the main revisions made in the revised version:

1. **MVPM Improvements**: We have clarified the specific improvements and innovations of MVPM compared to existing models (e.g., GIPL, Noah-MP, CLM, CryoGrid) in **"Section 1 Introduction"**, and provided further elaboration in **"Section 3.1 The Moving-Grid Permafrost Model"**. **"Section 5.3 Comparison with previous studies"**, now includes suggestions for integrating MVPM outputs into Earth System Models to improve global climate projections. **Model forcing limitations:** To address concerns regarding the use of remote sensing-based LST as model forcing, we provided a detailed description of preprocessing steps in **"Section 3.2.1 Model forcing"**, including how we mitigated uncertainties related to cloud cover, snow, and vegetation. In **"Section 4.1.1 Comparison to in situ data"**, we added a comparison with ERA5-Land skin temperature, original LST to highlight the improvements achieved by our reconstructed LST. **"Section 5 Discussion"** now summarizes the applicability and limitations of the reconstructed LST, as well as uncertainties introduced by the model forcing data 1 km grid resolution, which cannot fully capture micro-topographic effects. **"Section 6 Current model shortcomings and future improvements"** has been expanded to further address the uncertainties associated with using LST as an upper boundary condition and to outline future directions for refinement. **Model validation and uncertainty**: A sensitivity analysis was conducted to assess key parameters such as soil thermal conductivity, heat capacity, water/ice content, initial temperature profile, and upper boundary temperature. Results are summarized and discussed in **"Section 6 Current model shortcomings and future improvements"**.

2. **Soil stratigraphy representation**: We added more detail on soil stratigraphy in Section **"3.2.2 Ground thermal properties"**, and discussed related uncertainties in **"Section 6 Current model shortcomings and future improvements"**

3. **Permafrost thermal stability and implications for future projections**: **"Section 5 Discussion"** now includes a clearer and more deep discussion of permafrost thermal stability and warming trends, including subsurface processes such as latent heat effects and talik formation, along with the implications for future projections and potential impacts on the global permafrost carbon–climate feedback.

4. **Language and technical corrections**: We corrected all typographical errors and thoroughly reviewed the manuscript to ensure technical accuracy throughout. Unclear figures have been replotted to improve readability. We also carefully checked for any missing co-authors and verified

all affiliations. An additional affiliation was added for co-author Wenxin Zhang: *School of Geographical and Earth Sciences, University of Glasgow, Glasgow, G12 8QQ, UK.*

We look forward to hearing from you!

Best regards,

Prof. Dr. Lin Zhao

On behalf of all co-authors

Below is our detailed response to each comment, outlining how we have addressed the reviewer's feedback in the revised manuscript. For clarity, the original reviewer comments are presented in black font, our responses are in blue font, and the corresponding revisions in the manuscript are highlighted in red. References cited in our responses are listed at the end of this document.

**Response to Referee#1 egusphere-2024-3956**

**General Comments:**

1. In terms of the transient numerical model, First, I would like to understand the distinct advantages of the Moving-Grid Permafrost Model (MVPM) compared to existing models, like GIPL, Noah-MP, CLM, and CryoGrid. The authors state that the MVPM accounts for the thermal properties between frozen and thawed soil, unfrozen water content in frozen soil, ground ice distribution, thaw settlement of the ground surface, and geothermal heat flux to address model deficiencies. However, these physical processes and parameterization schemes are also implemented in other land surface models. Could the authors clarify what specific improvements or innovations MVPM provides over these existing models? Second, how dose the MVPM model deal with the water balance in the soil domain? Which scheme does it use for dynamics of soil water contents? No flow (constant water plus ice contents)? Bucket scheme? Richards equation? Third, although the snow cover on the Tibetan Plateau is relatively thin, it can significantly affect the hydrothermal state of the permafrost beneath it. dose the MVPM model consider the insulation and cooling effect of snow cover? Fourth, dose this study activate the ground subsidence module of MVPM? This means that does this study consider the existence of excess ice?

**Response:**

The specific improvements and innovations of MVPM are listed below:

**1. Moving-grid (Lagrangian) scheme:** The key innovation of MVPM lies in its dynamic vertical discretization. Unlike conventional models with fixed soil layers, MVPM employs a moving-grid (Lagrangian) approach that actively tracks freeze–thaw fronts. This design minimizes numerical diffusion and improves the simulation of latent heat effects and delayed thermal responses in deep permafrost layers, particularly under transient climate forcing. Such a scheme is not commonly found in widely used models such as CLM, GIPL, or CryoGrid.

**2. Integrated and flexible deep soil process representation**: MVPM integrates all major thermal processes—unfrozen water content, variations in soil thermal properties, geothermal heat flux, and the vertical heterogeneity of ground ice—within a computationally efficient moving-grid framework. Its flexible vertical discretization enables more accurate representation of deep soil stratigraphy and ice distribution, significantly improving long-term simulations of permafrost thermal dynamics. In contrast, many existing Land Surface Models (e.g., CLM, Noah-MP) use

simplified layering schemes and often neglect geothermal heat flux, leading to reduced accuracy in deep permafrost modeling. This limitation is particularly critical on the Qinghai−Tibet Plateau, which has a relatively high geothermal gradient compared to Arctic and sub-Arctic regions, resulting in a longer permafrost response time and slower ground temperature increase (Jin et al., 2011; Zou et al., 2017).

3. **Improved treatment of phase change**: While most LSMs assume a sharp phase transition at 0 °C, MVPM applies the apparent heat capacity method to simulate gradual phase change over a temperature range, based on observed unfrozen water–temperature relationships from boreholes (−0.3 °C to 0 °C). This approach is more consistent with field observations from the QTP and enhances the realism of freeze–thaw dynamics in the active layer.

4. **Thaw settlement module**: MVPM includes a dedicated thaw settlement module that simulates surface subsidence and landscape changes driven by the melting of excess ground ice—key processes for understanding thermokarst development and permafrost degradation. This process-based capability is rarely represented in existing Land Surface Models.

5. **Field observations calibrated and validated**: MVPM has been rigorously calibrated using detailed field-based thermophysical measurements and soil thermal properties, and has demonstrated strong performance in reproducing observed soil temperature profiles, ALT, and long-term permafrost dynamics across multiple sites along the Qinghai−Tibet Highway (QTH) (Sun et al., 2019, 2022, 2023; Zhao et al., 2022).

These advancements make MVPM particularly suitable for assessing permafrost thermal regimes and their long-term evolution under climate change, especially in the complex and heterogeneous terrain of high-altitude permafrost regions like the Qinghai–Tibet Plateau.

Second, how does the MVPM model deal with the water balance in the soil domain? Which scheme does it use for dynamics of soil water contents? No flow (constant water plus ice contents)? Bucket scheme? Richards equation?

We used a no-flow (static) water balance scheme: the total water + ice content in each soil layer remains constant (set from field estimates for each soil type), and moisture moves only through phase change. This simplification is common in permafrost models for regions with minimal infiltration, and it focuses on thermal effects of in-situ freeze/thaw. **We have clarified this in the section 3.1, line 174 to line 175.**

Third, although the snow cover on the Tibetan Plateau is relatively thin, it can significantly affect the hydrothermal state of the permafrost beneath it. dose the MVPM model consider the insulation and cooling effect of snow cover?

In the current simulation, the MVPM model does not explicitly incorporate snow cover effects. While snow can influence permafrost hydrothermal conditions, we decide to omit it due to the following reasons:

First, near-surface ground temperatures are subject to short-term fluctuations driven by air temperature, snow, and vegetation. However, the ground acts as a natural low-pass filter, damping high-frequency signals with depth. At the depth of zero annual amplitude (ZAA), temperature trends reflect long-term climatic patterns rather than short-term surface variability (Jin et al., 2011; Dobiński et al., 2022).

Second, snow cover across the QTP is generally sparse and short-lived due to strong solar radiation and wind. Outside alpine regions above 6,000 m, snow depth seldom exceeds 3 cm and usually melts within a day (Che et al., 2008; Zou et al., 2017).

Third, we used the modified MODIS LST dataset developed by Zou et al. (2017), which includes cloud-gap filling and calibration with ground-based AWS observations to better account for surface heterogeneity. Validation at three typical permafrost sites with distinct surface types—alpine steppe, alpine meadow, and alpine desert—showed strong agreement between modeled and observed LST, with $R^2$ values ranging from 0.91 to 0.93 and RMSE values between 2.28 °C and 2.42 °C. Further evaluation at the TSH AWS site in the WKL region during the 2016–2018 observation period confirmed the product's reliability, yielding an $R^2$ greater than 0.90 and an RMSE of 2.09 °C. These results demonstrate the dataset's effectiveness in capturing spatial variations in LST across the QTP. In our study, we further improved this product by applying a machine learning approach to reconstruct pre-2003 LST using multiple data sources. The reconstructed LST performed slightly better than the original product ($R^2 > 0.95$, MAE = 1.29–1.50 °C, RMSE = 1.62–1.91 °C) and showed substantial improvement over ERA5-Land LST.

In summary, while transient snow may introduce short-term surface cooling due to high albedo and rapid melt (Zhang et al., 2005), its overall influence is minor at the decadal simulation scale adopted here. Snow effects are more critical in longer-term (centennial to millennial) modeling scenarios.

Fourth, dose this study activate the ground subsidence module of MVPM? This means that does this study consider the existence of excess ice?

No, in this study, we did not activate the ground subsidence module of the MVPM model. Our focus in this work was primarily on the permafrost thermal regime evolution under climate change. Additionally, turn off the ground subsidence module helped improve modeling computational cost. Therefore, the existence of excess ice and its potential effects on ground subsidence were not considered in the present study.

To address limitations in existing models, we developed the Moving-Grid Permafrost Model (MVPM; Sun et al., 2019, 2022) to enhance the simulation of subsurface thermal dynamics in permafrost regions. Unlike conventional LSMs with shallow or fixed-depth soil grids, MVPM adopts a flexible, moving vertical structure that better resolves deep soil stratification and spatial variability in ground ice content. The model improves the simulation of freeze–thaw processes by adopting the apparent heat capacity method, which more realistically captures gradual phase transitions, consistent with field observations on the QTP. MVPM also explicitly incorporates geothermal heat flux as the lower boundary condition, which often neglected in many LSMs enhancing the accuracy of long-term ground temperature modeling. Moreover, the model includes a thaw settlement module, rarely represented in other models, that simulates surface subsidence and landscape changes caused by the melt of excess ground ice. These processes are crucial for shaping permafrost thaw trajectories and driving thermokarst development, with the potential to release vast amounts of frozen organic carbon from cold, ice-rich lowlands, thereby amplifying the global permafrost carbon–climate feedback (Turetsky et al., 2015; Westermann et al., 2016; Jan et al., 2020). Together, these advances enable MVPM to effectively capture both the attenuation and time lag of thermal signals in deep permafrost, making it well-suited for modeling permafrost thermal regimes under a changing climate. In our previous work, MVPM has been successfully applied to simulate heat transfer processes at multiple borehole sites and regions along the QTH. The model has demonstrated sufficient accuracy in reproducing both the annual dynamics of active layer thawing and refreezing, as well as long-term ground temperature evolution, when compared with multi-depth soil temperature observations and active layer thickness measurements (Sun et al., 2019, 2022, 2023; Zhao et al., 2022).

We provided further elaboration in **"Section 3.1 The Moving-Grid Permafrost Model", Line 168 to line 179:**

The Move-Grid Permafrost Model (MVPM) is a numerical framework used to infer time series of ground temperature with the land surface as the model's upper boundary (Sun et al., 2019). Its model physics is similar to other widely employed models, such as GIPL2.0 (Dmitry et al., 2017) and CryoGrid2.0 (Westermann et al., 2013): the change of internal energy and temperature in the ground is entirely determined by Fourier's law of heat conduction, and the latent heat generated or consumed by soil freezing and thawing within a specified temperature range of −0.3 to 0 °C based on observations. Movement of water or water vapor in the ground is not included, so the soil water content can only change over time due to freezing processes. Soil temperature dynamics are simulated by numerically solving the one-dimensional nonlinear conductive heat equation using the finite difference method (Schiesser, 1991; Westermann et al., 2013; Sun et al., 2019). The latest

version of the MVPM also includes a settlement module, which was not part of our previous model configuration (Zhao et al., 2022).

**"Section 6 Current Model Shortcomings and Future Improvements"** has been expanded to further address the uncertainties associated with using LST as an upper boundary condition and to outline future directions for refinement. Line 837 to line 847:

In addition, the subsurface thermal model MVPM uses satellite-derived LST as the upper boundary condition, which does not explicitly account for snow and vegetation canopy effects, potentially introducing uncertainties in densely vegetated areas. However, in the permafrost regions of the QTP, snow cover is typically thin (~3 cm), short-lived (lasting less than a day per event), and vegetation is sparse, with less than 10% cover in the west (Wu and Zhang, 2008; Che et al., 2008; Wang et al., 2016; Zou et al., 2017; Yan et al., 2022). Under these conditions, the thermal offset between ground surface temperature (GST) and LST is minimal (Hachem et al., 2012). While thin snow cover may briefly cool the surface due to high albedo and rapid melt (Zhang et al., 2005), this effect is likely negligible over the decadal timescale of our study. Still, the model's limitations highlight the need for further validation, especially regarding hydrogeological influences on permafrost thermal regimes and improved representation of surface heterogeneity in future developments.

2. Aiming at model forcing, as the only model forcing variable, this study adopted three statistical and machine-learning approaches to extent the land surface temperature from Zou et al (2014, 2017). I was wondering why the authors selected these eight specific input variables—surface air temperature, precipitation, skin temperature, soil temperature, fractional cold over, surface radiation budget, leaf area index, and digital elevation model—for the statistical and machine learning approaches. Could the authors clarify whether including more (or fewer) variables might help to avoid issues of model underfitting or overfitting? Besides, furthermore, what it is the basis for selecting the particular datasets used for these variables? For example, the study utilizes the uppermost soil temperature from CFSR, while skin temperature is taken from ERA5-land. Given that ERA5-land also provides uppermost soil temperature data at a higher spatiotemporal resolution compared to CFSR.

**Response:**

We selected these eight specific input variables based on physical relevance, expert knowledge, and a thorough review of related published literature (Wang et al., 2022; Xu et al., 2018; Janatian et al., 2017; Yang et al., 2023), which guided us to select variables having close relationships with LST as input. Moreover, data quality and availability on QTP were even more important considerations. We selected variables that had consistent, long-term records that satisfied the requirements of our study.

Regarding model complexity and model underfitting or overfitting, we conducted preliminary

experiments and for each machine learning approach, we tested both training and validation errors across different variable combinations. For example, models using only air temperature, as shown in our previous work (Xing et al., 2023), demonstrated signs of underfitting with systematic errors. However, adding related variables significantly improved model performance according to our cross-validation tests. Thus, we believe the eight-variable configuration that closely represents LST dynamics provided the optimal balance between model performance and parsimony, performing better than simpler models with fewer variables. Conversely, whether adding additional parameters, such as wind speed, humidity, soil moisture, and snow cover, would lead to model overfitting is uncertainly at present. These additional data products exhibit considerable variability in quality across the permafrost zone of the QTP. Investigating their potential integration will be the focus of future work.

As for soil temperature selection, our choices were primarily driven by data quality assessments. While ERA5-land is a good choice that provides soil temperature at different depths at a higher spatiotemporal resolution than CFSR, we selected CFSR soil temperature data based on its better performance in preliminary validation against our long-term continuous observations in the permafrost zone on the QTP. The validation results suggested CFSR soil temperature products were closer to the observations at different depths in the permafrost zone on the QTP despite its coarser resolution (Hu et al., 2018). This explains our use of different sources for shallow soil temperature and skin temperature. This mixed-source approach allowed us to leverage the strengths of each dataset while compensating for their respective limitations in our specific study area.

In any way, we will try to do more work on the issues raised by the reviewers.

**Specific comments:**

1. Line 40: Smith et al 2022? Maybe it is a wrong reference?

**Response:**

Yeah, it was indeed a typo. The correct reference should be **Smith et al., 2022**, referring to: "Smith, S., O'Neill, H., Isaksen, K., Noetzli, J., and Romanovsky, V.: The changing thermal state of permafrost, Nat. Rev. Earth Environ., 3, 10–23, https://doi.org/10.1038/s43017-021-00240-1, 2022." We have corrected this in the revised manuscript.

2. Line 49: the reference (Zhao et al., 2019) seemed missing?

**Response:**

Here should be cited as Zhao et al., 2019a, referring to the following reference: "Zhao, L., Hu, G., Zou, D., Wu, X., Ma, L., Sun, Z., Yuan, L., Zhou, H., and Liu, S.: Permafrost Changes and Its Effects on Hydrological Processes on the Qinghai-Tibet Plateau, Bull. Chin. Acad. Sci., 34, 1233–1246, DOI: 10.16418/j.issn.1000-3045.2019.11.006, 2019a."

3.   Line 51: should the Qinghai-Tibet Highway and Railway be abbreviated as QTH? Not sure.

**Response:**

You're right. Using QTH for both the Qinghai-Tibet Highway and Railway can be confusing. After reviewing the literature, we found that QTH is commonly used for the Qinghai-Tibet Highway, while QTR refers to the Qinghai-Tibet Railway. In the revised manuscript, we have adopted these standard abbreviations (QTH and QTR) separately to ensure clarity.

4.   Line 84: "ALT" should be given tis full name, this is the first time it has been abbreviated. And "refreezing" of what?

**Response:**

Here, we have given full name for ALT (active layer thickness). I want to express that, based on our previous simulations, the MVPM can provide sufficient accuracy to capture the annual dynamics of active layer thawing and refreezing, but it seems that 'thawing' was missing. In the revised manuscript, we have corrected this in revised manuscript.

5.   Line 161: dose this study activates the settlement module?

**Response:**

No, in this study, we did not activate the ground subsidence module of the MVPM model. Please refer to the answer to the first question.

6.   Line 183: what is the surface radiation budget? Net radiation? Net shortwave radiation? Or net longwave radiation? It is not clear.

**Response:**

Net radiation, and we have corrected this in revised manuscript.

7.   Line 189: the resolution of all input data is not daily.

**Response:**

Exactly, we have corrected in revised manuscript. The text there reads as:

"Monthly averages were then calculated from the available data (which varied in temporal resolution across datasets), and missing values were filled by interpolating from nearby data."

8. Line 271: how to deal with initial water/ice content?

**Response:**

Initial water and ice content were estimated from moisture measurements of borehole cores taken from various Quaternary sediment types, including fluvioglacial, lacustrine, alluvial, and aeolian deposits. These values were further refined through model calibration to improve simulation accuracy. Volumetric ice content was highest in fluvioglacial sediments, followed by lacustrine sediments and weathered residual slide rock. Vertically, ground ice is concentrated near the permafrost table, where the active layer thickness (ALT) is typically 2–3 m. Ice content generally increases between 3–10 m depth and stabilizes below 10 m.

9. Line 274: "approximately"? it should be an exact number for the grid cell to be simulated.

**Response:**

I agree, this was not an appropriate expression and has been corrected in the revised manuscript. The text there reads as:

After excluding lake and glacier-covered areas, simulations were conducted for 47,284 grid cells.

10. Line 300: "filed investigation and borehole monitoring datasets"? I guess it is "Field".

**Response:**

Yes, exactly, we made a typo. Thank you so much for pointing it out. We have also carefully checked and corrected similar typographical errors throughout the text.

11. Line 430: "Grey Shading"? only saw the grey line.

**Response:**

Yes, it should be 'grey line' instead of 'Grey Shading.' This has been corrected.

12. Section 4.2.4: the author states, "permafrost area in the West Kunlun kept stable from 1980 to 1999, decrease in the 2000s, while increase between 2010 and 2022." However, the MVPM model is just forced by land surface temperature, while showed an increasing trend between 1980 and 2022 (Figure 4). Could the author explain why permafrost area increase between 2010 and 2022?

**Response:**

This apparent contradiction can be attributed to two key factors:

First, although the regional average LST shows an overall warming trend from 1980 to 2022, substantial interannual and spatial variability exists within the study area. Some localized regions experienced short-term cooling phases within the broader warming trend, which may have supported new permafrost formation or local expansion.

Second, the thermal response of soil layers to surface warming occurs at different rates and includes significant time lags—especially in deeper, ice-rich layers. The high thermal inertia of permafrost delays its response to surface warming by years or even decades. As a result, the simulated slight increase in permafrost extent between 2010 and 2022 likely reflects a delayed thermal response to earlier, cooler conditions rather than a contradiction of recent warming.

These findings underscore the non-linear and lagged nature of permafrost response to climate change in the WKL region.

We added related clarify Permafrost thermal Stability: **"Section 5 Discussion" in our revised manuscript,** line 665 to line 670:

Interestingly, while regional average LST showed a steady increase from 1980 to 2022, considerable interannual and spatial variability was simulated. We hypothesize that intermittent cooling episodes may have triggered the formation or re-expansion of permafrost in certain areas through delayed responses, a view supported by our simulation, which showed a slight increase in permafrost extent between 2010 and 2022 despite continued warming.

13. Line 601-609: so how about the reanalysis data (like Chinese meteorological forcing datasets, ERA5 land)? Compared with the forcing data from ESMs, the spatiotemporal resolution of them is better.

**Response:**

Exactly. While reanalysis and assimilated data products offer higher spatiotemporal resolution than ESM outputs, their accuracy in the permafrost regions of the QTP remains limited due to sparse observational constraints (Jiao et al., 2023). For instance, Hu et al. (2018) found that GLDAS-NOAH and ERA-Interim performed poorly in the QTP when compared with soil temperature observations. Similarly, Yang et al. (2020) reported that reanalysis datasets such as CFSv2, ERA-Interim, GLDAS-Noah, and ERA5 can capture temporal trends but significantly underestimate soil temperatures during thawing periods.

For Chinese reanalysis specifically, Hu et al. (2024) showed that although CLDAS performs well over most of China, it exhibits large errors in the QTP permafrost zone (bias = 2.09°C, MAE = 3.64°C, RMSE = 4.67°C), largely due to limited observational input and inadequate representation of permafrost surface conditions.

Therefore, despite their resolution advantages, reanalysis products still face major limitations in permafrost modeling over the QTP.

In **"Section 5 Discussion"** of the revised manuscript, we have supplemented information about current reanalysis or assimilated soil temperature products and their application in the permafrost region of the QTP The text there reads as:

Previous studies have shown that currently widely reanalysis or assimilated soil temperature products, such as ERA-Interim (0.125° × 0.125°), ERA5-Land (0.1° × 0.1°) and CLDAS (0.0625° × 0.0625°) exhibit substantial uncertainties when applied to the QTP, particularly in permafrost regions (Hu et al., 2019; Qing et al., 2020; Yang et al., 2020).

14. Could the author explain how this study can be extended to be the future projections? Due to its inability to obtain the land surface temperature with higher resolution from remote sensing data in the future, how to diagnose the future condition of permafrost in West Kunlun.

**Response:**

Near-surface ground temperature closely follows seasonal air temperature variations, but these fluctuations diminish with depth and become negligible at the zero annual amplitude (ZAA) depth— typically 10–20 m in the permafrost zones of the QTP (Lunardini et al., 1995; Jin et al., 2011; Dobiński et al., 2022). At this depth, seasonal signals vanish, and ground temperatures reflect long-term climate trends rather than short-term variability (Smith and Riseborough, 1983; Buteau et al., 2004). This makes ZAA temperatures a robust indicator for modeling permafrost response to climate change, even without high-resolution remote sensing data.

To evaluate future permafrost dynamics, ground temperature profiles can be modeled under scenarios of linearly increasing air temperature. The IPCC Sixth Assessment Report (WG1 AR6) projects mean annual air temperature increases over the QTP from 1995–2014 to 2081–2100 of 0.013°C yr⁻¹ (RCP2.6), 0.028°C yr⁻¹ (RCP4.5), and 0.060°C yr⁻¹ (RCP8.5), based on the CMIP5 multi-model ensemble (21–29 models). CMIP6 projections under SSPs estimate warming rates of 0.017°C yr⁻¹ (SSP1-2.6), 0.032°C yr⁻¹ (SSP2-4.5), and 0.064°C yr⁻¹ (SSP5-8.5), based on 31–34 model outputs. A similar projection approach can be found in our following published literature: Hu et al. (2015), Li et al. (1999), Sun et al. (2020), and Zhao et al. (2022).

**Response to Referee#2 egusphere-2024-3956**

**Major Comments:**

1. Model validation and uncertainties

The authors report high model accuracy (±0.25 °C for ground temperature and ±0.25 m for active layer thickness). However, the discussion of model uncertainties could be expanded. Key areas for improvement include:

\* a sensitivity analysis of key model parameter such as soil thermal properties and initial boundary conditions to assess their impact on the results.

\* a deeper discussion on the limitations of the forcing datasets, particularly the machine-learning-based reconstruction of LST prior to 2003 and impact of the cold bias of 0.8degC (Compared to in situ measurements, we found a slight cold bias in our reconstructed LST series, averaging approximately -0.80°C)

**Response:**

A sensitivity analysis was conducted to assess the key parameters contributing to uncertainties in our modeling results, with findings summarized and discussed in **"Section 6 Current model shortcomings and future improvements",** Line 793 to line 811:

To quantify model parameter uncertainty, we conducted a one-at-a-time sensitivity analysis (Figure 12) using three representative boreholes located in stable permafrost, unstable permafrost, and seasonally frozen ground (see Table 4). Key model parameters were perturbed by ±10% to evaluate their effects on permafrost thermal regime (MAGT15 m and ALT). The result shown that among all parameters, upper boundary temperature (e.g., surface forcing) exerted the strongest influence on MAGT15 m, though the absolute impact was modest, around ±0.15 °C in seasonally frozen ground and ≤±0.1 °C in permafrost areas. ALT showed similarly limited sensitivity, varying by ~±0.1 m in stable permafrost and ±0.05 m in unstable zones. Soil thermal conductivity and water/ice content had a more pronounced effect on ALT, particularly in unstable permafrost, where a 10% change could lead to a ±0.10m~ ±0.15 m variation. In contrast, soil heat capacity had minimal influence on both MAGT and ALT.

The above analysis indicates that the model demonstrates robustness to parameterization uncertainties and that uncertainties associated with stratigraphy have a limited effect on overall performance. Although stratigraphic classification and spatial variability inevitably introduce some degree of uncertainty, our approach is well supported by field measurements and observed thermal properties. Despite these limitations, we are confident that the model accurately represents

the key thermal characteristics of each sediment class —key factors for simulating permafrost dynamics. Continued improvements in subsurface datasets, particularly in permafrost regions, will be essential for improving model performance in future applications.

**Table S1. Information on three representative borehole sites used for one-at-a-time sensitivity analysis**

| Borehole | Description |
|---|---|
| ZK30 | The borehole reaches a depth of 15 m, with the ground primarily composed of fine sand and silty sand. The MAGT is -1.66 °C, and the ALT is 2.4 m, classifying the site as stable permafrost. |
| ZK12 | The borehole has a drilling depth of 13.5 m, with a vegetation-free surface. The core consists primarily of Fluvial sand and sand. Frozen soil was first encountered at a depth of 4.9 m, where small ice crystals are evenly distributed within a granular soil structure. Below 5.5 m, the frozen layer disappears, accompanied by a noticeable increase in ground temperature. The 4.9–5.5 m interval represents a transition zone, and the site is classified as unstable permafrost. |
| ZK13 | No frozen soil was encountered during the drilling process, and the site is classified as seasonally frozen ground |

Note: This information is compiled from Li et al. (2012) and Zhao et al. (2019).

[Figure]

**Figure S1. One-at-a-time sensitivity analysis showing the effects of ±10% variation in individual model parameters, e.g., soil thermal conductivity, heat capacity, water/ice content, initial temperature, and upper boundary temperature—on (top row) mean annual ground temperature (MAGT) at 15 m depth and (bottom row) active layer thickness (ALT), across three ground conditions: stable permafrost (left), unstable permafrost (middle), and seasonally frozen ground (right).**

\* The effect of snow cover on LST reconstruction and subsurface thermal dynamics is not sufficiently addressed. How are snow-insulated period accounted for? Is there any bias introduced by could-covered or snow-covered days in the satellite-derived LST? (linked to point 2)

**Response:**

**Firstly,** in the vast permafrost zone of the QTP, strong solar radiation and wind lead to rare, thin snow cover (~3 cm) that typically persists for less than a day per snowfall event (Wu and Zhang, 2008; Che et al., 2008; Zou et al., 2017; Yan et al., 2022). Vegetation is sparse, dominated by dwarf alpine grassland, with cover often below 10% in the western QTP (Wang et al., 2016). Under these conditions, the average thermal offset between ground surface temperature (GST) and LST is minimal (Hachem et al., 2012).

**Secondly,** the MVPM model uses LST as the upper boundary forcing and employs a modified MODIS LST product developed by Zou et al. (2017, 2014). This product incorporates cloud-gap filling and calibration with ground-based AWS observations at three representative sites—alpine steppe (Xidatan), alpine meadow (Tangula), and alpine desert (Wudaoliang)—in the central QTP permafrost region, to better account for surface heterogeneity. Model validation at these sites showed strong performance, with $R^2$ values ranging from 0.91 to 0.93, mean absolute error (MAE) from 2.28 to 2.42°C, and root mean square error (RMSE) from 2.96 to 3.05°C. In our study, this product was further improved using a machine learning approach to reconstruct pre-2003 LST by integrating multiple data sources. The reconstructed LST slightly outperformed the original product ($R^2 > 0.95$, MAE = 1.29–1.50°C, RMSE = 1.62–1.91°C), and showed substantial improvement over ERA5-Land LST (see also our response to the second question regarding MODIS LST as forcing data).

Our model simulations reliably reproduce mean annual ground temperature, ALT, and permafrost distribution across different time periods. While thin snow cover may temporarily cool the ground due to high albedo and rapid melt (Zhang et al., 2005), its short duration suggests minimal impact at the decadal time scales of our simulation. Snow cover effects are more relevant for centennial to millennial time-scale modeling

2. Use of MODIS LST as forcing data

While MODIS LST provides high spatial resolution and extensive temporal coverage, it poses several challenges when used to force model models:

* MODIS measures skin temperature rather than subsurface ground temperature, which can differ significantly—especially under snow cover or vegetation (vegetation is acknowledged in this study)

* Snow cover introduces thermal insulation, decoupling surface LST from the subsurface thermal regime (Albeit often nonexistent or thin snow cover). However, wind driven snow drift can be significant and give spatial heterogeneity at high model resolutions. The significance of these effects needs to be addressed in this study region.

* Cloud cover causes data gaps, which can lead to temporal inconsistencies or biases if gap-filling methods are not robust—particularly problematic in winter months?

* MODIS LST captures only clear-sky conditions, potentially biasing the dataset toward colder nighttime or warmer daytime extremes, depending on retrieval timing.

The paper should better articulate how these limitations are mitigated in the LST reconstruction, and what implications they have for subsurface heat fluxes and permafrost thermal state. A comparison with measured air-temperature or reanalysis-based forcing would also help.

**Response:**

Surface (skin) temperature was used as the upper boundary forcing in our model, where we applied machine learning techniques to reconstruct remote-sensing-based LST. We acknowledge the reviewer's concern: MODIS LST reflects skin temperature (e.g., canopy or snow surface), not ground surface temperature (GST), and is subject to limitations such as thermal insulation from snow cover, data gaps due to cloud cover, and clear-sky sampling bias. To address these issues, we implemented several preprocessing and correction steps:

**First**, we used a modified MODIS LST product since 2003 developed by Zou et al. (2017), based on the MOD11A2 (Terra) and MYD11A2 (Aqua) Collection 6 datasets, which provide twice-daily observations. Gaps caused by cloud cover and other factors were filled using the Harmonic Analysis of Time Series (HANTS) method (Xu et al., 2013; Zou et al., 2017), effectively reducing cold bias and smoothing the time series, particularly over the QTP.

**Second**, daily mean LST values were estimated from MODIS instantaneous daytime and nighttime data using multi-step statistical models calibrated against GST data from three AWS in central QTP permafrost regions (Zou et al., 2014, 2017). This empirical correction model was applied across the QTP, with strong validation at three representative sites—alpine steppe (Xidatan), alpine meadow (Tangula), and alpine desert (Wudaoliang)—showing high accuracy (R² = 0.91–0.93, MAE = 2.28–2.42°C, RMSE = 2.96–3.05°C). In our work, additional validation in the West

Kunlun region using 2016–2018 data from the our established TSH AWS (81.4°E, 36.0°N, 5019 m) yielded $R^2 > 0.90$, MAE = 1.62°C, and RMSE = 2.09°C.

**Third,** in this study, the modified LST product from Zou et al. (2017) was further refined to reconstruct historical LST data prior to 2003—extending back to 1980, using machine learning models trained on eight variables from ground observations, satellite products, and reanalysis datasets. This reconstructed LST was validated using monthly in situ data from TSH AWS (2016–2018), achieving $R^2 > 0.95$, MAE = 1.29–1.50°C, and RMSE = 1.62–1.91°C. It outperformed the original MODIS LST and ERA5-Land in accuracy (Figure S2).

**Fourth**, although direct validation before 2003 was not possible due to lack of observations, we performed indirect validation. The reconstructed LST was used to drive MVPM simulations of permafrost dynamics from 1980 onward, which were compared with observed data and published permafrost distribution maps for multiple periods (Li et al., 1996; Wang et al., 2006; Cao et al., 2023; Zou et al., 2017). The model results aligned well with these independent sources. Moreover, the reconstructed LST captured the observed long-term warming trend in the West Kunlun region since the mid-1980s, with accelerated warming in the last decade, consistent with regional studies (Jin et al., 2011; You et al., 2021; Yao et al., 2019; Li et al., 2024).

In summary, while we acknowledge the limitations of our model forcing and the lack of direct historical data, the multi-step correction, machine learning reconstruction, and multiple validation strategies collectively provide strong support for the reliability of the LST dataset used in this study.

In revised manuscript, we added a detailed description of preprocessing steps to mitigate related uncertainties in **"Section 3.2.1 Model forcing"**, line 182 to line 192:

We used a modified MODIS LST product developed by Zou et al. (2014, 2017), which partially accounts for surface influences such as snow cover, vegetation, and cloud presence through a cloud-gap filling algorithm and calibration with AWS observations from three representative permafrost regions with distinct surface types—alpine steppe, alpine meadow, and alpine desert—in the central QTP. Validation showed strong agreement between the modeled and observed LST, with $R^2$ values ranging from 0.91 to 0.93 and RMSE values around 3 °C. Further evaluation at the TSH AWS site in the WKL region during the 2016–2018 observation period confirmed the product's reliability, with an $R^2$ greater than 0.90 and an RMSE of 2.09 °C, demonstrating its effectiveness in capturing spatial variations in LST across the QTP.

In **"Section 4.1.1 Comparison to in situ data"**, we added a comparison with ERA5-Land skin temperature to highlight the improvements achieved by our reconstructed LST.

[Figure]

**Figure S2. Monthly average LST at the TSH AWS from 2016 to 2018, including reanalysis-derived LST (ERA5-Land), modified satellite-derived LST (LST_Zou), estimates from three machine learning models (LR, MLR, and RFR), and in situ observations.**

**In Section 5 Discussion**, we summarized the applicability and limitations of the reconstructed LST, line 583 to line 623:

In contrast, satellite remote sensing products such as MODIS LST offer a long-term temporal coverage and broader regional coverage with higher spatial resolution. However, MODIS LST has several limitations when applied to permafrost modeling. It measures skin temperature rather than ground surface temperature, often reflecting vegetation canopy or snow temperature. Additionally, snow cover introduces thermal insulation effects, cloud cover creates data gaps, and only clear-sky conditions are captured. To address these challenges, we used a modified LST product developed by Zou et al. (2017), which incorporates cloud-gap filling and ground-based AWS observations to account for surface heterogeneity. Validation at three typical permafrost sites in central permafrost zone as well as WKL region showed strong performance. In our study, this product was further enhanced using machine learning to reconstruct pre-2003 LST by integrating multiple data sources. The reconstructed LST performed slightly better than the original product ($R^2 > 0.95$, MAE = 1.29–1.50°C, RMSE = 1.62–1.91°C), with notable improvement over ERA5-Land LST.

While direct validation of pre-2003 LST is not possible due to the lack of satellite or ground observations in the WKL region. We employed an indirect validation approach: the reconstructed LST was used to force the MVPM to simulate permafrost thermal dynamics from 1980 onward. The simulation results were evaluated against existing permafrost monitoring data and previously published permafrost distribution maps from various periods, i.e.1980s (Li et al., 1996), 2000s (Wang et al., 2006), 2010 (Cao et al., 2023), and post-2010 (Zou et al., 2017). The strong agreement

between the MVPM outputs and these independent sources supports the reliability of the pre-2003 LST reconstruction. Moreover, our analysis reveals pronounced LST warming in the WKL survey area since the mid-1980s, with accelerated warming over the last decade. This trend aligns with recent documented warming across the QTP (Jin et al., 2011; Yao et al., 2019; You et al., 2021; Li et al., 2024), providing further indirect validation of the reconstructed LST. Collectively, this multi-faceted validation approach provides reasonable confidence in our LST dataset, despite the lack of direct early-period observations. While we acknowledge this limitation, we believe our methodology offers a robust solution given the data constraints of this remote and observationally challenging region.

The above comparisons show that the reconstructed LST closely aligns with in situ data and is suitable for ground thermal modeling. However, a seasonal cold bias remains, especially in July–September (Figure 3), leading to a slight underestimation of shallow soil temperatures, resulting in a cold bias in ALT. Such bias is likely due to the sensitivity of near-surface ground temperature to seasonal forcing. Similarly, Westermann et al. (2015) found that an LST uncertainty of ±2 °C can lead to a ±3 cm uncertainty in simulated thaw depth. We conducted a sensitivity analysis (Figure 12) to evaluate the impact of uncertainties in model forcing (e.g., LST) on simulation results, and the findings confirm the model's robustness to LST biases. Moreover, since thermal signals attenuate with depth and ground temperatures at the ZAA level reflect long-term trends (Jin et al., 2011; Dobiński et al., 2022), the observed cold bias appears to be seasonal and has limited influence on long-term permafrost dynamics.

**In "Section 6 Current model shortcoming and future improvements",** has been expanded to discuss limitations of the model forcing, line 837 to line 847:

In addition, the subsurface thermal model MVPM uses satellite-derived LST as the upper boundary condition, which does not explicitly account for snow and vegetation canopy effects, potentially introducing uncertainties in densely vegetated areas. However, in the permafrost regions of the QTP, snow cover is typically thin (~3 cm), short-lived (lasting less than a day per event), and vegetation is sparse, with less than 10% cover in the west (Wu and Zhang, 2008; Che et al., 2008; Wang et al., 2016; Zou et al., 2017; Yan et al., 2022). Under these conditions, the thermal offset between ground surface temperature (GST) and LST is minimal (Hachem et al., 2012). While thin snow cover may briefly cool the surface due to high albedo and rapid melt (Zhang et al., 2005), this effect is likely negligible over the decadal timescale of our study. Still, the model's limitations highlight the need for further validation, especially regarding hydrogeological influences on permafrost thermal regimes and improved representation of surface heterogeneity in future developments.

3. Representation of soil stratigraphy

The study highlights variations in permafrost responses based on soil stratigraphy, but further clarity is needed regarding:

* The specific role of soil moisture and ice content in modulating permafrost temperature trends.

* Potential biases in stratigraphic classification and how these might affect regional variability in permafrost degradation.

* The extent to which subsurface heterogeneity is accounted for in the model.

**Response:**

In our modeling framework, we incorporated detailed thermophysical characterization of the subsurface based on measurements from 15 boreholes distributed across the WKL permafrost survey area, with depths ranging from 15 to 59 m. Core sampling, field observations, and borehole descriptions (Li et al., 2012; Zhao et al., 2019) indicate that ground ice content in the region varies between 5% and 50%, depending on the type of Quaternary sediments. Higher ice contents are typically found in fine-grained glarosional and lacustrine deposits due to the prevalence of segregation ice, while lower contents are associated with coarse-grained alluvial and colluvial materials. Vertically, ice-rich layers are consistently found near the upper boundary of permafrost, generally within the top 2–3 m. Ice content tends to slightly increase with depth between 3 m and 10 m, then remains relatively stable below 10 m (Zhao et al., 2010).

To extrapolate these site-level stratigraphic and thermophysical data spatially, we employed vector-based geomorphological classification maps of western China. Five major stratigraphic classes were identified in the WKL region: glarosional, alluvial plain, aeolian, colluvial valley, and lacustrine deposits.

Our simulation results highlight the critical role of ground ice content in controlling permafrost thermal dynamics. Modeled ground temperature and ALT vary across stratigraphic classes, with the deepest ALT observed in ice-poor alluvial sediments and the shallowest in ice-rich glarosional deposits.

During permafrost degradation, surface heat is used for two processes: (i) melting ground ice (latent heat) and (ii) warming the soil (sensible heat). The latent heat of the ice–water phase transition (334 kJ/kg) far exceeds the sensible heat needed to raise the temperature of ice or water (2 and 4.2 kJ/kg·K, respectively).

Heat distribution varies by degradation stage (Sun et al., 2023). In the early stage, low temperatures and minimal unfrozen water limit phase change, so soil warms rapidly through

sensible heat. In the zero-geothermal-gradient stage, as the entire permafrost nears the melting point, increased unfrozen water leads to greater latent heat consumption, slowing or halting further warming.

This mechanism explains our simulation results: the largest ALT increases (>0.17 m) occurred in ice-poor alluvial sediments in warmer zones, while the smallest increase (0.11 m) was found in ice-rich glarosional deposits at cold, high-elevation area. Ice-rich permafrost thus acts as a thermal buffer, delaying the downward propagation of surface warming and slowing active layer deepening despite continued atmospheric warming

While some uncertainty in stratigraphic classification and spatial representation are unavoidable, our approach is grounded in field data and measured thermal properties. Sensitivity analyses show that model outputs are robust to variation in key soil parameters.

We acknowledge the presence of fine-scale heterogeneity, especially in complex mountainous terrain, which is not fully resolved at the 1 km modeling resolution. Within-class variability may also contribute to local biases. Nonetheless, the model effectively captures the key thermal behavior of each sediment type, which is essential for simulating permafrost response to climate change. Future improvements in spatially resolved soil property datasets will be vital for enhancing model accuracy.

We have added more detail on soil stratigraphy in **Section "3.2.2 Ground Thermal Properties",** line 238 to line 246:

In our modeling framework, we incorporated detailed thermophysical characterization of the subsurface based on measurements from 15 boreholes with observations across the WKL permafrost survey area, with depths ranging from 15 to 59 m. Core samples, field observations, and borehole logs (Li et al., 2012; Zhao et al., 2019) indicate that ground ice content in the WKL region varies between 5% and 50%, depending on the type of Quaternary sediment. Higher ice contents are typically found in fine-grained glarosional and lacustrine sediments due to enhanced segregation ice formation, while coarse-grained alluvial and colluvial deposits generally show lower ice content. Vertically, ice-rich layers are consistently observed near the upper boundary of permafrost, typically between 2 and 3 m depth. Ice content tends to increase slightly between 3 and 10 m and remains relatively stable below 10 m (Zhao et al., 2010).

The discussed related soil properties uncertainties in **"Section 6 Current model shortcomings and future improvements" see details in response of "Model validation and uncertainties".**

4. Permafrost stability and warming trends

The study concludes that despite significant warming trends in LST, permafrost extent remains stable. While plausible given the thermal inertia of deep permafrost, the paper could be benefit from:

* A clearer discussion on why this stability is observed and how it compares with degradation rates reported in other high-altitude or Arctic regions.

* Consideration of potential threshold effect (e.g., rapid degradation once a critical warming threshold is exceeded)

* More discussion on subsurface processes such as latent heat effects and talik formation, which may delay degradation despite rising surface temperatures.

* The observed increase in permafrost area despite a warming trend of 0.4°C per decade is surprising and warrants closer examination and justification.

**Response:**

Permafrost thermal degradation is a complex and lagged response to climate warming, further modulated by local environmental factors such as soil type, ground ice content, geothermal heat flux, and the initial thermal state of the ground (Zhao et al., 2020; 2024; Hu et al., 2023). In response to climate change, permafrost does not degrade instantaneously but undergoes a gradual adjustment of its thermal regime over various timescales—ranging from years to centuries or even millennia (Wu et al., 2010).

On QTP, this response is particularly nuanced. Wu et al. (2010) proposed a classification of permafrost degradation stages based on ground temperature profiles, including the warming stage, the zero-geothermal-gradient stage, the talik development stage, and eventual disappearance. These thermal states reflect ongoing degradation processes since the Last Glacial Maximum (LGM), shaped by both climate history and local ground conditions.

Compared to high-latitude permafrost regions in the Arctic and sub-Arctic, permafrost on the QTP is generally warmer and occurs under a relatively higher geothermal gradient. This distinct thermal setting leads to a slower increase in ground temperature and prolongs the degradation response time despite pronounced atmospheric warming (Jin et al., 2011; Zou et al., 2017; Biskaborn et al., 2019). In contrast, Arctic permafrost tends to be colder and more sensitive to warming, resulting in faster thermal responses. These regional differences highlight the importance of accounting for reginal-specific thermal regimes when assessing the vulnerability of permafrost to climate change.

Our study found that approximately 70.98% of the permafrost in the Western Kunlun region is in the temperature-rising stage, characterized by an initial MAGT15m below –2.0 °C and ALT of less than 1.5 m. This type predominantly occurs in high-elevation zones above 4800 m a.s.l. An additional 17.58% of the permafrost is transitioning from the temperature-rising stage to the zero geothermal gradient stage. Only 11.44% is in the zero geothermal gradient stage or progressing toward talik development, which signals ongoing degradation. These permafrost types are primarily found at lower elevations (below 4800 m a.s.l.) and exhibit relatively high MAGTs (above –1 °C).

Permafrost forms when long-term heat loss at the ground surface exceeds heat input under a persistently cold climate (Wu et al., 2010). In a warming climate, sustained increases in ground surface temperature disrupt this thermal equilibrium. The active layer begins to retain more heat each year, initiating progressive warming from the surface downward and reducing the temperature gradient within the permafrost. During the early stages of warming, permafrost temperatures rise more rapidly than thawing occurs, as much of the incoming energy is consumed raising the frozen soil to its thawing point. This explains why, despite a pronounced warming trend, the areal extent of permafrost in the Western Kunlun region remained relatively stable during the simulation period.

Furthermore, while the regional average LST exhibited a consistent warming trend from 1980 to 2022, substantial interannual and spatial variability was observed. We suggest that periodic cooling events contributed to the formation or expansion of new permafrost in certain areas, producing a delayed and spatially heterogeneous response. This is supported by our simulated permafrost maps, which show a slight increase in permafrost extent between 2010 and 2022, despite overall warming during this period.

Under continued climate warming, MAGTs are projected to rise further. As heat penetrates deeper into the ground, the thermal gradient within the permafrost diminishes and eventually becomes smaller than the geothermal gradient, resulting in upward heat flow from underlying unfrozen ground. This process initiates basal thawing and leads to a gradual upward retreat of the permafrost base.

As permafrost temperatures approach 0 °C, ground ice near the permafrost table begins to melt, consuming large amounts of latent heat in a process known as the "zero curtain effect." This phase significantly slows or temporarily halts further warming, extending the duration of thermal inertia and dampening seasonal temperature variations in the upper permafrost. At the same time, geothermal heat from below is primarily consumed in thawing the permafrost from the bottom up.

The zero geothermal gradient stage represents a critical transitional phase in permafrost degradation. During this stage, nearly all surface-derived energy is expended on melting ground ice, accelerating the downward movement of the permafrost table. Once seasonal freezing no longer reaches the permafrost, a talik—a zone of unfrozen ground within permafrost—forms and expands.

Numerical simulations by Sun et al. (2019) demonstrate that talik development marks a tipping point, triggering accelerated thaw and irreversible permafrost degradation until complete loss.

A more detailed discussion of permafrost thermal stability and warming trends—incorporating subsurface processes such as latent heat effects and talik formation, along with the implications for future projections and potential feedbacks to the global carbon cycle—is provided in **Section 5: Discussion**, line 641 to line 688:

Permafrost thermal degradation is a complex and lagged response to climate warming, further modulated by local environmental factors such as soil type, ground ice content, geothermal heat flux, and the initial thermal state of the ground (Zhao et al., 2020; 2024; Hu et al., 2023). In response to climate change, permafrost does not degrade instantaneously but undergoes a gradual adjustment of its thermal regime over various timescales—ranging from years to centuries or even millennia.

On QTP, this response is particularly nuanced. Wu et al. (2010) proposed a classification of permafrost degradation stages based on ground temperature profiles, including the warming stage, the zero-geothermal-gradient stage, the talik development stage, and eventual disappearance. These thermal states reflect ongoing degradation processes since the Last Glacial Maximum (LGM), shaped by both climate history and local ground conditions.

Compared to high-latitude permafrost regions in the Arctic and sub-Arctic, permafrost on the QTP is generally warmer and occurs under a relatively higher geothermal gradient. This distinct thermal setting leads to a slower increase in ground temperature and prolongs the degradation response time despite pronounced atmospheric warming (Jin et al., 2011; Zou et al., 2017; Biskaborn et al., 2019). In contrast, Arctic permafrost tends to be colder and more sensitive to warming, resulting in faster thermal responses. These regional differences highlight the importance of accounting for reginal-specific thermal regimes when assessing the vulnerability of permafrost to climate change.

Our study investigated the spatiotemporal dynamics of the permafrost thermal regime in the WKL region of the northwestern QTP from 1980 to 2022. The most pronounced warming in MAGT15m and TTOP occurred between the 1990s and 2000s, whereas ALT changes aligned more closely with LST fluctuations, peaking between the 1980s and 1990s. Furthermore, TTOP showed a faster and more intense response to surface warming than deeper MAGT15m. Furthermore, our simulation results found that approximately 70.98% of permafrost in the region is currently in a warming phase, characterized by initial MAGT values below –2.0 °C and ALT less than 1.5 m, predominantly occurring at elevations above 4800 m a.s.l. and experiencing the most pronounced warming. An additional 17.58% is transitioning toward the zero geothermal gradient stage, while only 11.44% has reached or is progressing toward talik development. These latter zones are typically found at lower elevations (below 4800 m a.s.l.) and are associated with relatively high MAGT15m

(above –1 °C), indicating active degradation, where even modest temperature increases.

Permafrost forms when long-term heat loss at the ground surface exceeds heat input under a persistently cold climate (Wu et al., 2010). In a warming climate, sustained increases in ground surface temperature disrupt this thermal equilibrium. The active layer begins to retain more heat each year, initiating progressive warming from the surface downward and reducing the temperature gradient within the permafrost. During the early stages of warming, permafrost temperatures rise more rapidly than thawing occurs, as much of the incoming energy is consumed raising the frozen soil to its thawing point. This explains why, despite a pronounced warming trend, the areal extent of permafrost in the Western Kunlun region remained relatively stable during the simulation period.

Furthermore, while the regional average LST exhibited a consistent warming trend from 1980 to 2022, substantial interannual and spatial variability was observed. We suggest that periodic cooling events contributed to the formation or expansion of new permafrost in certain areas, producing a delayed and spatially heterogeneous response. This is supported by our simulated permafrost maps, which show a slight increase in permafrost extent between 2010 and 2022, despite overall warming during this period.

Looking ahead, under continued climate warming, MAGTs are projected to rise further. As heat penetrates deeper into the ground, the thermal gradient within the permafrost diminishes and eventually becomes smaller than the geothermal gradient, resulting in upward heat flow from underlying unfrozen ground. This process initiates basal thawing and leads to a gradual upward retreat of the permafrost base. Compared to Arctic and sub-Arctic regions, the QTP exhibits a relatively high geothermal gradient, which contributes to a longer permafrost response time and a slower ground temperature increase (Jin et al., 2011; Zou et al., 2017).

As permafrost temperatures approach 0 °C, ground ice near the permafrost table begins to melt, consuming large amounts of latent heat in a process known as the "zero curtain effect." This phase significantly slows or temporarily halts further warming, extending the duration of thermal inertia and dampening seasonal temperature variations in the upper permafrost. At the same time, geothermal heat from below is primarily consumed in thawing the permafrost from the bottom up.

The zero geothermal gradient stage represents a critical transitional phase in permafrost degradation. During this stage, nearly all surface-derived energy is consumed in melting ground ice, accelerating the downward retreat of the permafrost table. Once seasonal freezing no longer penetrates the permafrost, a talik—an unfrozen zone within permafrost—begin to form and expand. Numerical simulations by Sun et al. (2019) demonstrate that the onset of a talik development marks a tipping point, triggering accelerated thaw and irreversible permafrost degradation until complete loss.

5. Implications for future projections

While the paper effectively documents historical changes, it lacks a forward-looking component. While I realize this is beyond the scope of the paper perhaps some additional point could be added to the discussion to enhance its relevance, such as:

* Discussion how the observed trends might evolve under different climate scenarios.

* Assess the potential for abrupt future permafrost degradation/non-linearly of the system and its implications for infrastructure and carbon release.

* Offer suggestions for integrating MVPM outputs into Earth System Models to improve global climate projections.

**Response:**

Thank you for this thoughtful and constructive suggestion. Understanding the current status of permafrost in the context of its historical evolution is essential for projecting future changes, and we agree that incorporating a forward-looking component enhances the relevance of the paper.

In fact, in our previous work (Sun et al., 2019; Zhao et al., 2022), we used MVPM to simulate and assess permafrost thermal dynamics under various future climate change scenarios. Our modeling results indicated that permafrost degradation, particularly in terms of areal extent, does not follow a linear trajectory, and the response of permafrost temperature to climate warming is not as rapid as projected in many published reports (Guo et al., 2012; Ni et al., 2021). Even under the most extreme warming scenario (RCP8.5), the permafrost table was projected to deepen only gradually. By 2050, permafrost would still remain at a depth of 40 m at both Wudaoliang and Tanggula, two borehole sites located in the continuous permafrost zone characterized by cold ground temperatures and thick permafrost layers. In contrast, at Xidatan, a site located at the lower boundary of the permafrost zone, with warmer ground and a thinner permafrost layer of about 32m, the permafrost base is projected to move upward significantly. Nevertheless, permafrost is still expected to persist at this site through 2100 based on projected changes in the deep permafrost ground temperature, ground ice, and thermal gradients.

Along the northern margin of the permafrost zone on the QTP, MVPM simulations (Zhao et al., 2022) show that MAGT will continue to rise under both RCP and SSP scenarios, with slightly higher warming rates under SSPs. However, the projected differences in areal permafrost extent between the two scenario types were minimal. These findings imply that, although permafrost temperatures are rising rapidly under climate warming, the rate of areal permafrost loss remains relatively slow—a result with important implications for estimating the timing and magnitude of permafrost-related carbon and hydrological feedbacks.

It is worth noting that the slow response of the permafrost thermal regime to future climate warming may exhibit substantial variability in ice-rich permafrost zones (e.g., those containing excess ground ice), largely due to the melting of massive ground ice and the vertical movement of water. These processes exert a strong influence on permafrost thaw trajectories, often leading to landscape changes such as thermokarst pond formation and surface subsidence (Westermann et al., 2016). The associated hydrological dynamics can either accelerate or delay permafrost degradation. Specifically, when meltwater from thawed ice-rich layers drains effectively, both ground subsidence and talik formation are delayed (Westermann et al., 2016). In contrast, if meltwater accumulates at the surface, it can form ponds that enhance heat transfer into the ground, thereby accelerating talik development and intensifying permafrost thaw (Jan et al., 2020). This process holds substantial potential to unlock vast stores of currently frozen organic carbon, particularly greenhouse gases like $CO_2$ and $CH_4$ stored in cold, ice-rich lowlands. As such, thermokarst-related permafrost degradation in a warming climate could significantly amplify the global permafrost carbon–climate feedback (Turetsky et al., 2015).

We agree that integrating MVPM outputs into Earth System Models (ESMs) is essential for improving global climate projections. Based on our findings and previous modeling experience, we offer the following suggestions for improving land surface models (LSMs) within ESMs:

**First, lower boundary conditions:** Many existing LSMs use shallow soil profiles (e.g., less than 10 m) and simplified zero-flux lower boundaries, which are inadequate for capturing deep ground thermal processes. We recommend implementing deeper soil configurations (e.g., 50–100 m) combined with geothermal heat flux boundary conditions to better represent the ground thermal regime over decadal to centennial timescales.

**Second, vertical resolution and initialization:** Accurately simulating permafrost degradation requires high vertical resolution and robust initialization of subsurface thermal and moisture states. We suggest increasing the number of soil layers and conducting long spin-up runs, ideally constrained by in situ data, to reflect the extended thermal memory of permafrost systems (Razavi et al., 2015).

**Third, Ground ice representation**: Ground ice critically influences soil thermal properties and permafrost thaw dynamics. However, current LSMs often rely on overly simplified parameterizations that do not account for excess or segregated ice, nor their formation mechanisms (Lu et al., 2017). We advocate for incorporating sub-grid-scale representations of ground ice distribution, modeling the dynamic formation and melt of different ice types, and explicitly simulating processes such as thaw-induced subsidence and thermokarst ponding (Lee et al., 2014; Westermann et al., 2016; Sun et al., 2022).

**Fourth, Improving the accuracy and resolution of forcing data** necessary as the quality of

atmospheric forcing datasets plays a critical role in the performance of frozen soil simulations.

Finally, high-resolution, observation-constrained MVPM simulations can serve as valuable benchmarks for evaluating and calibrating LSMs across diverse permafrost environments. Additionally, integrating satellite-based remote sensing products for parameter optimization can substantially enhance model realism and reduce simulation uncertainty.

We added implications for future projections and potential impacts on the global permafrost carbon–climate feedback in **"Section 5 Discussion"**, line 689 to line 718:

However, the overall process of permafrost degradation tends to be slow and delayed, particularly in deep permafrost, as confirmed by previous studies showing that permafrost loss, particularly in terms of areal extent, does not follow a linear trajectory, and that permafrost thermal responses to climate warming occur more gradually than suggested by many earlier assessments (Guo et al., 2012; Ni et al., 2021). Even under the extreme RCP8.5 scenario, simulations project only gradual deepening of the permafrost table. For example, by 2050, permafrost is still expected to persist at a depth of 40 m at Wudaoliang and Tanggula—two borehole sites in the continuous permafrost zone, where ground temperatures are cold and permafrost layers are thick. In contrast, at Xidatan, located near the lower boundary of the permafrost zone with a warmer, thinner (~32 m) permafrost layer, the permafrost base is projected to retreat more significantly. Nevertheless, simulations suggest permafrost will still exist at this site through 2100, based on trends in deep ground temperature, ice content, and thermal gradients.

Similar results have been reported for the northern margin of the QTP permafrost zone. MVPM-based modeling (Zhao et al., 2022) indicates that MAGT will continue to rise under gradual warming. Warming rates are projected to be slightly higher under CMIP6 Shared Socioeconomic Pathways (e.g., 0.064°C yr⁻¹ for SSP5-8.5) compared to CMIP5 Representative Concentration Pathways (e.g., 0.060°C yr⁻¹ for RCP8.5), although little difference is projected in areal permafrost extent. These findings suggest that while permafrost temperatures are increasing, spatial loss remains relatively slow—an important consideration for modeling permafrost carbon feedback and related hydrological processes.

It is also important to recognize that the thermal response of permafrost to warming may vary considerably in ice-rich zones, particularly those with excess ground ice. In such areas, thawing of massive ground ice and associated water dynamics significantly shape degradation trajectories, often leading to landscape changes such as surface subsidence and thermokarst pond formation (Westermann et al., 2016). These hydrological feedbacks can either slow or accelerate thaw. Efficient drainage of meltwater delays talik development and surface collapse (Westermann et al., 2016), while surface water accumulation promotes heat transfer and deeper thawing (Jan et al., 2020). These processes increase the potential release of vast stores of frozen organic carbon—

particularly $CO_2$ and $CH_4$—trapped in cold, ice-rich lowlands. Therefore, thermokarst-driven permafrost degradation under continued warming could greatly amplify the global permafrost carbon–climate feedback (Turetsky et al., 2015).

**Section 5.3 Comparison with previous studies",** now includes suggestions for integrating MVPM outputs into Earth System Models to improve global climate projections, line 768 to line 776:

To improve long-term permafrost simulations in ESMs, we recommend the following key developments: i) Enhance bottom boundary conditions by extending soil profiles to 50–100 m and incorporating realistic geothermal heat fluxes to better capture deep ground thermal dynamics; ii) Improve vertical resolution and initialization, including high-resolution soil layering, longer spin-up periods, and calibration using in situ data to better capture the thermal memory of deep permafrost; iii) Advance the representation of ground ice processes, including sub-grid variability, the formation and melt of excess and segregated ice, and thaw-induced surface changes such as thermokarst; iv) Improve the accuracy and resolution of climate forcing data; v) Leverage MVPM outputs to calibrate LSMs, using high-resolution, observation-constrained simulations and remote sensing data to optimize parameters and reduce uncertainties.

**6    Spatial resolution**

The use of 1km spatial resolution may not adequately capture topographic effects (e.g., slope, aspect), which can critically influence local permafrost dynamics. The paper should be further justifying the adequacy of this resolution, particularly in complex mountainous terrain and variables snow cover (controlled by wind redistribution, slope, aspect)

**Response:**

Our choice of a 1 km resolution represents a trade-off between spatial coverage and detail, constrained primarily by the resolution of available forcing datasets, i.e. the satellite-derived LST product. Nonetheless, this resolution marks a substantial improvement over previous large-scale modeling studies that used coarser grids (e.g., 10 km in Zhang et al., 2022; ~62 km in Guo et al., 2012), while providing more regional relevance than purely site-specific simulations (e.g., Sun et al., 2019, 2022).

Importantly, the modeling region (WKL) is located within an alluvial and fluvial basin, and most part of the area is characterized by relatively gentle terrain (see Fig. S3). This reduces the influence of slope and aspect on permafrost processes compared to more rugged mountainous regions. The relatively uniform topography further supports the adequacy of the 1 km resolution for capturing the dominant patterns of permafrost thermal behavior in this area.

We fully acknowledge that a 1 km grid is insufficient to resolve micro-topographic features—such as slope, aspect, and wind-driven snow redistribution—which are known to exert strong local-scale controls on permafrost conditions. Therefore, we added the discussion that our results should be interpreted as first-order approximations of regional permafrost thermal distribution rather than slope-scale assessments.

Despite this limitation, our model successfully reproduces key spatial features of the permafrost thermal regime in the WKL region, including (a) differences in ALT across stratigraphic classes (e.g., alluvial, glarosional, lacustrine, colluvial), and (b) elevation-dependent patterns in ground temperature. These outputs are supported by validation against in situ ground temperature observations from regional monitoring sites and show consistency with published permafrost distribution maps.

Looking ahead, we agree that improving spatial resolution, particularly to better resolve topographic and snow-cover variability will be an important step. Future developments will benefit from higher-resolution remote sensing products and data assimilation techniques that can better account for small-scale heterogeneity in permafrost systems.

[Figure]

**Figure S3. Slope distribution derived from DEM across the WKL permafrost modeling region.**

**In Section 5, Discussion,** we further discuss the limitations of our forcing data, including the resolution constraint of the 1 km grid in capturing micro-topographic variability, line 624 to line 635:

In complex mountainous terrain, a 1 km grid cell is insufficient to capture micro-topographic features such as slope, aspect, and wind-driven snow redistribution—factors that strongly influence local permafrost hydrothermal dynamics. Therefore, our modeling scheme should be considered as a first-order approximation of permafrost thermal distribution, rather than a tool for detailed slopescale assessments in these areas. In addition, resampling coarse-resolution input datasets to match the model resolution introduces uncertainties in the LST reconstruction process. Despite these limitations, the model successfully reproduces regional permafrost thermal patterns in the WKL area, as confirmed by in situ observations and existing permafrost maps. Although constrained by the spatial resolution of satellite-derived LST, the approach performs well in simulating the thermal state and ALT of permafrost, providing valuable insights for remote, data-scarce regions of the western QTP. Future improvements will require the integration of higher-resolution datasets and enhanced representation of sub-grid variability.

7 Figure 5 and Figure 10

These figures display a gridded pattern at approximately 25km resolution, which appears inconsistent with the stated 1km model resolution. the source of this pattern should be clarified and discussed. Is it an artefact of the clustering used? This deserves explanation, especially given the emphasis on fine-scale modelling.

**Response:**

We believe the gridded pattern observed in the figures particularly in Figure 5 is not caused by the clustering methods used in our modeling. Figure 5 shows decadal anomalous LST maps for the WKL permafrost region based on our reconstructed LST dataset from 1980 to 2022. Importantly, no clustering algorithms were applied in generating these maps. Although clustering was used to produce the spatial distribution maps in Figures 6, 7, and 9, no gridded artifacts are present in those figures, further supporting that the patterns in Figure 5 are not artifacts of clustering.

Instead, the gridded appearance is primarily due to uncertainties introduced during the resampling of input datasets used in LST reconstruction. Key input variables to the machine learning model—such as skin temperature (ST, $0.312° \times 0.312°$), fractional cold cover (CFC, $0.25° \times 0.25°$), surface net radiation budget (SRB, $0.25° \times 0.25°$), and leaf area index (LAI, $0.05° \times 0.05°$)—have coarser native resolutions than our model's 1 km target. These were resampled to $1\,km \times 1\,km$ using the nearest-neighbor method. This upscaling process introduces artifacts, particularly from coarser inputs like ST, which has a native resolution of ~25–30 km at these latitudes. While resampling allows analysis at finer spatial scales, it can produce visible grid-like patterns in the output.

We acknowledge this limitation and have added a discussion on the uncertainties introduced by the resampling process during model forcing reconstruction in **Section 5 Discussion**. Please see our response under the comment titled 'spatial resolution' for further details.

**Minor Comments**

\* Line 283: Typo: "account for only 28.02% of the total model grid cells, **remarkable** reducing computation time"

**Response:**

Yes, we have corrected this typo in the revised manuscript. The revised sentence now reads: "account for only 28.02% of the total model grid cells, remarkably reducing computation time."

\* Line 300: Typo: Section title should read "3.3 Filed investigation and borehole monitoring datasets"

**Response:**

Yeah, thanks, we have made the revision. We have also carefully checked and corrected similar typos throughout the text.

*Figure 7: Missing units on the lagend: overall presentation could be improved.

**Response:**

In the revised manuscript, we have added the units to the legend in Figure 7

[Figure]

*Line 538: Add space in "with 74.20%" to read "with 74.20%"

**Response:**

We have added space in the revised manuscript.

\* Figure 11: Legend and labelling could be enhanced for readability.

**Response:**

We have revised Figure 11 to enhance readability by improving the resolution of the legend and increasing the font size of the labels for better clarity. We hope these improvements make the figure easier to interpret.

[Figure]

\* Line 600: Discussion: "Most previous evaluations indicated that soil temperature products derived from atmospheric circulation model or ESMs, which typically have coarse resolutions (~300km) …" Consider replacing "ESMs" with "GCMs" for clarity. Or revise to more moderate number typical of historical forcing datasets such as ERA5 (25km) or ERA5-Land (9km) as you do not use GCMs in this study.

**Response:**

Yeah, exactly. We agree that "GCMs" would be a more accurate term than "ESMs" in this context and have revised the sentence accordingly. The revised sentence now reads:

"Previous studies have shown that coarse-resolution soil temperature products from atmospheric reanalysis datasets—such as ERA-Interim ($0.125° \times 0.125°$) and ERA5-Land ($0.1° \times 0.1°$)—as well as assimilated products like the Chinese meteorological forcing dataset CLDAS ($0.0625° \times 0.0625°$), exhibit substantial uncertainties when applied to the QTP, particularly in permafrost regions (Hu et al., 2019; Qing et al., 2020; Yang et al., 2020)."

\* Line 613: Clarify what is mean by "compared to in situ measurements, we found a slight cold bias in our reconstructed LST series, averaging approximately -0.80°C"-is this computed over the entire period? Please specify the period.

**Response:**

The reconstructed LST was evaluated against monthly in situ observations from the TSH AWS for the period 2016–2018. The average bias across all months during this period was approximately –0.80°C. A systematic cold bias is evident in the mean annual cycle of LST, particularly during the summer months of July, August, and September, as shown in Figure 1.

* Line 858: Typo: "experiencing recover or degradation" should be revised to "experiencing recovery or degradation."

**Response:**

The revision has been made in the revised manuscript.

* Line 185: How and what in situ measurement integrated? Please add additional clarification here. (this dataset was created by integrating in situ observations with satellite-based LST from the Moderate Resolution Imaging Spectroradiometer (MODIS).)

**Response:**

To estimate daily mean LST values from Aqua and Terra's instantaneous daytime and nighttime observations, Zou et al. (2014, 2017) developed a multi-stepwise statistical model based on GST data from three AWS located in typical permafrost regions of the central QTP. These relationships capture actual climate conditions corresponding to satellite overpass times. The resulting empirical correction model was then applied across the entire QTP permafrost zone to upscale and generate reliable LST estimates. We therefore believe that the modified MODIS LST dataset developed by Zou et al. (2017) partially accounts for the influence of surface conditions—including snow cover, vegetation, and cloud cover—through the use of a cloud-gap filling algorithm and the incorporation of AWS observations from representative permafrost regions in the central QTP. Please see details in response of secondary question "Use of MODIS LST as forcing data"

**Reference**

Alexeev, V. A., Nicolsky, D. J., Romanovsky, V. E., Lawrence, D. M.: An Evaluation of Deep Soil Configurations in the CLM3 for Improved Representation of Permafrost: How Deep Should the CLM3 Soil Layer Be? Geophysical Research Letters 34, no. 9, L09502, https:// doi.org/ 10. 1029/ 2007G L029536, 2007.

Burn, C.R., Zhang, Y.: Permafrost and climate change at Herschel Island (Qikiqtaruq), Yukon Territory, Canada. J. Geophys. Res. Earth Surf. 114(F2), https://doi.org/10.1029/2008JF001087, 2009.

Buteau, S., Fortier, R., Delisle, G., Allard, M.: Numerical simulation of the impacts of climate

warming on a permafrost mound. Permafr Periglac Process., 15(1):41-57. https://doi.org/10.1002/ppp.474, 2004.

Biskaborn, B. K., Smith, S. L., Noetzli, J., Matthes, H., Vieira, G., Streletskiy, D. A., Schoeneich, P., Romanovsky, V. E., Lewkowicz, A. G., Abramov, A., Allard, M., Boike, J., Cable, W. L., Christiansen, H. H., Delaloye, R., Diekmann, B., Drozdov, D., Etzelmüller, B., Grosse, G., Guglielmin, M., Ingeman-Nielsen, T., Isaksen, K., Ishikawa, M., Johansson, M., Johannsson, H., Joo, A., Kaverin, D., Kholodov, A., Konstantinov, P., Kröger, T., **Lambiel, C., Lanckman, J.-P., Luo, D., Malkova, G., Meiklejohn, I., Moskalenko, N., Oliva, M., Phillips, M., Ramos, M., Sannel, A. B. K., Sergeev, D., Seybold, C., Skryabin, P., Vasiliev, A., Wu, Q., Yoshikawa, K., Zheleznyak, M., and Lantuit, H.:** Permafrost is warming at a global scale, *Nat. Commun.*, **10**, 264, https://doi.org/10.1038/s41467-018-08240-4, 2019.

Cao, Z., Nan, Z., Hu, J., Chen, Y., and Zhang, Y.: A new 2010 permafrost distribution map over the Qinghai–Tibet Plateau based on subregion survey maps: a benchmark for regional permafrost modeling, Earth Syst. Sci. Data, 15, 3905–3930, https://doi.org/10.5194/essd-15-3905-2023, 2023.

Che, T., Xin, L., Jin, R., Armstrong, R., and Zhang, T.: Snow depth derived from passive microwave remote-sensing data in China, Ann. Glaciol., 49, 145–154, https://doi.org/10.3189/172756408787814690, 2008.

Chen, D., Rojas, M., Samset, B.H.: Framing, Context, and Methods, in Climate Change, The Physical Science Basis (Cambridge, UK: Cambridge University Press): 147–286, 2021.

Chen, H., Nan, Z., Zhao, L., Ding, Y., Chen, J., Pang, Q.: Noah Modelling of the Permafrost Distribution and Characteristics in the West Kunlun Area, Qinghai-Tibet Plateau, China. Permafr. Periglac. Process. 26, 160–174, https://doi.org/10.1002/ppp.1841, 2015.

D. de Vries, Thermal Properties of Soils, in Physics of the Plant Environment, eds. W. R. Wijk and A. J. W. Borghorst (Amsterdam, The Netherlands: North-Holland): 210–235, 1963.

Dobiński, W., and Marek K.: Permafrost Base Degradation: Characteristics and Unknown Thread with Specific Example from Hornsund, Svalbard, Front. Earth Sci. 10:802157, doi: 10.3389/feart.2022.802157, 2022.

Etzelmüller, B., Schuler, T.V., Isaksen K, Christiansen, H., Farbrot, H, Benestad, R.: Modeling the temperature evolution of Svalbard permafrost during the 20th and 21st century. Cryosphere.5(1):67-79. https://doi.org/10.5194/tc-5-67-2011, 2011.

Guo, D., Wang, H., and Li, D.: A projection of permafrost degradation on the Tibetan Plateau during the 21st century, 117, D05106, J. Geophys. Res. Atmos., 117, D05106, https://doi.org/10.1029/2011JD016545, 2012.

He, H., N, G., Flerchinger, Y., Kojima., Dyck, M.: A Review and Evaluation of 39 Thermal Conductivity Models for Frozen Soils, Geoderma, 382: 114694, https://doi.org/10.1016/j.geoderma.2020.114694, 2021.

Hengl, T., de Jesus, J.M., Heuvelink, G.B., Gonzalez, M.R., Kilibarda, M., Blagoti´c, A., Shangguan, W., Wright, M.N., Geng, X., Bauer-Marschallinger, B.: SoilGrids250m: Global gridded soil information based on machine learning. PLoS. One 12 (2), e0169748, https://doi.org/10.1371/journal.pone.0169748, 2017.

Hermoso de Mendoza, I., Beltrami, H., MacDougall, A. H., and Mareschal, J.-C.: Lower boundary conditions in land surface models – effects on the permafrost and the carbon pools: a case study with CLM4.5, Geosci. Model Dev., 13, 1663–1683, https://doi.org/10.5194/gmd-13-1663-2020, 2020.

Hipp, T., Etzelmüller, B., Farbrot, H., Schuler, TV.: Modelling borehole temperatures in southern Norway–insights into permafrost dynamics during the 20th and 21st century. Cryosphere, 6(3):553-571. https://doi.org/10.5194/tc-6-553-2012, 2012.

Hu, G., Zhao, L., Li, R., Park, H., Wu, X., Su, Y., Guggenberger, G., Wu, T., Zou, D., Zhu, X., Zhang, W., Wu, Y., and Hao, J: Water and heat coupling processes and its simulation in frozen soils: Current status and future research directions, Catena, 222, 106844, 985 https://doi.org/10.1016/j.catena.2022.106844, 2023.

Hu, G., Zhao, L., Li, R., Wu, X., Wu, T., Xie, C., Zhu, X., and Su, Y.: Variations in soil temperature from 1980 to 2015 in permafrost regions on the Qinghai-Tibetan Plateau based on observed and reanalysis products, Geoderma, 337, 893-905, https://doi.org/10.1016/j.geoderma.2018.10.044, 2019.

Hu, G., Zhao, L., Wu, X., Li, R., Wu, T., Xie, C., Pang, Q., Xiao, Y., Li, W., Qiao, Y., Shi, J.: Modeling permafrost properties in the Qinghai-Xizang (Tibet) Plateau. Science China: Earth Sciences, 58: 2309–2326, doi: 10.1007/s11430-015-5197-0, 2015.

Hu, J., Zhao, L., Wang, C., Hu, G., Zou, D., Xing, Z., Jiao, M., Qiao, Y., Liu, G., Du, E.: Applicability evaluation and correction of CLDAS surface temperature products in permafrost region of Qinghai-Tibet Plateau, Climate Change Research, 20 (1): 10-25, 2024.

IPCC. Climate change 2021: the physical science basis, https://www.ipcc.ch/report/ar6/wg1/downloads/report/IPCC_AR6_WGI_Full_Report.pdf., 2021.

Janatian, N., Sadeghi, M., Sanaeinejad, S. H., Bakhshian, E., Farid, A., Hasheminia, S. M., Ghazanfari, S.: A statistical framework for estimating air temperature using MODIS land surface temperature data. International Journal of Climatology, 37(3), 1181-1194, 2017.

Jiao, M., Zhao, L., Wang, C., Hu, G., Li, Y., Zhao, J., Zou, D., Xing, Z., Qiao, Y., Liu, G.: Spatiotemporal Variations of Soil Temperature at 10 and 50 cm Depths in Permafrost Regions along the Qinghai-Tibet Engineering Corridor. Remote Sens. 2023, 15, 455. https://doi.org/10.3390/rs15020455, 2023.

Jin, H., Luo, D., Wang, S., Lü, L., and Wu, J.: Spatiotemporal variability of permafrost degradation on the Qinghai-Tibet Plateau, Sci. Cold Arid Reg., 3, 281–305, DOI:

10.3724/SP.J.1226.2011.00281, 2011.

Lee, H., Swenson, S., Slater, A., and Lawrence, D.: Effects of excess ground ice on projections of permafrost in a warming climate, Environ. Res. Lett., 9, 124006, https://doi.org/10.1088/1748-9326/9/12/124006, 2014.

Li, K., Chen, J., Zhao, L., Zhang, X., Pang, Q., Fang, H., Liu, G.: Permafrost distribution in typical area of west Kunlun Mountains derived from a comprehensive survey (in Chines with English abstract), J. GLACIOL.,2012.

Li, N., Cuo, L., Zhang, Y., and Ding, J.: The synthesis of potential factors contributing to the asynchronous warming between air and shallow ground since the 2000s on the Tibetan Plateau, Geoderma,441, 116753, https://doi.org/10.1016/j.geoderma.2023.116753, 2024.

Li, Q., Sun, S., 2015. The Simulation of Soil Water Flow and Phase Change in Vertically Inhomogeneous Soil in Land Surface Models. Chin. J. Atmos. Sci. 39 (4), 827–838.

Li, S., Cheng, G., and Guo, D.: The future thermal regime of numerical simulating permafrost on the Qinghai-Xizang (Tibet) Plateau, China, under a warming climate, Sci. China D, 39, 434–441, 1996.

Lu, J., Zhang, M., Zhang, X., Pei, W., 2017. Review of the coupled hydro-thermomechanical interaction of frozen soil. J. Glaciol. Geocryol. 39 (1), 102–111.

Lunardini, V.: Permafrost Formation Time. CRREL Report 95-8, US Army Corps of Engineers, Cold Regions Research and Engineering Laboratory, 1995.

Ni, J., Wu, T., Zhu, X., Hu, G., Zou, D., Wu, X., Li, R., Xie, C., Qiao, Y., Pang, Q., Hao, J., and Yang, C.: Simulation of the present and future projection of permafrost on the Qinghai-Tibet Plateau with statistical and machine learning models, J. Geophys. Res.-Atmos., 126, e2020JD033402, https://doi.org/10.1029/2020JD033402, 2021.

Nicolsky, D. J., Romanovsky, V. E., Panda, S. K., Marchenko, S. S., Muskett, R. R.: Applicability of the ecosystem type approach to model permafrost dynamics across the Alaska North Slope, Journal of Geophysical Research: Earth Surface, 122(1), 50-75, https://doi.org/10.1002/2016JF003852, 2017.

Nitzbon, J., Westermann, S., Langer, M., Martin, L. C., Strauss, J., Laboor, S., Boike, J. Fast response of cold ice-rich permafrost in northeast Siberia to a warming climate. Nature communications, 11(1), 2201, 2020

Pu, Z., Xu, L., and Salomonson, V. V.: MODIS/Terra observed seasonal variations of snow cover over the Tibetan Plateau, Geophys. Res. Lett., 34, L06706, https://doi.org/10.1029/2007GL029262, 2007.

Qin, D., Liu, S., and Li, P.: Snow cover distribution, variability, and response to climate change in western China, J. Clim., 19, 1820–1833, https://doi.org/10.1175/JCLI3694.1, 2006.

Razavi, S., Elshorbagy, A., Wheater, H., Sauchyn, D.: Toward understanding nonstationarity in climate and hydrology through tree ring proxy records. Water Resour. Res. 51 (3), 1813–1830, 2015.

Riseborough, D., Shiklomanov, N., Etzelmüller, B., Gruber, S., Marchenko, S.: Recent advances in permafrost modelling. Permafr. Periglac. Process.:19(2):137-156. https://doi.org/10.1002/ppp.615, 2009.

Shangguan, W., Dai, Y., Liu, B., Zhu, A., Duan, Q., Wu, L., Ji, D., Ye, A., Yuan, H., Zhang, Q., Chen, D., Chen, M., Chu, J., Dou, Y., Guo, J., Li, H., Li, J., Liang, L., Liang, X., Liu, H., Liu, S., Miao, C., Zhang, Y.: A China data set of soil properties for land surface modeling. J. Adv. Model. Earth Syst. 5 (2), 212–224, https://doi.org/10.1002/jame.20026, 2013.

Smith, M., Riseborough, D.: Permafrost sensitivity to climatic change. In Proceedings, 4th International Conference on Permafrost, Vol. 1. Fairbanks, Alaska, National Academy Press: Washington, DC; 1178–1183, 1983.

Sun, Z., Zhao, L., Hu, G., Qiao, Y., Du, E., Zou, D., Xie, C., 2020. Modeling permafrost changes on the Qinghai-Tibetan plateau from 1966 to 2100: A case study from two boreholes along the Qinghai-Tibet engineering corridor. Permafr. Periglac. Process. 31 (1), 156–171.

Sun, Z., Zhao, L., Hu, G., Zhou, H., Liu, S., Qiao, Y., Du, E., Zou, D., and Xie, C.: Effects of Ground Subsidence on Permafrost Simulation Related to Climate Warming, Atmosphere,15(1), 12, https://doi.org/ 10.3390/atmos15010012, 2023.

Sun, Z., Zhao, L., Hu, G., Zhou, H., Liu, S., Qiao, Y., Du, E., Zou, D., and Xie, C.: Numerical simulation of thaw settlement and permafrost changes at three sites along the Qinghai-Tibet 1145 Engineering Corridor in a warming climate, Geophys. Res. Lett., 49, e2021GL097334, https://doi.org/10.1029/2021GL097334, 2022.

Sun, Z.: Simulation of permafrost dynamics and thaw settlement along the Qinghai-Tibetan Engineering Corridor, University of Chinses Academy of Science, 2021(Chinese with English abstract).

Turetsky, M. R., Benscoter, B., Page, S., Rein, G., Van Der Werf, G. R., Watts, A.: Global vulnerability of peatlands to fire and carbon loss. Nature Geoscience,8(1), 11-14, 2015

Wang, T., Wang, N., and Li, S.: Map of the glaciers, frozen ground and desert in China, 1 V 4 000 000, Chinese Map Press, Beijing, China, 2006.

Wang, X., Zhong, L., Ma, Y.: Estimation of 30 m land surface temperatures over the entire Tibetan Plateau based on Landsat-7 ETM+ data and machine learning methods. International Journal of Digital Earth,15(1), 1038-1055, 2022.

Westermann, S., Langer, M., Boike, J., Heikenfeld, M., Peter, M., Etzelmüller, B., and Krinner, G.: Simulating the thermal regime and thaw processes of ice-rich permafrost ground with the land-surface model CryoGrid 3, Geosci. Model Dev., 9, 523–546, https://doi.org/10.5194/gmd-9-

523-2016, 2016.

Westermann, S., Østby, T. I., Gisnås, K., Schuler, T. V., and Etzelmüller, B.: A ground temperature map of the North Atlantic permafrost region based on remote sensing and reanalysis data, The Cryosphere, 9, 1303–1319, https://doi.org/10.5194/tc-9-1303-2015, 2015.

Westermann, S., Peter, M., Langer, M., Schwamborn, G., Schirrmeister, L., Etzelmüller, B., and Boike, J.: Transient modeling of the ground thermal conditions using satellite data in the Lena River delta, Siberia, The Cryosphere, 11, 1441–1463, https://doi.org/10.5194/tc-11-1441-2017, 2017.

Westermann, S., Schuler, T. V., Gisnås, K., and Etzelmüller, B.: Transient thermal modeling of permafrost conditions in Southern Norway, The Cryosphere, 7, 719–739, https://doi.org/10.5194/tc-7-719-2013, 2013.

Wu, J., Sheng, Y.,Wu, Q., andWen, Z.: Processes and modes of permafrost degradation on the Qinghai-Tibet Plateau, Sci. China D, 53, 150–158, https://doi.org/10.1007/s11430-009-0198-5, 2010.

Wu, Q. and Zhang, T.: Recent permafrost warming on the Qinghai-Tibetan Plateau, J. Geophys. Res., 113, 1–22, 2008.

Xing, Z., Zhao, L., Fan, L., Hu, G., Zou, D., Wang, C., Liu, S., Du, E., Xiao, Y., Li, R., Liu, G., Qiao, Y., and Shi, J.: Changes in the ground surface temperature in permafrost regions along the Qinghai–Tibet engineering corridor from 1900 to 2014: a modified assessment of CMIP6, Adv. Clim. Chang. Res.,14(1), 85-96, https://doi.org/10.1016/j.accre.2023.01.007, 2023.

Xu, Y., Knudby, A., Shen, Y., Liu, Y.: Mapping monthly air temperature in the Tibetan Plateau from MODIS data based on machine learning methods. IEEE journal of selected topics in applied earth observations and remote sensing, 11(2), 345-354, 2018.

Xu, Y., Shen, Y., and Wu, Z.: Spatial and Temporal Variations of Land Surface Temperature Over the Tibetan Plateau Based on Harmonic Analysis, Mt. Res. Dev., 33, 85–94, 2013.

Yan, D., Ma, N., Zhang, Y.: Development of a fine-resolution snow depth product based on the snow cover probability for the Tibetan Plateau: Validation and spatial–temporal analyses. Journal of Hydrology,604, 127027,2022.

Yang, S., Li, R., Wu, T., Hu, G., Xiao, Y., Du, Y., Qiao, Y.: Evaluation of reanalysis soil temperature and soil moisture products in permafrost regions on the Qinghai-Tibetan Plateau. Geoderma, 377, 114583, 2020.

Yang, Y., You, Q., Jin, Z., Zuo, Z., Zhang, Y., Kang, S.: The reconstruction for the monthly surface air temperature over the Tibetan Plateau during 1901–2020 by deep learning. Atmospheric Research, 285, 106635, 2023.

Yao, T., Xue, Y., Chen, D., Chen, F., Thompson, L., Cui, P., Koike, T., Lau, W. K., Lettenmaier, D.,

Mosbrugger, V., Zhang, R., Xu, B., Dozier, J., Gillespie, T., Gu, Y., Kang, S., Piao, S., Sugimoto, S., Ueno, K., Wang, L., Wang, W., Zhang, F., Sheng, Y., Guo, W., , Yang, X., Ma, Y., Shen, S. S. P., Su, Z., Chen, F., Liang, S., Liu, Y., Singh, V. P., Yang, K., Yang, D., Zhao, X., Qian, Y., Zhang, Y., and Li, Q.: Recent Third Pole's Rapid Warming Accompanies Cryospheric Melt and Water Cycle Intensification and Interactions between Monsoon and Environment: Multidisciplinary Approach with Observations, Modeling, and Analysis, B. Am. Meteorol. Soc., 100, 423-444, https://doi.org/10.1175/BAMS-D-17-0057.1, 2019.

Yershov, E.: Principles of Geocryology, Lanzhou University Press, Lanzhou, China, ISBN 9787311048570, 2016 (in Chinse).

You, Q., Cai, Z., Pepin, N., Chen, D., Ahrens, B., Jiang, Z., Wu, F., Kang, S., Zhang, R., Wu, T., Wang, P., Li, M., Zou, Z., Gao, Y., Zhai, P., and Zhang, Y.: Warming amplification over the Arctic Pole and Third Pole: Trends, mechanisms and consequences. Earth Sci. Rev.,217, 103625, https://doi.org/10.1016/j.earscirev.2021.103625, 2021.

Zhang, G., Nan, Z., Hu, N., Yin, Z., and Zhao, L., Cheng, G., and Mu, C.: Qinghai-Tibet Plateau permafrost at risk in the late 21st Century, Earth's Future, 10, e2022EF002652, https://doi.org/10.1029/2022EF002652, 2022.

Zhao L, Sheng Y. Permafrost and environment changes on the Qinghai-Tibetan Plateau. Beijing, China: Science Press.; 2019.

Zhao, J., Zhao, L., Sun, Z., Niu, F., Hu, G., Zou, D., Liu, G., Du, E., Wang, C., Wang, L., Qiao, Y., Shi, J., Zhang, Y., Gao, J., Wang, Y., Li, Y., Yu, W., Zhou, H., Xing, Z., Xiao, M., Yin, L., and Wang, S.: Simulating the current and future northern limit of permafrost on the Qinghai–Tibet Plateau, The Cryosphere, 16, 4823–4846, https://doi.org/10.5194/tc-16-4823-2022, 2022.

Zhao, L., Ding, Y., Liu, G., Wang, S., and Jin, H.: Estimates of the reserves of ground ice in permafrost regions on the Tibetan plateau, J. Glaciol. Geocryol., 32, 1–9, 2010.

Zhao, L., Hu, G., Liu, G., Zou, D., Wang, Y., Xiao, Y., Du, E., Wang, C., Xing, Z., Sun, Z., Zhao, Y., Liu, S., Zhang, Y., Wang, L., Zhou, H., and Zhao, J.: Investigation, Monitoring, and Simulation of Permafrost on the Qinghai-Tibet Plateau: A Review, Permafrost Periglac. Process., 35:412–422, https://doi.org/10.1002/ppp.2227, 2024.

Zhao, L., Zou, D. Hu, G., Du, E., Pang, Q., Xiao, Y., Li, R., Sheng, Y., Wu, X., Sun, Z., Wang, L., Wang, C., Ma, L., Zhou, H., and Liu, S.: Changing climate and the permafrost environment on the Qinghai–Tibet (Xizang) Plateau, Permafrost Periglac., 31, 396– 405, https://doi.org/10.1002/ppp.2056, 2020.

Zou, D., Zhao, L., Sheng, Y., Chen, J., Hu, G., Wu, T., Wu, J., Xie, C., Wu, X., Pang, Q., Wang, W., Du, E., Li, W., Liu, G., Li, J., Qin, Y., Qiao, Y.,Wang, Z., Shi, J., and Cheng, G.: A new map of permafrost distribution on the Tibetan Plateau, The Cryosphere, 11, 2527–2542, https://doi.org/10.5194/tc-11-2527-2017, 2017.

Zou, D., Zhao, L., Wu, T., Wu, X., Pang, Q., and Wang, Z.: Modeling ground surface temperature

by means of remote sensing data in high-altitude areas: test in the central Tibetan Plateau with application of moderate-resolution imaging spectroradiometer Terra/Aqua land surface temperature and ground based infrared radiometer, J. Appl. Remote Sens., 8, 083516, https://doi.org/10.1117/1.JRS.8.083516, 2014.

---

## Author Response (AR2)

Dear Editor and Reviewers,

We sincerely appreciate the time and effort you invested in evaluating our work, and we are especially grateful for your recognition of our study. Your thoughtful suggestions have undoubtedly improved the clarity and overall quality of the manuscript. Accordingly, in response to the reviewers' feedback, we have carefully revised the manuscript and addressed all comments in detail.

1.Please ensure that every abbreviation is defined at first use; the abbreviation "ALT" appears before its definition in Line 304。

We have carefully checked the entire manuscript and ensured that all abbreviations are defined upon their first use. In particular, we have revised the definition of "ALT" accordingly.

2.As a small addition to strengthen the manuscript, it may be worth citing one of the many studies discussing this issue, for example:

Orsolini, Y., Wegmann, M., Dutra, E., Liu, B., Balsamo, G., Yang, K., de Rosnay, P., Zhu, C., Wang, W., Senan, R., and Arduini, G. (2019): Evaluation of snow depth and snow cover over the Tibetan Plateau in global reanalyses using in situ and satellite remote sensing observations, The Cryosphere, 13, 2221–2239. https://doi.org/10.5194/tc-13-2221-2019

We have added references related to snow depth and snow cover over the Qinghai−Tibet Plateau (Orsolini et al., 2019; Cao et al., 2020) in" **Section 5.1 Applicability of the forcing data**" to better contextualize our findings. The revised text in Line 620 now reads:

"ERA5-Land skin temperature exhibits a notable winter cold bias over the QTP, likely due to overestimated snow cover persistence and excessive snowfall in the ERA5-Land snow reanalysis product. These factors enhance surface albedo, leading to exaggerated surface cooling—a bias well documented by Cao et al. (2020) and Orsolini et al. (2019)."

Reference

Orsolini, Y., Wegmann, M., Dutra, E., Liu, B., Balsamo, G., Yang, K., de Rosnay, P., Zhu, C., Wang, W., Senan, R., and Arduini, G. (2019): Evaluation of snow depth and snow cover over the Tibetan Plateau in global reanalyses using in situ and satellite remote sensing observations, The Cryosphere, 13, 2221–2239. https://doi.org/10.5194/tc-13-2221-2019

Cao, B., Gruber, S., Zheng, D., and Li, X.: The ERA5-Land soil temperature bias in permafrost

regions, The Cryosphere, 14, 2581–2595, https://doi.org/10.5194/tc-14-2581-2020, 2020.

Thank you once again for your time, effort, and support throughout the review process.

Best regards,

Lin Zhao

On behalf of all co-authors